# Chromothripsis-associated chromosome 21 amplification orchestrates transformation to blast-phase MPN through targetable overexpression of *DYRK1A*

Chromothripsis, the chaotic shattering and repair of chromosomes, is common in cancer. Whether chromothripsis generates actionable therapeutic targets remains an open question. In a cohort of 64 patients in blast phase of a myeloproliferative neoplasm (BP-MPN), we describe recurrent amplification of a region of chromosome 21q ('chr. 21amp') in 25%, driven by chromothripsis in a third of these cases. We report that chr. 21amp BP-MPN has a particularly aggressive and treatment-resistant phenotype. *DYRK1A*, a serine threonine kinase, is the only gene in the 2.7-megabase minimally amplified region that showed both increased expression and chromatin accessibility compared with non-chr. 21amp BP-MPN controls. *DYRK1A* is a central node at the nexus of multiple cellular functions critical for BP-MPN development and is essential for BP-MPN cell proliferation in vitro and in vivo, and represents a druggable axis. Collectively, these findings define chr. 21amp as a prognostic biomarker in BP-MPN, and link chromothripsis to a therapeutic target.

The term chromothripsis describes a massive genomic rearrangement event, caused by shattering and haphazard realignment of a chromosomal region, that is pervasive across solid tumors and associated with an adverse prognosis[1–3]. Chromothripsis is associated with defective DNA repair pathways, including *TP53* mutation (m*TP53*), although 60% of chromothripsis cases occur in *TP53* wild-type (WT) tumors[1]. While oncogene amplification and tumor suppressor gene loss are well-described consequences of chromothripsis[1], the mechanism and impact on disease biology conferred by specific chromothripsis events have not been elucidated. Consequently, whether chromothripsis itself constitutes an actionable and therapeutically targetable molecular event remains an open question.

BP-MPN is associated with a treatment refractory and typically rapidly fatal disease course, with a distinct molecular and clinical profile when compared with de novo acute myeloid leukemia (AML)[4,5]. Conventional AML treatment approaches are ineffective and few patients are cured by allogeneic stem cell transplant[6,7]. There is consequently a major unmet need to identify new treatments.

The mutational landscape associated with progression to BP-MPN is well-described, with frequent presence of multiple 'high-risk' mutations that are associated with a poor prognosis in chronic phase MPN, including *ASXL1*, *IDH1/2*, *RAS*, *RUNX1*, spliceosome mutations and a particularly high incidence of *TP53* pathway alterations[5,7]. Furthermore, while copy number alterations (CNAs) and structural variants (SVs) are infrequent in chronic phase MPN, these events occur with a high frequency in BP-MPN. This includes recurrent regions of deletions of 17p or 5q, monosomy 7, trisomy 8, 12q rearrangements and gains of chr. 1q (refs. [8–11]). Copy number-neutral loss of heterozygosity (CNN-LOH) events affecting *JAK2* and *TP53* loci on 9p and 17p, respectively, are also well-described[9,10,12]. However, aside from *JAK2* and *IDH1/2* mutations[13–15], few of these molecular events are associated with known actionable therapeutic targets.

Due to the long latency between chronic and blast phase in the majority of patients, MPN has long been studied as an exemplar tractable model of genetic evolution in cancer[16–19]. Although chromothripsis has been reported to occur in ~7% of de novo AML[20], chromothripsis

✉e-mail: charlotte.brierley@imm.ox.ac.uk; john.crispino@stjude.org; adam.mead@imm.ox.ac.uk

has not been described in BP-MPN, and the contribution of recurrent chromosome rearrangements to transformation in MPN remains poorly delineated. Herein, we set out to identify the prevalence and downstream consequences of chromothripsis-associated chromosome 21 amplification in BP-MPN, and to determine how these events contribute to leukemic progression.

## Chr. 21amp is a recurrent and adverse genomic event in BP-MPN

We studied a cohort of 64 patients with BP-MPN with a median follow-up of 6.2 months (range 0–48) and a median age of 70 yr (range 29–84) (Fig. 1a and Supplementary Table 1). We performed integrated copy number (CN) and mutation profiling by single nucleotide polymorphism (SNP) array karyotyping and targeted sequencing. Analysis of SNP array data using MoCha[21,22], identified 344 CNAs in 54 of 64 (84.4%) cases with a median of 3.5 events (range 0–23). Of these, 24 (7%) were CNN-LOH events, 103 (30%) were gains and 217 (63%) losses (Extended Data Fig. 1a,b). The majority of recurrent events had been previously described, including chr. 1q gain (in 10 of 64 cases, 16%), monosomy 7 (6 of 64, 9%), partial or complete loss of chr. 5q (17 of 64, 27%) and loss of 17p (10 of 64, 16%; Extended Data Fig. 1a). CNN-LOH on chr. 9p occurred in six *JAK2* mutant cases, and on chr. 17p in three *TP53* mutant cases.

In total, 11 of 64 (17.2%) patients showed at least one chromothripsis event, a higher rate than the ~7% incidence demonstrated in AML[1,20,23]. As expected[1], there was a positive association between the presence of chromothripsis and m*TP53*/loss (m*TP53* and/or loss $n = 29$ (45.3%), chromothripsis 10 of 29 (34.5%), $P = 0.002$).

A number of patients (5 of 64, 8%) had evidence of chromothripsis affecting chromosome 21, with focal and multiple amplifications of Chr. 21q22-23 (Fig. 1b). Three of five (60%) cases harbored further chromothriptic events involving other chromosomes (chr. 19p, chr. 17p and chr. 22p, respectively). A further 11 (totaling 16 of 64, 25%) had a regional CN gain event over chromosome 21q, resulting in amplification of chr. 21q22 ('chr. 21amp') in a quarter of patients. Overlaying of samples enabled identification of the shared minimally amplified region (MAR) across all 16 cases (Fig. 1c). This spanned 2.7 megabases and contained 24 genes, with a median CN of 3.5 (range 2.7–8.3) (Fig. 1d). The amplification event affecting chr. 21 was significantly recurrent across the cohort (GISTIC2.0, $Q = 0.00059$; Fig. 1e) and constituted the most common chromosome amplification event. Patients with chr. 21amp had a greater number of non-chr. 21 CNAs compared with those without (median 6.5, range 4–15 versus median 1, range 0–16, $P < 0.001$ (Wilcoxon rank-sum test); Fig. 1f). Chr. 21amp occurred with a range of co-mutations and clinical phenotypes, age and sex, and significantly co-occurred with m*TP53* (Fig. 1g and Extended Data Fig. 1c).

Patients with chr. 21amp had a particularly aggressive clinical phenotype with none of the patients surviving 1 yr, compared with 41.8% (95% confidence interval (95% CI) 28.9–60.5%) of non-chr. 21amp cases ($P = 0.00007$; Fig. 1h). The adverse impact of chr. 21amp on overall survival was maintained on multivariate analysis when adjusting for age, sex and high-risk molecular risk, including m*TP53* status (hazard ratio (HR) 4.9, $P < 0.001$; see Supplementary Table 2 for Cox regression analysis).

Together, these data identify chr. 21amp as a previously unrecognized and prevalent CN event occurring in BP-MPN that is associated with an adverse clinical outcome.

## Chr. 21amp also confers an adverse prognosis in de novo AML

To understand whether enrichment for chr. 21amp occurred more broadly in AML, we interrogated two published AML cohorts. The incidence of chr. 21amp was 9 of 191 (4.5%) in The Cancer Genome Atlas (TCGA) cohort, and 117 of 3,653 (3.3%) in the UK trials cohort[24,25]. As in our BP-MPN cohort, in the de novo AML context, chr. 21amp also co-occurred significantly with *TP53* mutations or deletions (31 of 117,

26.5% versus 7.7%, $P < 0.001$, Fisher's exact test; Extended Data Fig. 1d) and complex karyotype (65 of 117, 55.6% versus 8.8%, $P < 0.001$, Fisher's exact test), and was associated with adverse survival in both univariable (HR 1.59 (95% CI 1.29–1.97), $P < 0.001$) and multivariable analyses after adjusting for m*TP53* status (HR 1.3 (95% CI 1.1–1.7), $P = 0.009$; Extended Data Fig. 1e,f). The TCGA cohort was underpowered for a survival analysis (Extended Data Fig. 1g). These data confirm that chr. 21amp is less common in de novo AML than in BP-MPN (3–5% versus 25%), but where it occurs, it correlates with an adverse prognosis.

## Whole genome sequencing of chromothripsis-associated chr. 21amp

To determine the precise genetic architecture of the SV events that led to chr. 21 amplification, and to confirm that this is driven by bona fide chromothripsis events in some cases, we performed high-depth whole genome sequencing (WGS) in five chr. 21amp cases, to a median coverage of 81× (range 77–86) and purity 79% (range 58–88%) (Fig. 2a–f and Extended Data Fig. 2a–d)[26]. Each case demonstrated a unique pattern of rearrangement, ranging from a simple tandem duplication event (Extended Data Fig. 2a), to multiple gains and losses along the body of chr. 21 (Fig. 2c), to a highly complex amplicon involving multiple chromosomes (Fig. 2a). The nonrecurrent translocation partners differed, with chr. 19 involved in two cases (Fig. 2e and Extended Data Fig. 2c) and chr. 7 (Extended Data Fig. 2a), chr. 22 (Fig. 2c) and chr. 17 and chr. 12 (Fig. 2a) implicated for others. For all, chr. 21 formed a focus of rearrangement across the genome (Fig. 2b,d,f and Extended Data Fig. 2b,d). The median CN over the shared amplified region in chr. 21 was 6.5 (range 3.4–8.2). In all cases, the amplification event occurred on one allele only. Of the median 150 coding small nucleotide variants called (range 130–160), none was recurrent, and none affected the amplified region on chr. 21. We deployed ClusterSV, an SV clustering and classification pipeline (Methods), to identify and classify SVs as simple or complex (≥3 interconnected SVs)[27]. In four of five cases (Fig. 2a,c,e and Extended Data Fig. 2c), the chr. 21amp event was classed as complex (Supplementary Table 3). In the case classed as a simple amplification event (Extended Data Fig. 2a), this was demarcated by a fold-back inversion rearrangement in keeping with a breakage–fusion–bridge cycle.

Review of breakpoint features highlighted that these were frequently characterized by small (0–6-base pair (bp)) insertions, most consistent with non-homologous end joining (NHEJ) as the predominant mechanism of repair[1,28–30]. There was no evidence of templated insertions.

The presence of fold-back loops in the cases profiled is consistent with breakage–fusion–bridge cycles as the initiating event. One of the cases (Patient 1, Fig. 2a) demonstrated CN oscillations between one low (CN = 2) and one very high (CN ≥ 10) event, possibly representing the presence of chromothripsis-associated circular extrachromosomal DNA (ecDNA)[31]. We further investigated for the presence of ecDNA by applying Decoil, an ecDNA detection algorithm[32], to long-read sequencing data obtained from Patient 1 after enriching for circular DNA structures (Supplementary Information)[32–34]. This excluded the presence of ecDNA. Furthermore, DNA fluorescence in situ hybridization (FISH) analysis of a chr. 21amp sample from Patient 3 with a high-level CN gain (Fig. 2e) confirmed that the amplification event was intrachromosomal (Fig. 2g,h). Together, these analyses confirmed that multiple different genomic events of variable complexity, including chromothripsis, converge to cause amplification of a specific genomic region on chr. 21.

## TARGET-seq prioritizes gene targets amplified by chr. 21amp

To delineate clonal hierarchies, relationship and timing of the chr. 21amp event relative to m*JAK2* and m*TP53*, we leveraged a dataset of four chr. 21amp, m*JAK2* and m*TP53* patients with BP-MPN, who had undergone TARGET-seq analysis, a multiomic approach enabling genotype-informed analysis of CN status and transcriptome in single

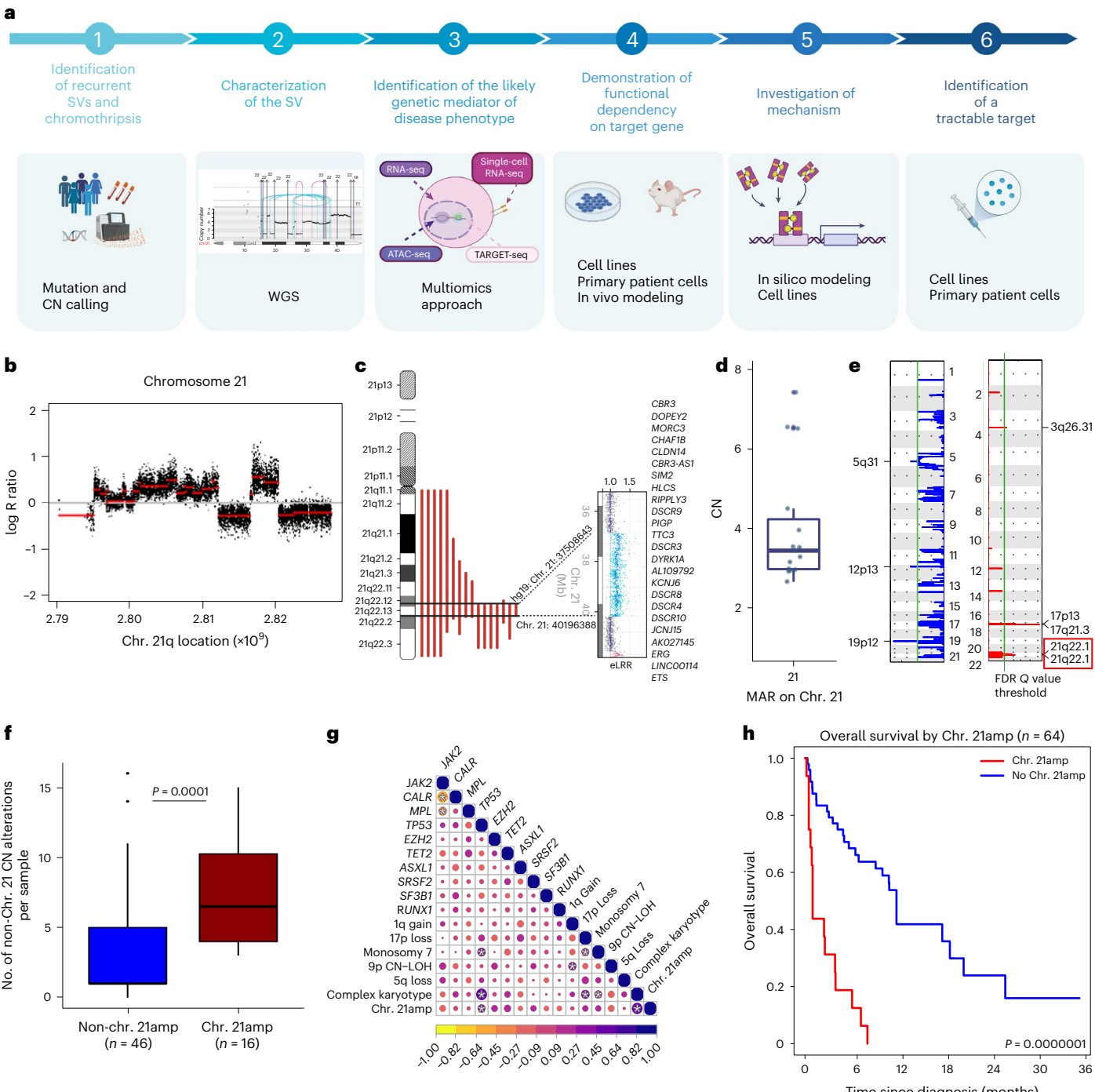

**Fig. 1 | Chromothripsis-associated chr. 21amp is a recurrent and adverse prognosis genome amplification event in BP-MPN. a**, Study overview. **b**, log R ratio plot of chromosome 21 derived from SNP karyotyping assay (DNACopy analysis) showing chromothripsis of chromosome 21 ('chr. 21amp') in a representative case of BP-MPN. SNP karyotyping performed for *n* = 64 samples. **c**, Graphic displaying the MAR in common across all chr. 21amp cases (*n* = 16). **d**, Boxplot of median/interquartile range (IQR) of CN overlying the chr. 21amp MAR for all cases (*n* = 16, the lower and upper hinge correspond to the IQR (25th and 75th percentiles), with the upper and lower whiskers extending from the hinge to ±1.5 × IQR). **e**, GISTIC analysis of recurrently lost (blue) and amplified

(red) focal regions across all cases. Green horizontal line depicts the false discovery rate (FDR)-adjusted *Q* value threshold of 0.05 (*n* = 64). **f**, Boxplot of median/IQR (as in **d**) showing that chr. 21amp cases have a greater number of non-chr. 21 CN abnormalities compared with non-chr. 21amp cases (median 6.5 (IQR 4–10.3) versus median 1 (IQR 1–5), *P* = 0.0001 by two-sided Wilcoxon rank-sum test). **g**, Heatmap shows Pearson correlation coefficient of myeloid mutations and most frequent CNAs. Purple denotes positive co-variance, yellow negative; *$P_{adj}$ < 0.05. **h**, Kaplan–Meier analysis of patients with BP-MPN stratified by presence/absence of chr. 21amp event. Schematic in **a** created using BioRender. com. eLRR, estimated log R ratio; Mb, megabase.

cells[19]. Genotyping and CN information was available for 1,903 of 2,205 cells (86.3%) (Fig. 3a), with 107 cells WT for chr. 21amp, *TP53* and/or *JAK2* mutation, 179 single *JAK2* mutant and 162 *JAK2/TP53* co-mutant, non-chr. 21amp cells. Chr. 21amp was highly clonal and

co-occurred with m*JAK2* and m*TP53* in 1,455 of 1,903 cells (76.5%), supporting that the chr. 21amp event occurs after m*JAK2* and m*TP53* (Fig. 3a).

The late timing of chr. 21amp acquisition was supported by analyzing somatic mutations occurring in the amplified region in WGS

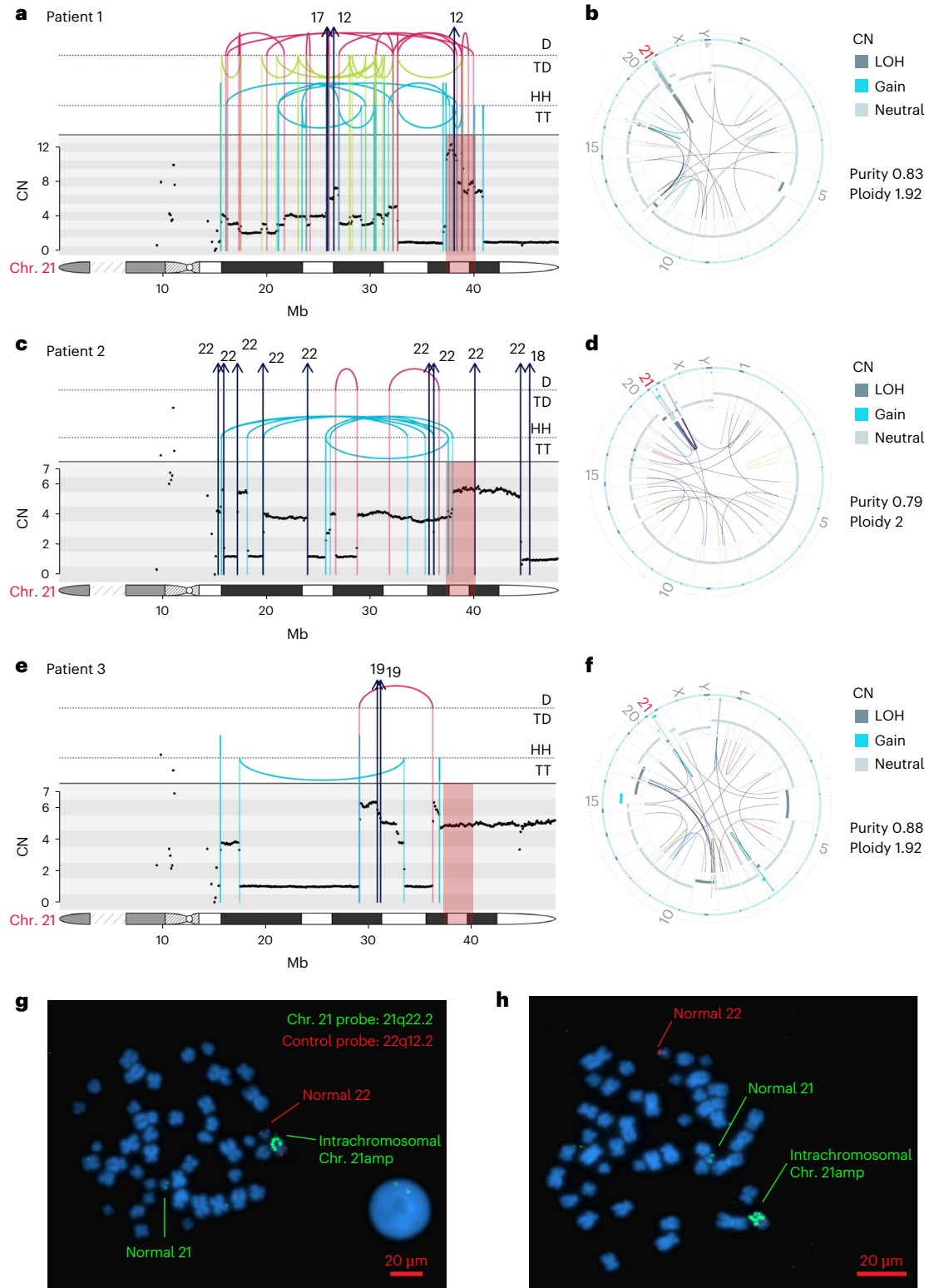

**Fig. 2 | WGS of chromothripsis-associated chr. 21amp events at high resolution. a,c,e,** Integrated CN and SV plots showing the complex SV in three chr. 21amp cases. The top panel shows intrachromosomal events as arcs between breakpoint loci, and color denotes the type of SV (black, translocation; red, deletion; blue, duplication; green, inversion). Rearrangements are further separated and annotated based on orientation. Interchromosomal events are shown with arrows denoting the likely partner chromosome. The middle panel shows the consensus CN across the chr. 21 ideogram, depicted in the lowest section of each plot to indicate breakpoint location. **b,d,f,** Circos plots showing global SV burden corresponding to the patients in **a, c** and **e,** demonstrating

clustering around chr. 21. The outer ring shows the chromosome ideogram. The middle ring shows the B allelic frequency and the inner ring shows the intra- and interchromosomal SVs with the same color scheme as in **a, c** and **e. g,h,** Two representative images of metaphase spreads and interphase cells from bone marrow cells from patient 3 (**e** and **f**) after FISH with two probes targeting the amplified region on chr. 21q22.2 (green) and a control region on chr. 22q12.2 (red). The chr. 21 amplification event is intrachromosomal. The experiment was performed once and 30 metaphase cells examined. Images were taken at ×1,000 magnification; scale bars, 20 μm. D, deletion; TD, tandem duplication; HH, head-to-head inverted; TT, tail-to-tail inverted.

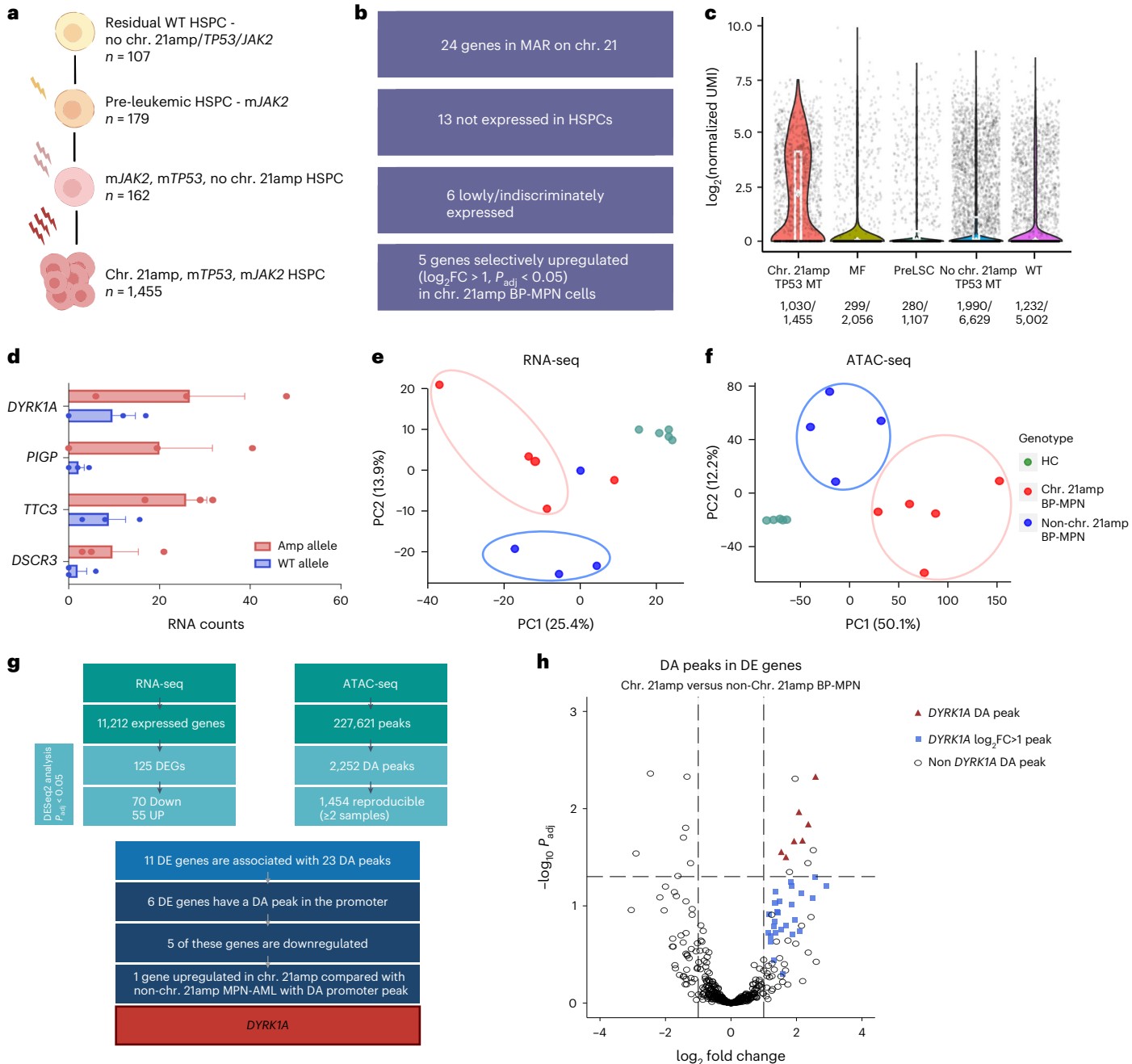

**Fig. 3 | Integrated RNA-seq and ATAC-seq pinpoint *DYRK1A* as the putative mediator of the adverse chr. 21amp phenotype. a**, TARGET-seq analysis of *n* = 1,903 cells from four chr. 21amp donors with allelic resolution of mutant *JAK2/TP53* and chr. 21amp event in single cells enables inference of clonal hierarchy. In total, 107 cells had no genomic aberration, while 179 cells were mutated for *JAK2V617F* alone. Further, 162 cells were double *JAK2* and *TP53* mutant, with no evidence of chr. 21amp, while 1,455 cells carried all three genomic aberrations. **b**, Analysis of TARGET-seq gene expression data from single HSPCs enables prioritization of 5 of the 24 genes in the chr. 21amp MAR. **c**, Violin plots showing that *DYRK1A* is overexpressed in chr. 21amp HSPCs compared with non-chr. 21amp control cells including myelofibrosis (MF, *n* = 2,056 cells from eight MF donors), pre-leukemic stem cells (preLSC, *n* = 1,107 nonmutant phenotypic HSCs, identified in 12 BP-MPN donors), *TP53*-mutant-non-chr. 21amp BP-MPN (no chr. 21amp m*TP53*, *n* = 6,629 cells from 14 BP-MPN donors) and WT cells (*n* = 5,002 from nine healthy donors). Each dot represents the expression value (log₂-normalized UMI count) for a single HSPC, with median and quartiles shown in

white. Expressing cell frequencies are shown on the bottom of each violin plot for each group. **d**, Bar plot (mean ± s.e.m.) demonstrating allele-specific expression of genes in the chr. 21amp MAR. All genes with informative heterozygous SNPs (*y* axis; SNP information in Supplementary Table 4) demonstrated allelic skew with a read bias towards the amplified allele (red) over the WT (blue). **e,f**, Principal component analysis of RNA-seq (**e**) and ATAC-seq (**f**) data shows clustering by chr. 21amp status. HC samples are depicted in green, chr. 21amp BP-MPN in red and non-chr. 21amp BP-MPN in blue. **g**, Integration of the RNA-seq and ATAC-seq datasets comparing chr. 21amp versus non-chr. 21amp BP-MPN Lin⁻CD34⁺ cells identifies 125 DE genes (DEGs) and 2,252 DA peaks (DESeq2 analysis, *P* values (adjusted for multiple comparisons) < 0.05). Only *DYRK1A* is DE with a DA promoter peak. **h**, Volcano plot of DA peaks in DE genes comparing chr. 21amp versus non-chr. 21amp BP-MPN samples (DESeq2 analysis), *y* axis scaled to log₁₀($P_{adj}$) ± 3 to highlight *DYRK1A* peaks. Of the 92 DA ATAC-seq peaks with log₂FC > 1, 33 occur in the *DYRK1A* gene body. HSC, hematopoietic stem cell; UMI, unique molecular identifier.

data using AmplificationTimeR[37]. Across all assessable samples, multiple chr. 21 gains occurred likely at the same time or in very rapid succession, in keeping with a single catastrophic chromothripsis event (Extended Data Fig. 2e,f). Gains encompassing the MAR on chr. 21amp were universally timed as late clonal events, occurring after all mutations within the gained region, suggesting that they occurred just before leukemic transformation (Extended Data Fig. 2e,f). These single-cell and WGS timing analyses support that chr. 21amp triggers leukemic evolution.

To prioritize candidate genes within the MAR, we compared expression in individual chr. 21amp human hematopoietic stem and progenitor cells (HSPCs) with non-chr. 21amp HSPCs, incorporating genotyping information for *TP53*. Of the 24 genes in this region, only five (*DYRK1A, DSCR3, MORC3, PIGP, TTC3*; Fig. 3b,c and Extended Data Fig. 3a–d) were significantly upregulated and differentially expressed (DE) in chr. 21amp single cells compared with controls. All five of these candidate genes were also upregulated in patients with gain of chr. 21q in de novo AML (TCGA; Extended Data Fig. 3e–i).

## Integrated RNA-seq and assay for transposase-accessible chromatin with sequencing implicates *DYRK1A* in chr. 21amp

To further characterize candidate genes in the amplified region, we performed mini-bulk RNA sequencing (RNA-seq) ($n = 200$ cells) and assay for transposase-accessible chromatin with sequencing (ATAC-seq) ($n = 1,000$ cells) on CD34+Lineage (Lin)− HSPCs in five chr. 21amp BP-MPN patients, four non-chr. 21amp patients and five age-matched healthy controls (HCs). All candidate genes with informative heterozygous SNPs showed a clear read bias in the RNA-seq dataset towards the amplified allele (Fig. 3d and Supplementary Table 4). Unsupervised principal component analysis using highly variable genes and peaks in both the RNA-seq (Fig. 3e) and ATAC-seq (Fig. 3f) datasets demonstrated that chr. 21amp status accounted for a high percentage of variation and cell identity.

There were 125 DE genes, of which 55 were upregulated in chr. 21amp versus non-chr. 21amp Lin−CD34+ HSPCs. The only gene from the MAR that was upregulated in chr. 21amp cells compared with non-chr. 21amp BP-MPN cells was *DYRK1A* ($P = 0.0005$, adjusted $P$ value ($P_{adj}$) = 0.03) (Extended Data Fig. 3j and Supplementary Tables 5 and 6). Integrated analysis of differentially accessible (DA) and DE genes comparing chr. 21amp versus non-chr. 21amp Lin−CD34+ cells identified 11 DE genes associated with 23 DA peaks (Fig. 3g). Only *DYRK1A* was DE with a DA promoter peak (log$_2$ fold change (log$_2$FC) 2.36, $P_{adj}$ 0.015)—along with six further DA peaks along the gene body ($P_{adj} < 0.05$) and a further 26 peaks with log$_2$FC > 1 (Fig. 3h and Extended Data Fig. 3k).

## Impact of *DYRK1A* overexpression in AML cohorts

Overexpression of *DYRK1A* in the Beat AML cohort was associated with adverse overall survival even in the absence of chr. 21amp (HR 1.44, 95% CI 1.07–1.93, $P$ value 0.03; Extended Data Fig. 4a), which was not the case for other genes in the chr. 21amp amplified region (Extended Data Fig. 4b,c). Patients with AML in the top versus bottom quintile of *DYRK1A* expression (Extended Data Fig. 4d) showed distinct gene expression (Extended Data Fig. 4e), including enrichment for multiple signaling pathways (JAK–STAT, TNF, TGFβ) and downregulation of DNA repair pathways (Extended Data Fig. 4f and Supplementary Table 7).

## Chr. 21amp influences cell state and transcriptional landscape

Next, we explored the impact of chr. 21amp on the transcriptional and cellular landscape in BP-MPN. Geneset enrichment analysis (GSEA) comparing DE genes in the chr. 21amp versus non-chr. 21amp Lin−CD34+ HSPC RNA-seq data revealed JAK–STAT signaling-associated genes

among the top upregulated pathways, with downregulated pathways including those regulating cell division and survival (Fig. 4a and Supplementary Table 8). GSEA between chr. 21amp and HC HSPCs similarly demonstrated upregulation of JAK–STAT signaling pathway gene expression, with downregulation of G2M checkpoint and DNA repair pathways (Fig. 4b and Supplementary Table 9).

To investigate the effect of chr. 21amp on cell differentiation, we performed droplet-based, high-throughput single-cell RNA-seq on Lin−CD34+ cells and total mononuclear cells (MNCs) for two chr. 21amp patients, eight non-chr. 21amp patients and five HC bone marrows. The chr. 21amp event was readily identified in individual cells (Fig. 4c) and highly clonal. Projection of the cells from BP-MPN patients onto the HC reference (Fig. 4d,e) showed that chr. 21amp cells are present from the apex of the hematopoietic differentiation hierarchy, with chr. 21amp HSPCs particularly expanded at the multipotent progenitor (MPP)-precursor stage (Fig. 4e,f). Chr. 21amp cells were notably less frequent in late erythroid precursors, implying presence of a differentiation block, with leukemic cells carrying the chr. 21amp event frequently stalled in a progenitor state (Fig. 4f,g). Chr. 21amp HSCs, MPPs, granulocyte-monocyte progenitors and megakaryocyte-erythroid progenitors showed significantly elevated *DYRK1A* expression relative to non-chr. 21amp cells (Fig. 4h). GSEA comparing chr. 21amp and *DYRK1A*-upregulated HSCs with non-chr. 21amp HSCs again demonstrated upregulation of multiple signaling pathways, including MYC, Notch and PI3 kinase signaling, with downregulation of apoptosis and *TP53* pathways (Extended Data Fig. 4g and Supplementary Table 10).

Single-cell regulatory network inference and clustering (SCENIC) analysis by cell type demonstrated that the chr. 21amp event had a global effect on shaping active gene regulatory networks compared with non-chr. 21amp and HC HSPCs (Extended Data Fig. 4h,i)[38,39]. Key transcription factor networks including signaling pathways such as STAT5A and STAT5B, along with negative regulators of apoptosis such as SOX4 (ref. 40), were globally upregulated in chr. 21amp HSPCs compared with both non-chr. 21amp BP-MPN and HCs (Extended Data Fig. 4h,i). Conversely, and in keeping with the GSEA analyses, downregulation of the TP53 transcription factor network was observed (Extended Data Fig. 4h,i). Collectively, these findings identify *DYRK1A* as the lead candidate gene for further independent validation, functional and mechanistic studies.

## *DYRK1A* expression and dependency in cell line models

We next sought to functionally validate *DYRK1A* as a gene conferring a cell survival advantage in the BP-MPN context. In silico screening of Broad's Cancer Dependency Map (DepMap) showed that cancer cell line dependency scores were linked to *DYRK1A* gene expression ($P < 0.0001$ by linear regression; Extended Data Fig. 4j). Myeloid cell lines were among the highest expressors of *DYRK1A* (Extended Data Fig. 4k) and demonstrated the highest gene dependency (Extended Data Fig. 4l). Conversely, myeloid leukemia cell lines with low *DYRK1A* expression did not show dependency on DYRK1A, for example, K562 (Extended Data Fig. 4j)[35].

A kinase domain-focused CRISPR screen previously highlighted that two *JAK2* mutant BP-MPN cell lines (human erytholeukemia (HEL) and the megakaryoblastic leukemia line (SET2)) are hypersensitive to *DYRK1A* targeting compared with other AML cell lines[35]. Both SET2 and HEL have a high CN over the *DYRK1A* locus relative to other cell lines in the Cancer Cell Line Encyclopedia (3.28 versus 1.83, respectively)[36]. HEL cells harbor a duplication of chr. 21q21.1-term (ref. 41) and are a clear outlier among AML cell lines, both highly expressing *DYRK1A* and highly dependent on *DYRK1A* (CRISPR dependency score −0.72, DepMap screening tool; Extended Data Fig. 4j)[42–45]. Taken together, these data support that the BP-MPN cell lines HEL and SET2 are relevant models to study *DYRK1A*'s functional role.

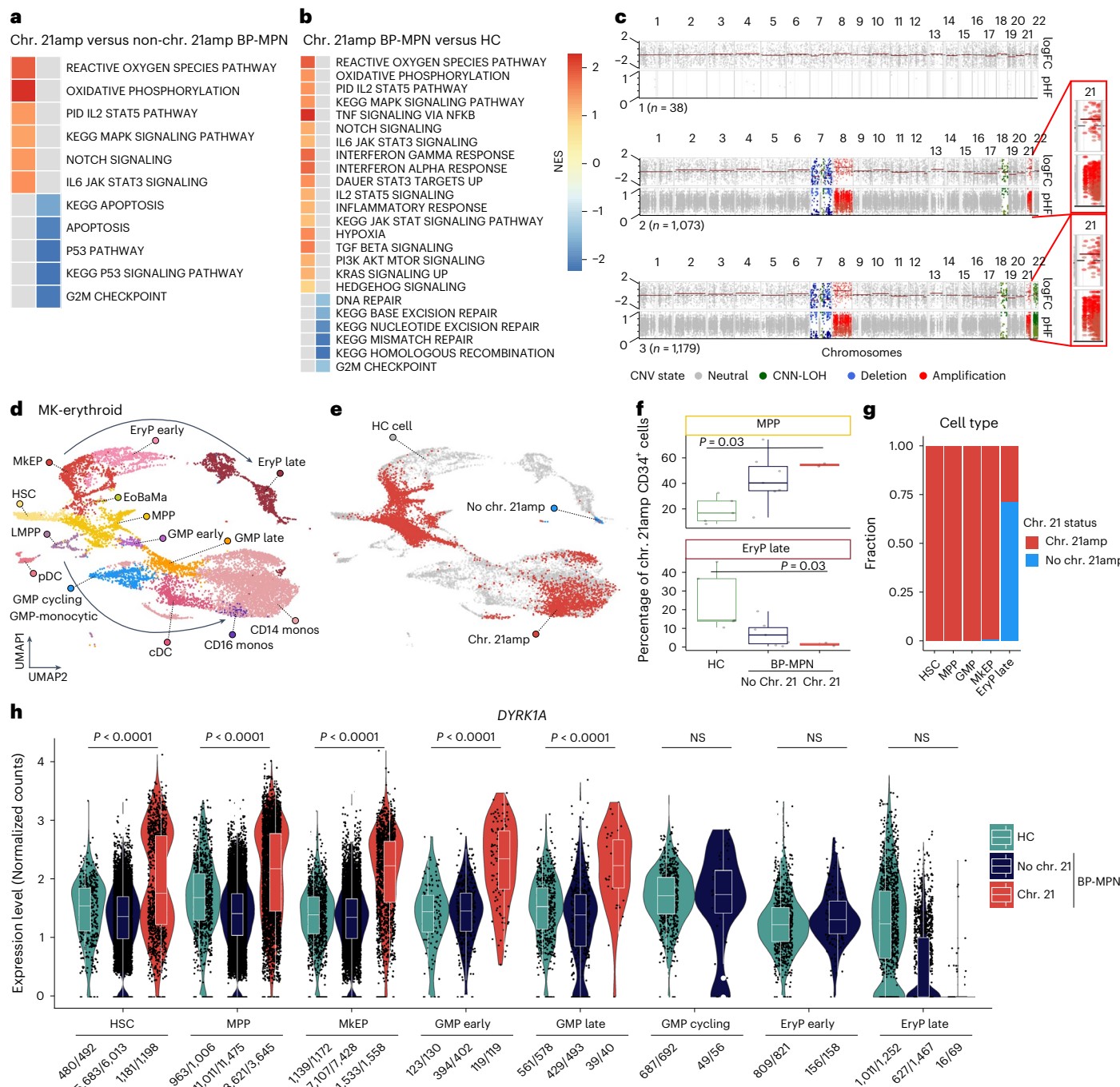

**Fig. 4 | Investigating the chr. 21amp-associated cell state and transcriptional landscape. a,b,** GSEA for selected KEGG and HALLMARK pathways with NES > 1 shown in the heatmap for chr. 21amp (*n* = 5) versus non-chr. 21amp BP-MPN (*n* = 4) (**a**) and chr. 21amp BP-MPN (*n* = 5) versus HC (*n* = 5) (**b**) RNA-seq datasets (Supplementary Tables 8 and 9). **c,** Clone-specific pseudobulk profile for a representative patient showing detection of the chr. 21amp event in single cells by the CN-calling software numbat. Each of the three plot subpanels defines a CN-defined clone, with the chromosomal location along the *x* axis. Each subpanel contains two sections; the top section shows the log₂FC of normalized CN and the bottom panel the parental haplotype frequency (pHF), inferred from haplotype phasing of SNPs genotyped from single-cell transcriptomes. CNA calls are colored by type of alteration (amplification in red, deletion in blue, CNN-LOH in green). The red magnified box highlights the chr. 21amp event. **d,** UMAP representation of a healthy donor hematopoietic hierarchy of *n* = 6,143 HSPCs and myeloid cells. **e,** UMAP projection of *n* = 6,572 cells from two chr. 21amp BP-MPN donors onto the healthy donor hematopoietic atlas colored by chr. 21amp status (chr. 21amp cell, red; non-chr. 21amp cell, blue; HC, gray). **f,** Box-and-whisker plots of the

percentage of CD34⁺ cells called as MPP and EryP based on projection analysis in **e**, showing expansion of MPP and depletion of EryP compared with HCs (plot shows median ± IQR with the whiskers extending ±1.5 × IQR; significance testing by paired Wilcoxon rank-sum test, *n* = 2 chr. 21amp BP-MPN, *n* = 8 non-chr. 21amp BP-MPN, *n* = 5 HCs). **g,** Barchart depicting the fraction of cells called as chr. 21amp from two chr. 21amp donors, demonstrating the differentiation block into erythroid cells. **h,** Violin plots of *DYRK1A* overexpression in chr. 21amp HSPC progenitors compared with non-chr. 21amp BP-MPN and HC cells. Each dot represents the expression value (log₂-normalized UMI count) for each single cell; box-and-whiskers plot as in **f**. Expressing cell frequencies are shown at the bottom of each violin plot. *P* values by Wilcoxon rank-sum test (*n* = 6,143 HSPCs from HCs, *n* = 27,492 non-chr. 21amp BP-MPN and *n* = 6,572 chr. 21amp BP-MPN cells, same donors as in **f**). UMAP, Uniform Manifold Approximation and Projection; MkEP, megakaryocyte-erythroid progenitors; EryP, erythroid progenitors; EoBaMa, eosinophil-basophil-mast progenitors; LMPP, lymphoid-primed MPP; GMP, granulocyte-monocyte progenitors; cDC, classical dendritic cell; pDC, plasmacytoid dendritic cell; monos, monocytes; NS, not significant.

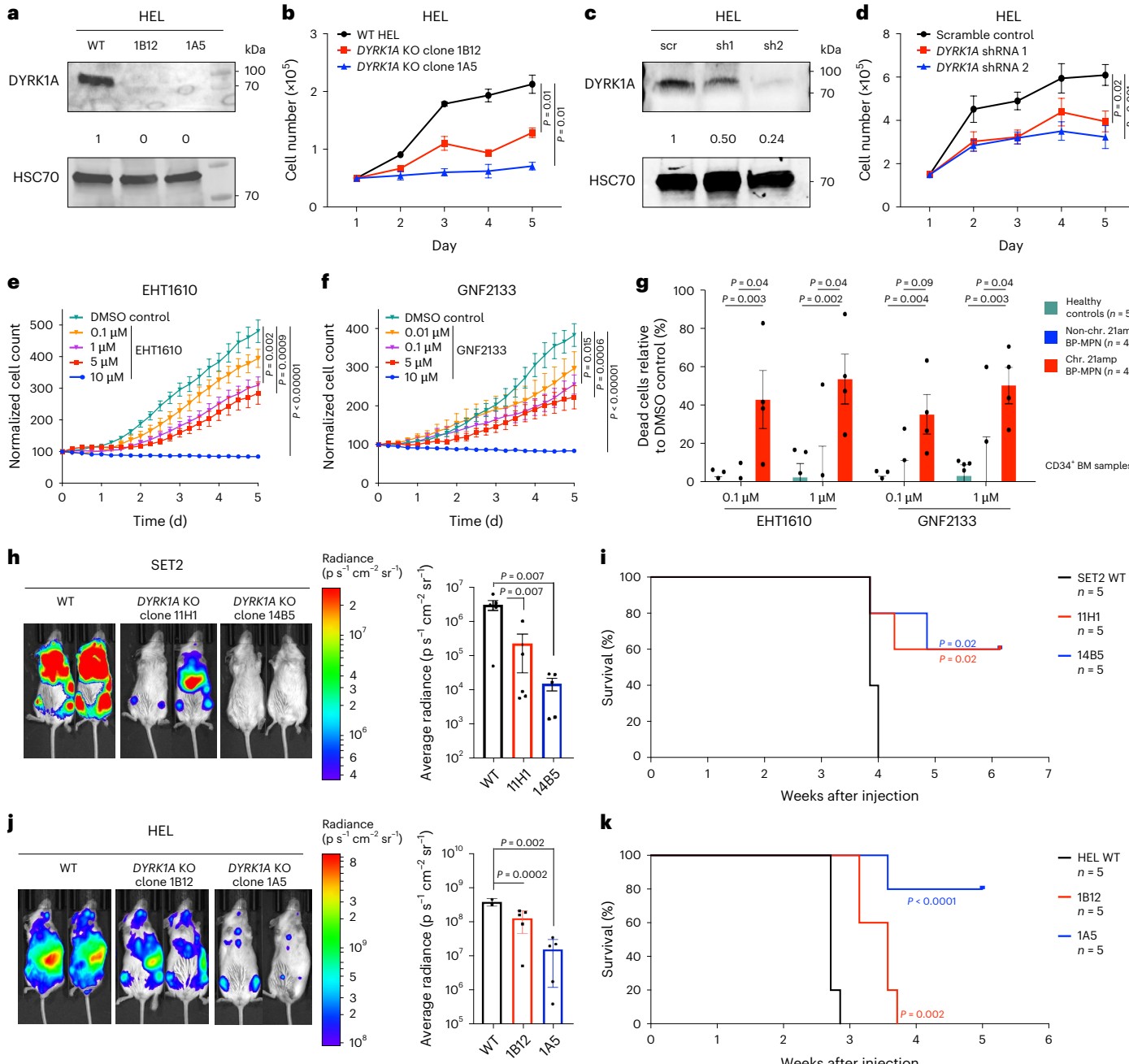

**Fig. 5 | _DYRK1A_ promotes cell proliferation and survival in chr. 21amp BP-MPN. a**, Western blot showing reduced DYRK1A expression in _DYRK1A_ KO HEL cells. Densitometric values were normalized to HSC70 (representative of $n = 3$ experiments). **b**, Cell counts for cultured HEL WT and two _DYRK1A_ KO clones (1B12 and 1A5) (mean ± s.e.m., $n = 3$ independent experiments in triplicate per condition, significance calculated by two-way ANOVA). **c**, Western blot showing the knockdown of DYRK1A expression in HEL cells with target-specific shRNA or scramble control. Densitometric values were normalized to HSC70 (representative of $n = 3$ experiments). **d**, Cell counts for transduced HEL cells in culture (mean ± s.e.m., $n = 3$ independent experiments in triplicate per condition, significance calculated by two-way ANOVA). **e,f**, Dose-dependent reduction of HEL cell proliferation in culture with the DYRK1A inhibitor EHT1610 (**e**) or GNF2133 (**f**) ($n = 6$ replicates, significance calculated by two-way ANOVA with Bonferroni's post-test, mean ± s.e.m.). **g**, Primary patient chr. 21amp BP-MPN ($n = 4$) versus HC ($n = 5$) CD34$^+$ and non-chr. 21amp BP-MPN ($n = 4$) cell

viability at day 5 after treatment with 0.1 µM or 1 µM GNF2133 or 0.1 µM or 1 µM EHT1610. Boxplot shows mean ± s.e.m. Groups were compared by multiple _t_-tests with the Benjamini–Hochberg procedure applied to control the FDR. The FDR threshold was set at $Q < 0.05$. **h**–**k**, Impact of _DYRK1A_ KO on BP-MPN cells in vivo. **h,j**, Bioluminescent images of representative mice following transplantation of $3 \times 10^6$ luciferase-expressing WT SET2 versus _DYRK1A_ KO clones 11H1 and 14B5 at 2 weeks (**h**) or WT HEL versus _DYRK1A_ KO clones 1B12 and 1A5 cells at 3 weeks (**j**) ($n = 5$ each). In both **h** and **j** the intensity of luminescence is normalized and shown as average radiance (p s$^{-1}$ cm$^{-2}$ sr$^{-1}$); boxplots show mean ± s.e.m., significance calculated by ANOVA and $P_{adj}$ values given. **i,k**, Kaplan–Meier survival curves of mice ($n = 5$ each) injected with luciferase-expressing WT SET2, _DYRK1A_ KO clone 11H1 or 14B5 cells (**i**) or WT HEL, _DYRK1A_ KO clone 1B12 or 1A5 cells (**k**) (significance calculated by one-sided Mantel–Cox log-rank test).

## *DYRK1A* promotes proliferation and survival in chr. 21amp BP-MPN

We tested the impact of *DYRK1A* knockout (KO) and knockdown using CRISPR and short hairpin RNA (shRNA) approaches in HEL and SET2 BP-MPN cell lines (Fig. 5a–d and Extended Data Fig. 5a–f). *DYRK1A* KO/knockdown was confirmed by western blot (Fig. 5a,c and Extended Data Fig. 5b,d). *DYRK1A* targeting by both CRISPR KO (Fig. 5b and Extended Data Fig. 5a) and shRNA knockdown (Fig. 5d and Extended Data Fig. 5c,e) significantly slowed proliferation of HEL and SET2 cells. We next explored whether pharmacological inhibition of DYRK1A using the small-molecule inhibitors GNF2133 and EHT1610 would have the same impact. We first confirmed that phosphorylation of the known DYRK1A substrates LIN52 and FOXO1 (refs. 46,47) was reduced following EHT1610 treatment of SET2 cells (Extended Data Fig. 5g). We then showed that pharmacologic inhibition of DYRK1A led to a dose-dependent reduction in HEL cell proliferation (Fig. 5e,f).

We next tested the impact of DYRK1A inhibition in CD34+ HSPCs cells from patients with BP-MPN. HSPCs from patients with chr. 21amp BP-MPN were treated with GNF2133 and EHT1610 at 0.1 μM and 1 μM doses in comparison with HC and non-chr. 21amp BP-MPN samples (Extended Data Fig. 5h). By day 5, there was a substantial and selective reduction in the viability of chr. 21amp BP-MPN cells with DYRK1A inhibition, while cells from non-chr. 21amp BP-MPN cases as well as HCs were unaffected (mean 46% cells viable in chr. 21amp versus 99% in non-chr. 21amp and 97% in HCs, $Q < 0.05$ for EHT1610 1 μM; Fig. 5g and Extended Data Fig. 5i).

To investigate the effect of *DYRK1A* on the leukemia-propagating capacity of *JAK2* mutant BP-MPN cell lines in vivo, we performed CRISPR-mediated *DYRK1A* KO in luciferase-tagged SET2 and HEL cell lines and compared their leukemogenic capacity in xenografts using immunodeficient mice (Fig. 5h–k). The intensity of luminescence was significantly reduced in the CRISPR KO context in both cell lines (Fig. 5h,j). This was associated with a significant survival advantage (median survival 3.9 weeks for SET2 WT versus not reached for KO clones 14B5 and 11H1, $P = 0.02$ (Mantel–Cox log-rank test), median survival post-injection 2.7 weeks for HEL WT versus 3.6 weeks for KO clone 1B12 versus not reached for KO clone 1A5, $P < 0.001$ (Mantel–Cox log-rank test); Fig. 5i,k).

Overall, these data validate DYRK1A as the key driver of leukemic progression in a significant proportion of patients with BP-MPN and confirm that chr. 21amp confers a selective vulnerability to DYRK1A inhibition.

## *DYRK1A* regulates the DREAM complex and DNA repair

Chr. 21amp BP-MPN is associated with genetic instability (Fig. 1f) and, as a quarter of chr. 21amp cases were WT for *TP53*, we reasoned that the chr. 21amp event itself might perturb DNA repair and/or cell survival pathways.

DYRK1A-dependent phosphorylation of LIN52 is a requisite initiating step in the assembly of the DREAM complex, a key repressor

of DNA repair implicated in oncogenesis and chemoresistance in solid tumors[46,48–50]. We hypothesized that DYRK1A overexpression in BP-MPN may activate the DREAM complex, repress DREAM target genes and thereby promote genomic instability (Extended Data Fig. 6a). In support of this, in primary patient chr. 21amp BP-MPN cells versus controls, as well as in Beat AML top *DYRK1A* expressors versus bottom, the DREAM DNA repair geneset was downregulated (Fig. 6a–d; normalized enrichment score (NES) −1.74, family-wise error rate (FWER) $P$ value 0.01 for chr. 21amp versus non, NES −2.13, FWER $P < 0.001$ for Beat AML top DYRK1A expressors versus bottom). Conversely, *DYRK1A* CRISPR KO SET2 cells showed significant upregulation of DREAM complex target genes (Fig. 6e,f; NES 1.76, FWER $P < 0.001$).

To assess whether loss of *DYRK1A* in BP-MPN might restore DNA repair pathways, we induced DNA damage in WT and *DYRK1A* CRISPR KO SET2 cells by treatment with etoposide. *DYRK1A* KO cells showing greater resistance to etoposide, suggesting reduced DNA damage induced apoptosis (Fig. 6g; $P < 0.001$). *DYRK1A* KO also reduced double-stranded DNA breaks as ascertained by γ-H2AX staining after 8-h treatment with 3 μM etoposide in *DYRK1A* CRISPR KO SET2 cells (Fig. 6h and Extended Data Fig. 6b). Consistent with this, induction of DNA damage by irradiation in *DYRK1A* KO versus WT SET2 cells led to fewer detectable double-stranded DNA breaks at 8 h in KO than WT (Fig. 6i), which we infer may be due to enhanced kinetics of repair.

Taken together, these data support that chr. 21amp-induced *DYRK1A* overexpression leads to suppression of DNA repair through aberrant DREAM complex activity, in keeping with the increased number of CNAs we observed in chr. 21amp BP-MPN cases (Fig. 1f).

## *DYRK1A* activates JAK–STAT, driving upregulation of *BCL2*

We consistently observed transcriptional upregulation of the JAK–STAT signaling axis across single-cell and bulk datasets in association with chr. 21amp and *DYRK1A* overexpression (Fig. 4a,b and Extended Data Fig. 4f,h,i). To further explore this, we analyzed gene expression data from SET2 cells with and without *DYRK1A* KO. SET2 cells showed downregulation of STAT5 target genes after *DYRK1A* CRISPR KO (NES −2.08, FWER $P$ value 0.001; Fig. 7a and Extended Data Fig. 7a). Furthermore, chr. 21amp BP-MPN cases showed enrichment of STAT3 (Fig. 7b) and STAT5 (Fig. 7c) genesets in comparison with HCs. Consequently, we sought to investigate whether *DYRK1A* might drive disease progression in MPN by amplifying JAK–STAT signaling.

DYRK1A and JAK2 have both been shown to activate STAT3 at residue Tyr705 (refs. 47,51–54). In line with previous observations, STAT3-Tyr705 phosphorylation occurred in both HEL and SET2 BP-MPN cell lines, and DYRK1A inhibition led to a dose-dependent reduction in STAT3-Tyr705 phosphorylation. (Extended Data Fig. 7b). We next assessed the effect of *DYRK1A* overexpression on JAK–STAT pathway activation, using a STAT5 luciferase reporter assay system[55] in human embryonic kidney 293T (HEK293T) cells with co-expression of either WT *Jak2* or *Jak2V617F* (see the Supplementary Methods for

**Fig. 6 | Amplified *DYRK1A* perturbs DNA damage response via regulation of the DREAM complex. a**, Volcano plot of expressed genes comparing chr. 21amp (*DYRK1A* upregulated $n = 5$) versus non-chr. 21amp ($n = 5$) CD34+ HSPCs, highlighting DREAM complex target genes in red. The transcriptional signature of DREAM complex DNA repair genes is downregulated. **b**, GSEA analysis showing downregulation of the DREAM complex DNA repair geneset in chr. 21amp BPMPN HSPCs (NES −1.74, FWER $P$ value 0.01). **c**, Volcano plot of expressed genes comparing Beat AML top quintile *DYRK1A* expressors ($n = 72$) versus bottom quintile *DYRK1A* expressors ($n = 72$); the transcriptional signature of DREAM complex DNA repair genes is downregulated. **d**, GSEA analysis showing downregulation of the DREAM complex DNA repair geneset in BEATAML high *DYRK1A*c expressors (NES −2.13, FWER $P$ value ≤ 0.01). **e,f**, The transcriptional signature of DREAM complex genes involved in DNA repair is upregulated after CRISPR KO. **e**, Volcano plot of DE genes

comparing CRISPR KO ($n = 5$, 2 clones) versus WT ($n = 3$), highlighting target genes of the DREAM DNA repair complex. **f**, GSEA demonstrating significant enrichment for DREAM DNA repair complex genes (NES 1.76, FWER $P$ value 0.008). **a,c,e**, Genes DE by DESeq2 analysis after adjustment for multiple comparisons. **g**, Proliferation of CRISPR KO versus WT SET2 clones assessed by CellTitreGlo proliferation assay after 48-h treatment with indicated concentration of etoposide. Half-maximum inhibitory concentration (IC$_{50}$) = 8.4 μM for *DYRK1A* KO and 3.3 μM for SET2 WT clones ($n = 2$ independent replicates). **h**, Percentage of cells staining positive for γH2AX by flow cytometry at 8 h post 3 μM etoposide treatment by cell type ($n = 3$ replicates per condition; comparison by ANOVA adjusted for multisample testing). **i**, Percentage of cells staining positive for γH2AX on flow cytometry at 2 h post 200-rad irradiation treatment by cell type ($n = 3$ replicates per condition; comparison by ANOVA adjusted for multisample testing).

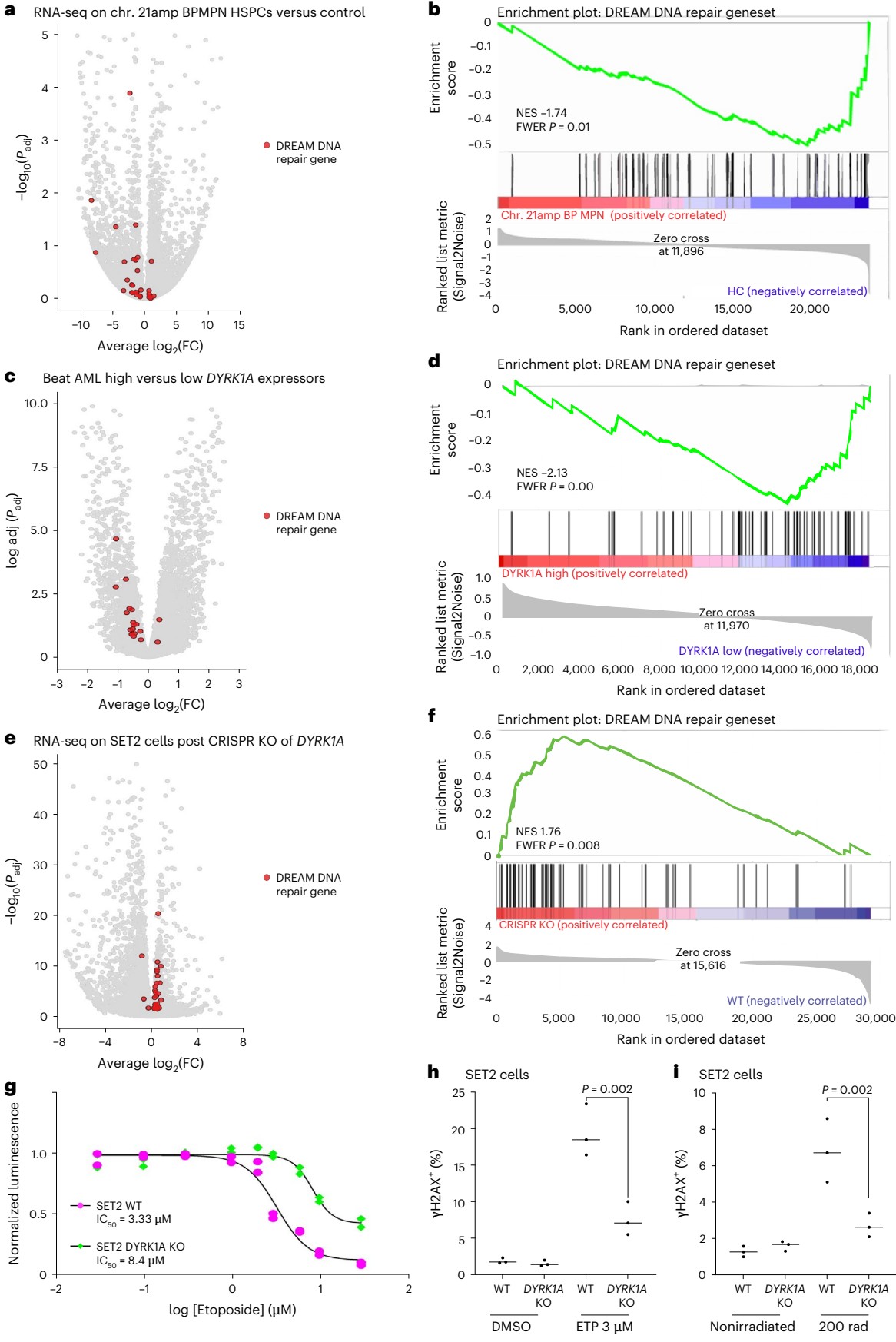

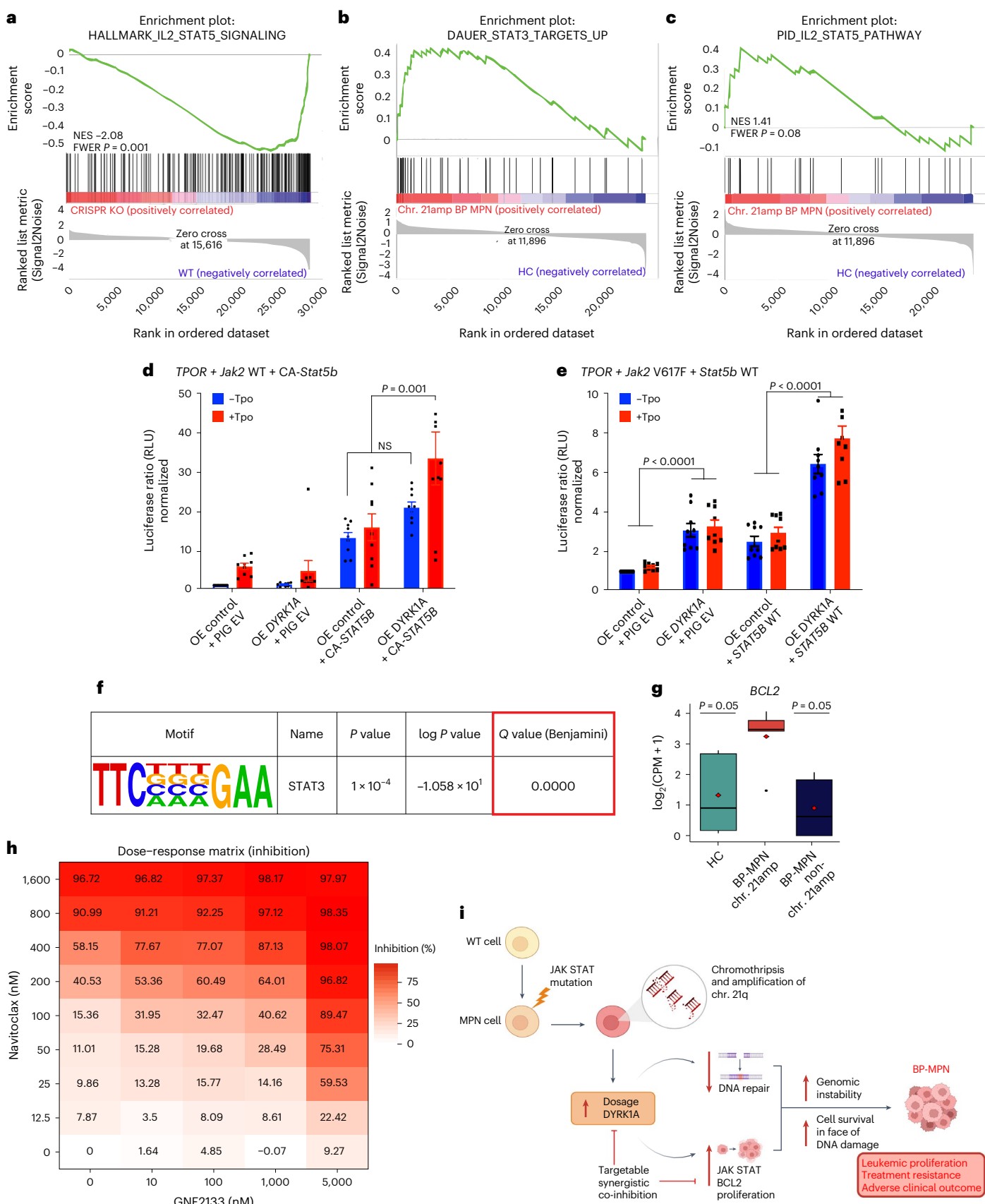

**a** Enrichment plot: HALLMARK_IL2_STAT5_SIGNALING

NES −2.08
FWER *P* = 0.001

CRISPR KO (positively correlated)
Zero cross at 15,616
WT (negatively correlated)

**b** Enrichment plot: DAUER_STAT3_TARGETS_UP

Chr. 21amp BP MPN (positively correlated)
Zero cross at 11,896
HC (negatively correlated)

**c** Enrichment plot: PID_IL2_STAT5_PATHWAY

NES 1.41
FWER *P* = 0.08

Chr. 21amp BP MPN (positively correlated)
Zero cross at 11,896
HC (negatively correlated)

**d** *TPOR + Jak2* WT + CA-*Stat5b*

*P* = 0.001
NS

**e** *TPOR + Jak2* V617F + *Stat5b* WT

*P* < 0.0001
*P* < 0.0001

**f**

| Motif | Name | *P* value | log *P* value | *Q* value (Benjamini) |
|---|---|---|---|---|
| TTCₛₛₛGAA | STAT3 | $1 \times 10^{-4}$ | $-1.058 \times 10^{1}$ | 0.0000 |

**g** *BCL2*

*P* = 0.05   *P* = 0.05

HC | BP-MPN chr. 21amp | BP-MPN non-chr. 21amp

**h** Dose−response matrix (inhibition)

**i**

further details). As expected in this system, Stat5b transcriptional activity was activated in the presence of constitutively active *Stat5b* (CA-*Stat5b*) in the absence of TPO (Fig. 7d). When co-expressed with CA-*Stat5b* and WT *Jak2*, *DYRK1A* overexpression led to increased Stat5b

transcriptional activity following TPO stimulation (Fig. 7d). When *DYRK1A* was overexpressed in the context of WT *Stat5b* and *Jak2V617F*, *DYRK1A* overexpression increased STAT5B transcriptional activity independent of TPO stimulation (Fig. 7e). These data support that an

**Fig. 7 | DYRK1A upregulation is associated with activation and amplification of the JAK–STAT signaling axis and consequent upregulation of BCL2.**
**a–c**, GSEA analyses showing that the HALLMARK IL2 STAT5 geneset is downregulated in CRISPR KO ($n = 5$, 2 clones) versus WT ($n = 3$) control SET2 cells (**a**), and that STAT3 (**b**) and STAT5 (**c**) genesets are upregulated in chr. 21amp BP-MPN versus control cells (RNA-seq, $n = 5$ cases per condition, GSEA analysis (Broad Institute, adjusted for multiple comparisons)). **d,e**, Luciferase reporter assays for STAT5 transcriptional activity in the context of *DYRK1A* WT overexpression versus control. **d**, HEK293T cells were transfected with WT human-*TPOR*, murine-Jak2 WT and either empty vector (PIG EV) or CA-*Stat5b* and an overexpression control (OE control) or a *DYRK1A* WT overexpression vector (OE *DYRK1A*). At 24 h post transfection, *Stat5b*-dependent transcriptional activity with (red) or without (blue) TPO treatment at 6 h was measured by the firefly luciferase assay system with Spi-Luc reporter (STAT5 response elements) as an internal control. **e**, A WT *Stat5b* vector rather than the constitutively active form was transfected alongside *Jak2V617F* rather than WT *Jak2*. For **d** and **e**, the boxplots show mean ± s.e.m. of three independent experiments in triplicate. Significance was assessed using Tukey's multiple comparison test. **f**, HOMER motif discovery analysis searching for the palindromic core STAT binding motif, demonstrating significant enrichment in chr. 21amp peaks versus background (cumulative hypergeometric distribution, adjusted for multiple comparisons). **g**, *BCL2* gene expression is upregulated in chr. 21amp BP-MPN versus controls, assessed by RNA-seq (paired Wilcoxon rank-sum test; the box-and-whiskers plots show the median and the IQR, with the whiskers extending ±1.5 × IQR; the mean is shown as a diamond). **h**, Synergy matrix scores between GNF2133 (0–5 µM) and navitoclax (0–1.6 µM) treatment of HEL cells. Results represent mean percentage viability assessed by annexin V/propidium iodide staining by flow cytometry, normalized to DMSO-treated control wells for six replicates per condition. **i**, Schematic of proposed model of chr. 21amp driving BP-MPN transformation.

important effect of *DYRK1A* overexpression is to activate STAT5, further amplifying activation of JAK–STAT signaling which is a cardinal feature of MPN in chronic phase.

We then looked for evidence of transcriptional activation and STAT binding in ATAC-seq data generated from chr. 21amp primary patient cells. The palindromic core motif in sequences recognized by all STATs is well-described (TTCN3GAA) and was significantly enriched in chr. 21amp DA peaks compared with background controls ($P_{adj} < 0.001$; Fig. 7f).

A key STAT3 target is the pro-survival oncogene B-cell lymphoma 2 (*BCL2*). Consistent with a functional link between *DYRK1A* and STAT transcriptional regulation, *BCL2* was one of the top ten co-dependencies with *DYRK1A* in the DepMap database (Extended Data Fig. 7c)[42–44,56,57]. In the *DYRK1A* CRISPR KO SET2 clones, *BCL2* was downregulated compared with control cells (log₂FC −0.76, $P_{adj} < 0.001$ on DeSeq2 analysis; Extended Data Fig. 7d). Furthermore, in chr. 21amp versus non-chr. 21amp primary patient BP-MPN cells, we observed upregulation of *BCL2* RNA expression (Fig. 7g) and chromatin accessibility (Extended Data Fig. 7e).

The synchronized upregulation of both *BCL2* and *DYRK1A* in chr. 21amp cells provided a strong rationale to look for therapeutic synergy between DYRK1A and BCL2 targeting. Co-inhibition of HEL cells with the DYRK1A inhibitor GNF2133 and the BCL2 inhibitor navitoclax demonstrated evidence of substantial therapeutic synergy (Bliss synergy score 15.02 (ref. 58); Fig. 7h and Extended Data Fig. 7f,g).

Collectively, these data support that *DYRK1A* overexpression in the context of basal JAK–STAT activation leads to further activation and potentiation of JAK–STAT signaling, driving oncogenicity and cell survival in part by the upregulation of *BCL2*. BCL2 can be therapeutically targeted with an inhibitor licensed for current clinical use, with synergy between BCL2 and DYRK1A inhibition.

## Discussion

Here we describe a frequent intrachromosomal amplification event affecting chromosome 21 in BP-MPN, uncovering a potentially actionable therapeutic vulnerability. Chr. 21amp leads to overexpression of *DYRK1A* which orchestrates perturbation of DNA repair, exacerbated JAK–STAT signaling and pro-survival pathways (Fig. 7i). Chr. 21amp occurs through several mechanisms, which include simple CN gains, breakage–fusion–bridge cycles and chromothripsis. For an additional discussion relating to chr. 21amp in other disease contexts, and the pathobiological impact of *DYRK1A* overexpression, please see the Supplementary Note.

It is increasingly acknowledged that CNAs are a major contributor to cancer evolution, and that patterns of aneuploidy events are nonrandom and tissue-specific[31,56,57,59,60]. Recent longitudinal data in patients with Fanconi anemia also provide an example of how a CNA can drive leukemic transformation[61]. In m*TP53* BP-MPN, convergent clonal evolution occurs, with loss of both *TP53* WT alleles acting in concert with the gain of CNAs[19]. While certain CNAs are recognized

as predictors of adverse outcome[9], this analysis provides a detailed analysis of how a specific event mechanistically supports leukemic transformation.

As *DYRK1A* overexpression is orchestrating multiple cellular processes to promote disease progression in MPN (Supplementary Note), it is interesting to speculate which component (amplified JAK–STAT signaling versus increased genomic instability) is dominant. In our view, the strong synergy between presence of p53 mutation and chr. 21amp, together with the striking increase in non-chr. 21amp CNAs in cases with *DYRK1A* amplification, support that the impact on DNA repair is critical. The lack of durable responses to JAK2 inhibition in BP-MPN also supports that inhibition of amplified JAK–STAT signaling alone is insufficient to ameliorate the disease[62,63]. We speculate that JAK2 mutation provides 'fertile ground' for the acquisition of chr. 21amp, but once acquired the disease evolution is primarily driven by *DYRK1A* overexpression-associated genomic instability.

Limitations of our study include that we used SNP arrays rather than WGS to call the initial incidence of chr. 21amp and chromothripsis, and performed WGS in a smaller selected cohort to validate and extend these findings. Additionally, the link between *DYRK1A* overexpression and regulation of DNA repair via the DREAM complex is correlative and further study is required to confirm this mechanistically. It is also important to note that other genes in the MAR were DE (*PIGP*, *TTC3*, *MORC3* and *DSCR3*). Although none of these genes show dependency in BP-MPN cell lines and they have not previously been implicated in leukemogenesis, it remains possible that they might act in concert with *DYRK1A* overexpression.

In summary, we describe a high frequency of chromosome 21 amplification in BP-MPN, and identify this as a prognostic biomarker. Through multiomic analysis of patient samples coupled with in vitro and in vivo functional assays, we describe how chr. 21amp creates a therapeutic vulnerability in BP-MPN through a druggable DYRK1A–BCL2 axis. This provides a paradigm for the translation of recurrent regions of aneuploidy to an actionable molecular target.

## Online content

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

**Charlotte K. Brierley** [1,2,3,29]✉, **Bon Ham Yip** [4,29], **Giulia Orlando** [1,29], **Jeremy Wen** [4], **Sean Wen** [1], **Harsh Goyal** [5,6,7,8], **Max Levine** [9], **G. Maria Jakobsdottir** [10,11], **Avraam Tapinos** [10,11], **Alex J. Cornish** [12], **Antonio Rodriguez-Romera** [1], **Alba Rodriguez-Meira** [1,13,14], **Matthew Bashton** [15], **Angela Hamblin** [3], **Sally Ann Clark** [1], **Joseph C. Hamley** [1], **Olivia Fox** [16], **Madalina Giurgiu** [17,18], **Jennifer O'Sullivan** [1,19], **Lauren Murphy** [1], **Assunta Adamo** [1], **Aude Anais Olijnik** [1], **Anitria Cotton** [4], **Emily Hendrix** [20,21], **Shilpa Narina** [20,21], **Shondra M. Pruett-Miller** [20,21], **Amir Enshaei** [22], **Claire Harrison** [19], **Mark Drummond** [23], **Steven Knapper** [24], **Ayalew Tefferi** [25], **Iléana Antony-Debré** [26,27,28], **James Davies** [1], **Anton G. Henssen** [17,18], **Supat Thongjuea** [1], **David C. Wedge** [10,11], **Stefan N. Constantinescu** [5,6,7,8], **Elli Papaemmanuil** [2,9], **Bethan Psaila** [1,3,8,30], **John D. Crispino** [4,30]✉ & **Adam J. Mead** [1,3,30]✉

[1]Medical Research Council (MRC) Weatherall Institute of Molecular Medicine (WIMM) and NIHR Biomedical Research Centre, University of Oxford, Oxford, UK. [2]Computational Oncology Service, Department of Epidemiology & Biostatistics, Memorial Sloan Kettering Cancer Center, New York, NY, USA. [3]Department of Haematology, OUH NHS Foundation Trust, Oxford, UK. [4]Division of Experimental Haematology, St Jude Children's Research Hospital, Memphis, TN, USA. [5]Ludwig Institute for Cancer Research Brussels, Brussels, Belgium. [6]de Duve Institute, Université Catholique de Louvain, Brussels, Belgium. [7]Walloon Excellence in Life Sciences and Biotechnology (WELBIO) Department, WEL Research Institute, Wavre, Belgium. [8]Ludwig Institute for Cancer Research, Nuffield Department of Medicine, Oxford University, Oxford, UK. [9]Isabl Inc., New York, NY, USA. [10]Division of Cancer Sciences, University of Manchester, Manchester Academic Health Science Centre, Manchester, UK. [11]Christie Hospital, The Christie NHS Foundation Trust, Manchester Academic Health Science Centre, Manchester, UK. [12]Division of Genetics and Epidemiology, Institute of Cancer Research, London, UK. [13]Broad Institute of MIT and Harvard, Cambridge, MA, USA. [14]Department of Cancer Biology, Dana-Farber Cancer Institute, Boston, MA, USA. [15]The Hub for Biotechnology in the Built Environment, Department of Applied Sciences, Faculty of Health and Life Sciences, Northumbria University, Newcastle upon Tyne, UK. [16]Oxford Regional Genetics Laboratories, Oxford University Hospitals NHS Foundation Trust, Oxford, UK. [17]Department of Pediatric Oncology/Hematology, Charité-Universitätsmedizin Berlin, Berlin, Germany. [18]Experimental and Clinical Research Center (ECRC) of the MDC and Charité Berlin, Berlin, Germany. [19]Department of Haematology, Guys and St Thomas' NHS Foundation Trust, London, UK. [20]Center for Advanced Genome Engineering,

St. Jude Children's Research Hospital, Memphis, TN, USA. [21]Department of Cell and Molecular Biology, St. Jude Children's Research Hospital, Memphis, TN, USA. [22]Wolfson Childhood Cancer Research Centre, Newcastle University, Newcastle upon Tyne, UK. [23]Department of Haematology, Beatson West of Scotland Cancer Centre, Glasgow, UK. [24]Division of Cancer & Genetics, School of Medicine, Cardiff University, Cardiff, UK. [25]Division of Hematology, Mayo Clinic, Rochester, MN, USA. [26]INSERM, UMR 1287, Villejuif, France. [27]Gustave Roussy, Villejuif, France. [28]Université Paris Saclay, Gif-sur-Yvette, France. [29]These authors contributed equally: Charlotte K. Brierley, Bon Ham Yip, Giulia Orlando. [30]These authors jointly supervised this work: Bethan Psaila, John D Crispino, Adam J Mead. ✉e-mail: charlotte.brierley@imm.ox.ac.uk; john.crispino@stjude.org; adam.mead@imm.ox.ac.uk

## Methods

### Primary patient samples

Peripheral blood and bone marrow samples were collected from patients with BP-MPN and healthy donors from the PHAZAR study (A phase Ib study to assess the safety and tolerability of oral Ruxolitinib in combination with 5-azacitidine in patients with advanced phase myeloproliferative neoplasms (MPN), including myelodysplastic syndromes (MDS) or acute myeloid leukaemia (AML) arising from MPN, Research Ethics Committee: 4/WM/1260; 19 January 2015, West Midlands), the INForMeD Study (Investigating the genetic and cellular basis of sporadic and Familial Myeloid Disorders, Research Ethics Committee: 199833, 26 July 2016, University of Oxford) and the INSERM biobank (approved by the Inserm Institutional Review Board Ethical Committee, project C19-73, agreement 21-794, CODECOH no. DC-2020-4324). Patients and healthy donors provided written, informed consent in accordance with the Declaration of Helsinki for sample collection and use in research.

Cells were subjected to Ficoll gradient centrifugation and, for some samples, CD34 enrichment was performed using immunomagnetic beads (Miltenyi). Total MNCs or CD34$^+$ cells were frozen in FBS supplemented with 10% DMSO for further analysis. Cryopreserved peripheral blood MNCs stored in FCS with 10% DMSO were thawed and processed by warming briefly at 37 °C, followed by gradual dilution into RPMI-1630 supplemented with 10% FCS and 0.1 mg ml$^{-1}$ DNase I, centrifugation at 500$g$ for 5 min and washing in FACS buffer (PBS + 2 mM EDTA + 10% FCS). Before sorts for proliferation and viability assays, cells were thawed and left overnight in StemSpan (StemCell) supplemented with 100 ng ml$^{-1}$ of SCF, TPO and FLT3-L (Peprotech).

### Cell lines

HEL and SET2 cells were obtained from the American Type Culture Collection (ATCC) and were maintained in culture in RPMI-1630 supplemented with 10% FCS and 1% penicillin-streptomycin. SET2 cells were supplemented with 20% FCS. HEK293T cells were maintained in culture in DMEM supplemented with 10% FCS and 1% penicillin-streptomycin. All cell lines underwent regular mycoplasma testing.

### Targeted bulk next generation sequencing

Bulk genomic DNA (gDNA) from patient samples' mononuclear or CD34$^+$ cells was isolated using DNeasy Blood & Tissue Kit (Qiagen) per the manufacturer's instructions. Targeted sequencing was performed using an International Organization for Standardization (ISO 15189:2012) accredited Illumina TruSeq Custom Amplicon Panel including 32 gene mutation hotspots and exons frequently mutated in myeloid malignancies (~56,000 bp, 341 amplicons)[64]. See Supplementary Table 11 for the gene list. Sequencing was performed with a MiSeq sequencer (Illumina), according to the manufacturer's protocols. Results were analyzed after alignment of the reads using an in-house pipeline[64]. All pathogenic variants were manually checked using Integrative Genomics Viewer software.

### SNP array sample preparation

Bulk gDNA from patients' MNCs was isolated using the DNeasy Blood & Tissue Kit (Qiagen) per the manufacturer's instructions. We used 250 ng of gDNA for hybridization on an Illumina Infinium OmniExpress v.1.3 BeadChips platform.

### WGS

Bulk gDNA from patient samples' CD3$^+$ depleted cells was isolated using DNeasy Blood & Tissue Kit (Qiagen) per the manufacturer's instructions. The concentration and purity of gDNA were verified on Qubit and nanodrop, and samples underwent PCR-free library preparation before 80–100× WGS.

### Complex SV clustering analysis

SV rearrangements were grouped using ClusterSV (v.1.1.0; https://github.com/cancerit/ClusterSV/) to identify complex events. ClusterSV takes into consideration the total number and orientation of SVs in a sample, grouping rearrangements that occur in close chromosomal proximity and are unlikely to have occurred by chance. The genetic proximity and occurrence of specific SVs suggest that they arise from the same biological processes. SV rearrangement groups were then classified as simple or complex genomic events, as described previously[27]. In brief, some clusters contain single or <3 SV events, often of the same type, and are considered 'simple' SV clusters, while others contain ≥3 interconnected SVs of varying types and are considered 'complex' events. Events were classed as chromothripsis-like where three of four criteria were met: Cluster Size: the cluster must contain at least 5 grouped SVs; Fragment Join: the cluster's fragment join must be ≥0.05, indicating the specific distribution of the SVs; Interleaved Chromothripsis Events: the cluster should include at least 4 interleaved events, where SVs occur close to each other on the same chromosome; CNA Oscillations: the cluster must exhibit either ≥4 CNA oscillations between 2 states or ≥5 CNA oscillations between 3 states.

### DNA FISH

Primary patient cells were thawed as previously described and suspended in 90% RPMI, 10% FCS. Cells were cultured for 24 h at 37 °C with exposure to Colcemid (KaryoMAX Colcemid solution in HBSS, Gibco, cat. no. 15210040, 10 µl ml$^{-1}$) 16 h before collection. Cell culture, collection and slide making were undertaken according to standard protocols[65,66].

FISH investigations were undertaken using the Cytocell EWSR1/ERG probe (Cytocell, cat no. LPS 008). Co-denaturation and hybridization were carried out using an Abbott Thermobrite system; co-denaturation, hybridization and wash were carried out according to standard protocols[65,66]. Images were visualized using a Nikon eclipse fluorescence microscope and captured using Cytovision software v.7.4 (Applied Imaging, ×1,000 resolution).

### FACS

Bulk and single-cell FACS were performed using Becton Dickinson (BD) Fusion I and BD Fusion II instruments, as previously described[19,67–69]. Experiments involving isolation of Lin$^-$CD34$^+$ (HSPCs) included single-color-stained controls and fluorescence minus one controls. Antibodies used for cell staining are detailed in Supplementary Table 12 and included 0.5 µg of hash-tagged oligonucleotides (BioLegend), to enable hashing and subsequent demultiplexing and doublet exclusion for samples. HSPCs were stained for 30 min at 4 °C, washed in PBS + 5% FCS twice and passed through a 70-mm mesh cell strainer before sorting. For bulk sorts (10×, RNA-seq, ATAC-seq and functional validation studies), live Lin$^-$CD34$^+$ cells were sorted into 1.5-ml Eppendorf tubes or round-bottomed 96-well plates (Corning). We used 7-aminoactinomycin D for dead cell exclusion. Flow cytometry profiles were analyzed using FlowJo software (v.10.7.1, BD Biosciences). See Extended Data Fig. 8a for the gating strategy.

### RNA-seq of HSPCs

In total, 200 CD34$^+$Lin$^-$ cells were isolated by FACS and sorted directly into 8 µl of lysis buffer (0.2% Triton X-100, Sigma) containing oligo-dT primers (2.5 µM, IDT, cat. no. 51-01-15-01), dNTP mix (2.5 µM, Life Technologies, cat. no. 19155) and RNase inhibitor (10 U µl$^{-1}$, Takara (Clontech), cat. no. 2313A) aliquoted into a 96-well PCR plate (Thermo Fisher, segmented semi-skirted, cat. no. AB-0900) on ice. Cell lysis, reverse transcription and PCR amplification (20 cycles) were performed using the Smart-Seq 2 kit (SMARTScribe, Takara (Clontech), cat. no. 639537), as previously published[70]. PCR products were purified using Ampure XP beads (0.6:1 bead ratio, Becker Coulter, A63881) and quantified using Qubit High-Sensitivity kit (Thermo Fisher, Q32854), before the

tagmentation and indexing of 1 ng of complementary DNA using the Illumina Nextera XT DNA sample preparation kit (Illumina, cat. no. FC-131-1024), according to the manufacturer's instructions. Libraries were purified using Ampure XP beads (0.8:1 bead ratio), before quantification using the Qubit High-Sensitivity kit. The quality of cDNA traces and indexed libraries was assessed using the High-Sensitivity DNA Kit in a Bioanalyzer instrument (Agilent, 5067-4626). Libraries were pooled and sequenced on the NextSeq 500 platform (Illumina) using a NextSeq 500/550 High Output v.2.5 (75 cycle) sequencing kit (Illumina, 20024906), generating 75-bp single-end reads.

## ATAC-seq of HSPCs

ATAC-seq was performed on 1,000 CD34$^+$Lin$^-$ cells. Primary patient cells were sorted directly into 11.25 µl of lysis buffer mix containing 0.25 µl of 1% digitonin (Promega, G9441), 0.25 µl of 10% Tween-20 (Sigma), 8.25 µl of 1 × nuclease-free PBS (Thermo Fisher, AM9625) and 2.5 µl of nuclease-free water (Thermo Fisher, 10977049) aliquoted into a well of a 96-well PCR plate (Thermo Fisher, segmented semi-skirted, cat. no. AB-0900) on ice. After sorting, 12.5 µl of 2 × tagmentation DNA (TD) buffer and 1.25 µl of Tn5 transposase (Illumina, 20034198) were added before incubation at 37 °C for 30 min. The reaction was stopped using MinElute Reaction Cleanup kit (Qiagen 28204) and the samples eluted in 10 µl of warmed EB (10 mM Tris-HCL, pH 8).

Samples were amplified and indexed using NEBNext High-fidelity 2X Mastermix (NEB M0541L) and customized HPLC-purified Nextera indexed primers (IDT) using the following PCR program: 72 °C 5 min, 98 °C 30 s, 13 cycles of 98 °C 10 s, 63 °C 30 s, 72 °C 1 min. Library quality was assessed using High-Sensitivity DNA Kit in a Bioanalyzer instrument (Agilent, 5067-4626) and quantitated by qPCR using the NEBNext library quantitation kit (NEB, E7630L). Libraries were pooled at 4 nM and sequenced on the NextSeq 500 platform (Illumina) using a NextSeq 500/550 High Output v.2.5 (75 cycle) sequencing kit (Illumina, 20024906), generating 40-bp pair-end reads.

## In vitro liquid culture primary patient viability assays

A total of 500 cells per well were isolated by FACS into round-bottomed 96-well plates (Corning). Cells were plated in 100 µl per well of media (Stemspan SFEM (StemCell Technologies, cat no. 09650) + 1% Pen/Strep) supplemented with the cytokines detailed in Supplementary Table 13 at 10%. Cells were rested overnight at 37 °C in 5% CO$_2$ and treated with either EHT1610 (MedChem Express, cat. no. HY-111380) or GNF2133 (MedChem Express, cat. no. 555725) ± DMSO or DMSO only control the following day. Medium was replenished on day 3 with the full cytokine cocktail of EPO, FLT3-L, G-CSF, IL-6, GM-CSF, IL-3, TPO and SCF (Supplementary Table 13). Whole wells per condition were analyzed for viability by FACS using the BD Fortessa X20 (BD Biosciences) on days 1, 5 and 8.

## DYRK1A CRISPR KO cell line creation

DYRK1A$^{-/-}$ cells were created using CRISPR–Cas9 technology by the Center for Advanced Genome Engineering (CAGE), St Jude Children's Research Hospital, Memphis, TN, USA. Briefly, 500,000 HEL or SET2 cells were transiently transfected with precomplexed ribonucleoproteins consisting of 100 pmol of chemically modified single guide RNA (CAGE694.DYRK1A.g1; Supplementary Table 14)), 33 pmol of SpCas9 protein (St. Jude Protein Production Core) and 200 ng of pMaxGFP (Lonza) via nucleofection (Lonza, 4D-Nucleofector X-unit) using solution P3 and program EO100 (SET2) or SF solution and program DC102 (HEL). Nucleofections were done in a 20-µl cuvette according to the manufacturer's recommended protocol. At 5 d post nucleofection, transfected cells (GFP$^+$) were single-cell sorted by flow cytometry into 96-well tissue culture-treated plates. Cells were clonally expanded and screened for the desired targeted modification via targeted deep sequencing using gene-specific primers with partial Illumina adapter overhangs (Supplementary Table 14)[71]. Genotyping of clones was performed using CRIS.py (v.1)[72]. KO clones were identified as clones containing only out-of-frame indels. Final clones were confirmed negative for mycoplasma using MycoAlert Plus Mycoplasma Detection Kit (Lonza) and authenticated using the PowerPlex Fusion System (Promega) at the Hartwell Center for Biotechnology at St. Jude.

## Lentiviral production and transduction

293T cells (ATCC) were grown to 70–80% confluence in 10-cm dishes. Cells were then transfected in 1:9:9 ratio (packaging plasmid pMD2.G/packaging plasmid psPAX2/lentiviral plasmid) using TurboFect Transfection Reagent (Thermo Fisher) according to the manufacturer's instructions. Fresh medium was replaced after 24 h, and viral supernatant was collected at 48 h after transfection and 0.45-µm filtered to remove cell debris. Transfection of TRIPZ inducible lentiviral human DYRK1A shRNA clone V3THS_376671, V3THS_376672 and nonsilencing shRNA control (Supplementary Tables 14 and 15) was used to produce shRNAs for DYRK1A knockdown. Transduction of SET2 and HEL cells was performed by centrifugation at 800g, 30°C for 90 min. Puromycin selection of transduced cells was performed at 2 µg ml$^{-1}$. Doxycycline induction of shRNA expression at 1 µg ml$^{-1}$ was performed at the same time. Transfection of lentiviral vector SJL12 EF1a-Luciferase-P2A-GFP (St Jude Vector Core) was used to produce lentiviruses for stable luciferase expression. Transduction of WT SET2, SET2 KO clone 11H1, SET2 KO clone 14B5, WT HEL, HEL KO clone 1B12 and HEL KO clone 1A5 by SJL12 EF1a-Luciferase-P2A-GFP lentiviruses was performed by centrifugation at 800g, 30 °C for 90 min. Cell sorting of the GFP$^+$ population was performed by BD FACSAria III to establish stable cell lines.

## Cell growth assay

DYRK1A knockdown and KO cells were seeded at the same density into 96- or 24-well plates. Viable cell counts were determined by trypan blue exclusion for 5 consecutive days. Medium was replenished every second day to maintain the same volume. Dose–response curves were calculated in PRISM v.8.0.

## IncuCyte cell count proliferation assay

Cell growth was measured utilizing the IncuCyte Live Cell Imager system (Essen BioSciences). Briefly, HEL cells treated with DYRK1A inhibitor EHT1610 or GNF2133 at various concentrations and HEL cells transduced with DYRK1A or scramble control shRNA were plated in a 24-multiwell culture plate at 1,000 cells per cm$^2$. Culture plates were sited into the IncuCyte Live Cell imager, and images were captured using the phase contrast channel and were taken every 4 h in the IncuCyte ZOOM platform (Essen BioSciences). Nine image sets were acquired from several points of the well, using a ×10 objective lens, and all the conditions were run in triplicate.

## In vivo bioluminescence imaging of murine xenograft model

NOD scid gamma mice (NSG, stock no. 5557, The Jackson Laboratory) were sub-lethally irradiated with 100 rad and intravenously transplanted with luciferase-expressing WT and KO cell lines from SET2 and HEL cells (1 × 10$^6$ cells per mouse). Transplanted mice underwent in vivo bioluminescence imaging at various times as specified for each experiment. Animals were monitored daily and were euthanized upon signs of leukemia onset (decreased activity and hind limb paralysis). Bioluminescent imaging and data analysis were performed using a Xenogen IVIS Spectrum system and Living Image v.4.7 software (Perkin Elmer). Mice were injected intraperitoneally with D-luciferin (Perkin Elmer) at 150 mg per kg body weight and, after 3–5 min to allow substrate distribution, anesthetized for imaging using 2% isoflurane delivered at 2 l min$^{-1}$ in O$_2$. Images were acquired using 1-min exposures with small binning and with shortening of exposure times when signals were saturated. Total flux measurements (photons per second) were quantified through application of a contour drawn around the target region. Images were normalized to the same color scale by setting maximum signal of luminescent activity as appropriate for the experiment.

All animal experiments were approved by the St. Jude Children's Research Hospital Institutional Animal Care and Use Committee and performed under protocol number 657-100655.

## Stranded messenger RNA-seq for SET2 cell line RNA-seq data

Total RNA was extracted by NucleoSpin RNA Plus (Takara). RNA was quantified using the Quant-iT RiboGreen RNA assay (Thermo Fisher) and quality checked by the 2100 Bioanalyzer RNA 6000 Nano assay (Agilent) or 4200 TapeStation High Sensitivity RNA ScreenTape assay (Agilent) before library generation. Libraries were prepared from total RNA with the TruSeq Stranded mRNA Library Prep Kit according to the manufacturer's instructions (Illumina, PN 20020595). Libraries were analyzed for insert size distribution using the 2100 BioAnalyzer High Sensitivity kit (Agilent), 4200 TapeStation D1000 ScreenTape assay (Agilent) or 5300 Fragment Analyzer NGS fragment kit (Agilent). Libraries were quantified using the Quant-iT PicoGreen dsDNA assay (Thermo Fisher) or by low-pass sequencing with a MiSeq nano kit (Illumina). Paired-end 100 cycle sequencing was performed on a NovaSeq 6000 (Illumina).

## Evaluation of DNA damage

To evaluate the effect of DNA damaging agent etoposide on the proliferation of SET2 cells, parental SET2 cells or *DYRK1A* KO clones were serum-starved for 16 h to synchronize the cells to G0 phase[50]. After serum starvation, cells were seeded in a 96-well plate ($2 \times 10^4$ cells in 100 µl of media) and treated with DMSO or different concentrations of etoposide (cat. no. E1383, Sigma). After 48 h of treatment, 100 µl of TiterGlo reagent (cat. no. G7571, Promega) was added to cells according to the manufacturer's manual. Luminescence was read with the Agilent BioTek Microplate Reader. All the luminescence readings were normalized to DMSO. The dose–response curves were generated using GraphPad (Prism v.9) software.

To evaluate DNA damage caused by etoposide treatment or irradiation, γ-H2AX staining was performed. Parental SET2 cells or *DYRK1A* KO clones were serum-starved for 16 h and then treated with DMSO or 3 µM etoposide, or irradiated at 200 rad of γ radiation. At 8 h after DMSO or etoposide treatment, or 2 h after irradiation, cells were collected and washed once with ice-cold PBS. Cells were then fixed with 2% paraformaldehyde buffered in PBS at 37 °C for 10 min. Permeabilization of cells was conducted in 90% methanol on ice for 30 min. After washing in PBS, cells were stained with AF488-conjugated γ-H2AX antibody (cat. no. ab195188, clone EP854(2)Y, Abcam) at 1:50 for 1 h at room temperature. After washing in PBS, the cells were stained with DAPI at 1 µg ml$^{-1}$ in PBS with 0.5% BSA and 0.1% saponin for 5 min. The samples were analyzed with a BD FACSymphony A3 flow cytometer.

## Apoptosis assays with cell lines

HEL cell lines were cultured in vitro in the media conditions outlined above and plated at 20,000 cells per well in a 96-well plate. Cells were treated with GNF2133 (MedChem Express, cat. no. 555725), alone or in combination with navitoclax (MedChem Express, cat. no. HY-10087), at indicated concentrations, incubated at 37 °C in 5% $CO_2$ for 24 h and then analyzed by flow cytometry on an Attune NxT (Invitrogen, Model AFC2) using the Annexin kit (eBioscience, cat. no. 88-8007-74), the per manufacturer's instructions. See Extended Data Fig. 8b for the gating strategy. The drug synergy score for the drug combination matrix was calculated using the SynergyFinder R package available through the SynergyFinder web application v2: visual analytics of multi-drug combination synergies (https://github.com/IanevskiAleksandr/SynergyFinder). Synergy scores indicate the percentage of response beyond the expected drug effect when each drug is used in isolation.

## Statistics and reproducibility

Statistical analyses are detailed in figure legends and were performed using GraphPad Prism software (7 or later) or R software (v.4.0.4).

Welch *t*-tests or Mann–Whitney tests for comparisons of individual groups were used for parametric or nonparametric data, respectively. Two-way analysis of variance (ANOVA) and multiple *t*-tests were used to compare experimental groups, as indicated in the figure legends. All *P* values were two-sided and adjusted for multiple comparisons using Benjamini–Hochberg correction, unless otherwise stated. Exact *P* values are given unless $<1 \times 10^{-4}$, in which case they are notated as <0.0001. The numbers of independent experiments, donors and replicates for each experiment are specified in each figure legend. No statistical method was used to predetermine sample size.

## Reporting summary

Further information on research design is available in the Nature Portfolio Reporting Summary linked to this article.

## Data availability

All raw and processed sequencing data generated in this study will be made publically available at the NCBI Gene Expression Omnibus (GEO; https://www.ncbi.nlm.nih.gov/geo/) under accession numbers GSE228060 for CRISPR KO clones, GSE240407 for RNA/ATAC and GSE292030 for single-cell primary patient data. The TARGET-seq single-cell dataset is available in raw and processed format at GEO accession number GSE226340 and SRA accession number PRJNA930152. The raw and processed SNP array data and single-cell (10×) Seurat object generated in this manuscript are available via Zenodo at https://doi.org/10.5281/zenodo.14749739 (ref. 73). Whole genome sequencing data have been deposited at the European Genome-phenome Archive (EGA), which is hosted by the EBI and the CRG, under accession number EGAS00001007483. Further information about EGA can be found at https://ega-archive.org, 'The European Genome-phenome Archive of human data consented for biomedical research' (http://www.nature.com/ng/journal/v47/n7/full/ng.3312.html). Due to ethical restrictions, these datasets cannot be made publicly available. Access to the data can be obtained upon application and approval by the EGA Data Access Committee. Researchers may request access via the EGA portal (https://ega-archive.org), following the appropriate data access procedures, and applications will be reviewed within a 4-week timeframe. Source data are provided with this paper.

## Code availability

A full list of R packages used and scripts to reproduce all figures are available via GitHub at https://github.com/wimm-hscb-lab-published/Brierley_NG_chr21amp and via Zenodo at https://doi.org/10.5281/zenodo.14969163 (ref. 74).

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

## Acknowledgements

We thank the patients and donors, without whom this study would not have been possible. We thank P. Ciccone, N. Hayder and N. Sousos who helped with sample banking; L. Palmer who assisted with data upload; K. Clark, C. Waugh and P. Sopp in the MRC WIMM Flow Cytometry facility which is supported by the MRC Human Immunology Unit and MRC Molecular Haematology Unit; N. Ashley in the MRC WIMM Single Cell Facility; R. Beveridge in the MRC WIMM Virus Screening Facility; and all the staff at Isabl Inc. for facilitating the whole genome sequencing analyses. We also thank the Oxford Genomics Centre at the Wellcome Centre for Human Genetics (funded by Wellcome Trust grant reference no. 203141/Z/16/Z) for the generation and initial processing of the OmniExpress SNP array data. This work was supported by a CRUK Advanced Clinician Scientist Fellowship/Senior Fellowship (to B.P., grant nos. C67633/A29034 and RCCSCF-May24/100001), the Ludwig Cancer Research Institute Associate Member Program (to B.P.), a CRUK Senior Cancer Research Fellowship (to A.J.M., grant no. C42639/A26988), a John Goldman Fellowship (to G.O., grant no. 2021/JGF/005) and a Sir Henry Wellcome Clinical Doctoral Training Fellowship (to C.K.B.; grant no. 220586/Z/20/Z). We acknowledge the CRUK Oxford Centre (grant no. CTRQQR-2021\100002), the National Institute for Health Research (NIHR) Oxford Biomedical Research Centre (BRC), the John Fell Fund (grant nos. 131/030 and 101/517), the EPA fund (grant nos. CF182 and CF170), the MRC WIMM Strategic Alliance awards no. G0902418 and no. MC_UU_12025, and the contribution of the WIMM Sequencing Facility, supported by the MRC Human Immunology Unit and by the EPA fund (grant no. CF268). Additional funding was provided by the National Institutes of Health (NIH) (grant nos. R35CA253096 and P01CA108671), the Wellcome Trust (grant no. 222800/Z/21/Z to A.R.-M.), the MPN Research Foundation (to J.D.C. and A.T.), the Leukemia & Lymphoma Society (to J.D.C.), the Samuel Waxman Cancer Research Foundation (to J.D.C.), the St Jude Children's Research Hospital Cancer Center Grant (no. NCI P30 CA021765) and St. Jude/ALSAC (to J.D.C.). In addition, this research was funded by a Télévie PhD fellowship, Belgium (H.G.), the Ludwig Institute for Cancer Research, Fondation contre le cancer F/2022/2048 (S.N.C.), Salus Sanguinis, Fondation 'Les avions de Sébastien', Projet de recherche FNRS no. T.0043.21 (S.N.C.), WelBio F grant no. 44/8/5–MCF/UIG–10 955 (S.N.C.) and Cancer Research UK RadNet Manchester (grant no.

C1994/A28701 to G.M.J., A.T., D.C.W.). D.C.W. is co-funded by the NIHR Manchester BRC (grant no. NIHR203308), The University of Manchester and the Christie NHS Foundation Trust. The views expressed are those of the authors and not necessarily those of the National Health Service (NHS), the NIH, the NIHR or the Department of Health and Social Care. The results published here are in part based upon data generated by the TCGA Research 1250 Network (https://www.cancer.gov/tcga) and the Beat AML team.

## Author contributions

C.K.B. was responsible for conceptualization, investigation, validation, funding acquisition, methodology, computational analyses and writing (original draft and editing). B.H.Y. and J.W. were responsible for investigation, validation, visualization and writing (editing). G.O. was responsible for investigation, validation, visualization and writing (editing). S.W., A.J.C., J.D. and A.G.H. supervised the computational analyses. H.G., A.R.-R., A.R.-M., O.F., M.G., A.A., A.C. and A.E. performed investigations. M.L., G.M.J., A.T., M.B. and J.C.H. implemented the computational analyses. A.H. was responsible for resources and investigations. S.A.C. and S.N.C. were responsible for investigations and methodology. J.O., L.M., E.H., S.N., S.M.P.-M., C.H., M.D., S.K., A.T. and I.A.-D. were responsible for resources. A.A.O. edited the figures. S.T. was responsible for software. E.P. was responsible for supervision, resources and investigations. B.P. was responsible for conceptualization, resources, formal analysis, supervision, funding acquisition, methodology, writing (original draft and editing) and project administration. J.D.C. was responsible for conceptualization, resources, formal analysis, supervision, funding acquisition, methodology and writing (editing). A.J.M. was responsible for conceptualization, resources, formal analysis, supervision, funding acquisition, methodology, writing (original draft) and project administration. All authors read and approved the submitted manuscript.

## Competing interests

A patent relating to the TARGET-seq technique is licensed to Alethiomics Ltd, a spin out company from the University of Oxford with equity owned by B.P. and A.J.M. E.P. is a founder, equity holder and holds a fiduciary role in Isabl Inc. A.J.C. is an employee of Owkin UK Ltd. The other authors declare no competing interests.

## Additional information

**Extended data** is available for this paper at https://doi.org/10.1038/s41588-025-02190-6.

**Correspondence and requests for materials** should be addressed to Charlotte K. Brierley, John D. Crispino or Adam J. Mead.

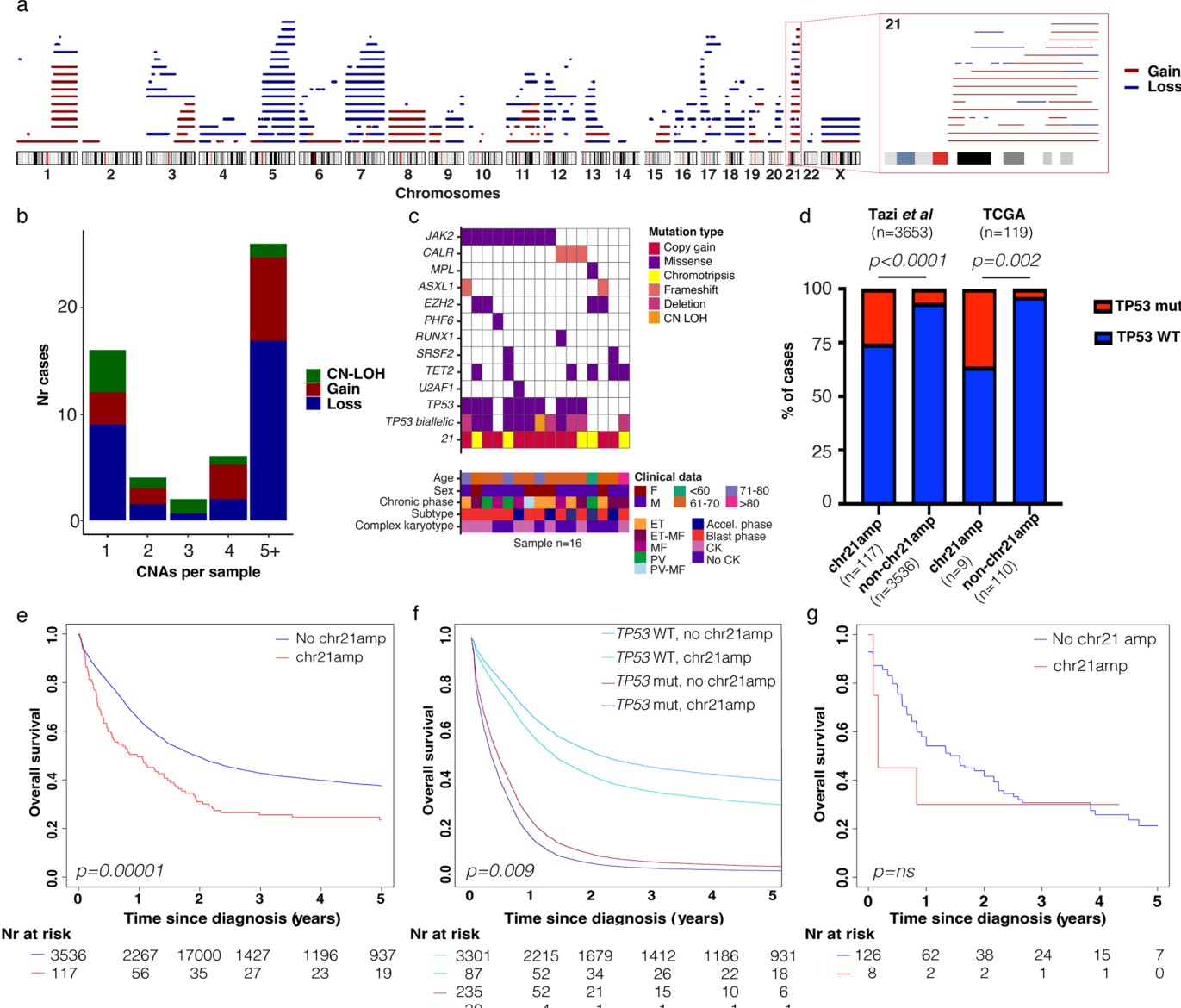

**Extended Data Fig. 1 | Supplemental information on copy number alterations in BP-MPN cohort and chr21amp in external AML datasets. a.** Graphical representation of all copy number gains and losses identified across the cohort by MoChA[21,22]. Events are colored by type (gain=red, loss=blue) and overlay chromosomal ideograms to indicate location. **b.** Number of cases by number of chromosomal alterations. Events are colored by event type (gain=red, loss=blue, CN-LOH=green). **c.** Mutations, *TP53* allelic status, and baseline clinical data for 16 chr21amp patients. CN-LOH=Copy-neutral loss of heterozygosity, F=female, M=Male, ET=Essential Thrombocythemia, ET-MF=ET progressing to Myelofibrosis, MF=Myelofibrosis, PV=Polycythemia rubra vera, PV-MF=PV progressing to MF, Accel phase= Accelerated phase (bone marrow blast % >10 < 20%), CK=complex karyotype. **d.** In both the TCGA L-AML cohort and a large cohort of 3653 *de novo* AML patients, there was a significant enrichment for

mutated and/or deleted *TP53* amongst chr21amp cases (Fisher's exact test, *p=0.0000000002* for Tazi et al., *p=0.002* for the TCGA comparison) **e.** Kaplan Meier survival curve stratified by chr21amp status showing significantly impaired overall survival (OS) for chr21amp amongst 3653 *de novo* AML cases (median OS from diagnosis 0.97 (95%CI 0.55-1.47) vs 1.93years (95% CI 1.77-2.09) *p<0.01* by one-sided Mantel Cox log-rank test). **f.** Cox survival curve stratifying patients in (b) by *TP53* and *chr21amp* status, showing that the adverse impact on survival conferred by chr21amp persists after stratifying by *TP53* status (*p=0.009*, HR for chr21amp alone 1.3 (95% CI 1.1-1.7), mutant *TP53* alone 3.75 (95%CI 3.3-4.3), HR for m*TP53* with chr21amp= 4.71 (95%CI 3.3-6.8), Cox proportional hazards survival model). **g.** The TCGA dataset is underpowered for a survival analysis by chr21amp (median OS in years chr21amp 0.17 years (95%CI 0.17-NA) vs no chr21amp 1.58 years (95%CI 0.92-2.2), *p=0.4*).

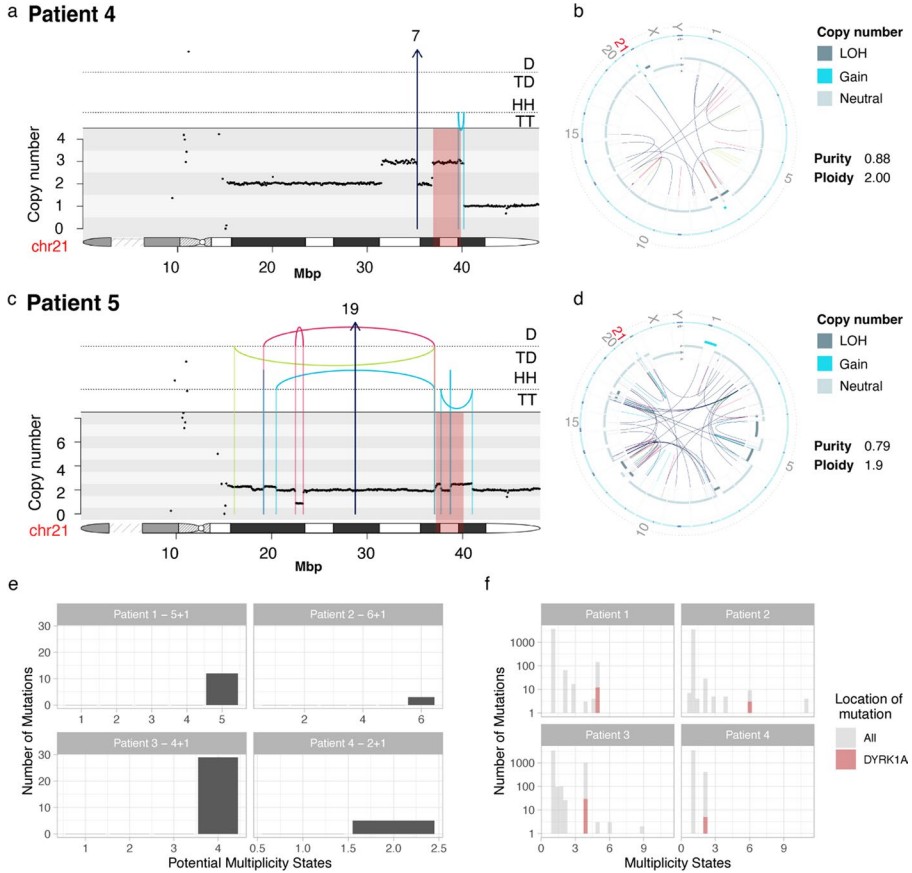

**Extended Data Fig. 2 | Supplemental data on whole genome sequencing and multiplicity states. a & c**. integrated CN & structural variant plots for patients 4 & 5. As in Fig. 2, the top panel shows intra-chromosomal events as arcs between breakpoint loci, and color denotes the type of SV (black=translocation, red=deletion, blue=duplication, green=inversion). Rearrangements are further separated and annotated based on orientation. D: Deletion, TD: Tandem duplication; HH, head-to-head inverted, TT: tail-to-tail inverted. Inter-chromosomal events are shown with arrows denoting the likely partner chromosome. The middle panel shows the consensus copy number across the chr21 ideogram, depicted in the lowest section of each plot to indicate breakpoint location. Patient 4 has a simple breakage-fusion-bridge event, whereas patient 5 shows a low-level copy number gain and inversion event. **b & d**. Circos plots for patients 4 & 5 showing global SV burden, demonstrating clustering around chr21. The outer ring shows the chromosome ideogram. The

middle ring shows the B allelic frequency and the inner ring shows the intra-and inter-chromosomal SVs with the same color scheme as in **a & c**. **e**. Multiplicity states across the gained copy number segment spanning *DYRK1A* for each case. The x-axis indicates the potential multiplicity states that would be expected to be observed for the segment spanning *DYRK1A* in each sample given its copy number state. Shaded bars indicate the number of mutations observed in each multiplicity state. **f**. Multiplicity states observed in segments spanning *DYRK1A* compared to the entire genome. Red shaded regions indicate the number of mutations in each multiplicity state identified in regions spanning *DYRK1A* for each sample. Grey shaded regions indicate the number of mutations at each multiplicity state across the entire genome, including the segment spanning *DYRK1A*. The red and grey bars overlap, rather than being additive. There were insufficient mutations in the region of amplification to assess multiplicity in Patient 5.

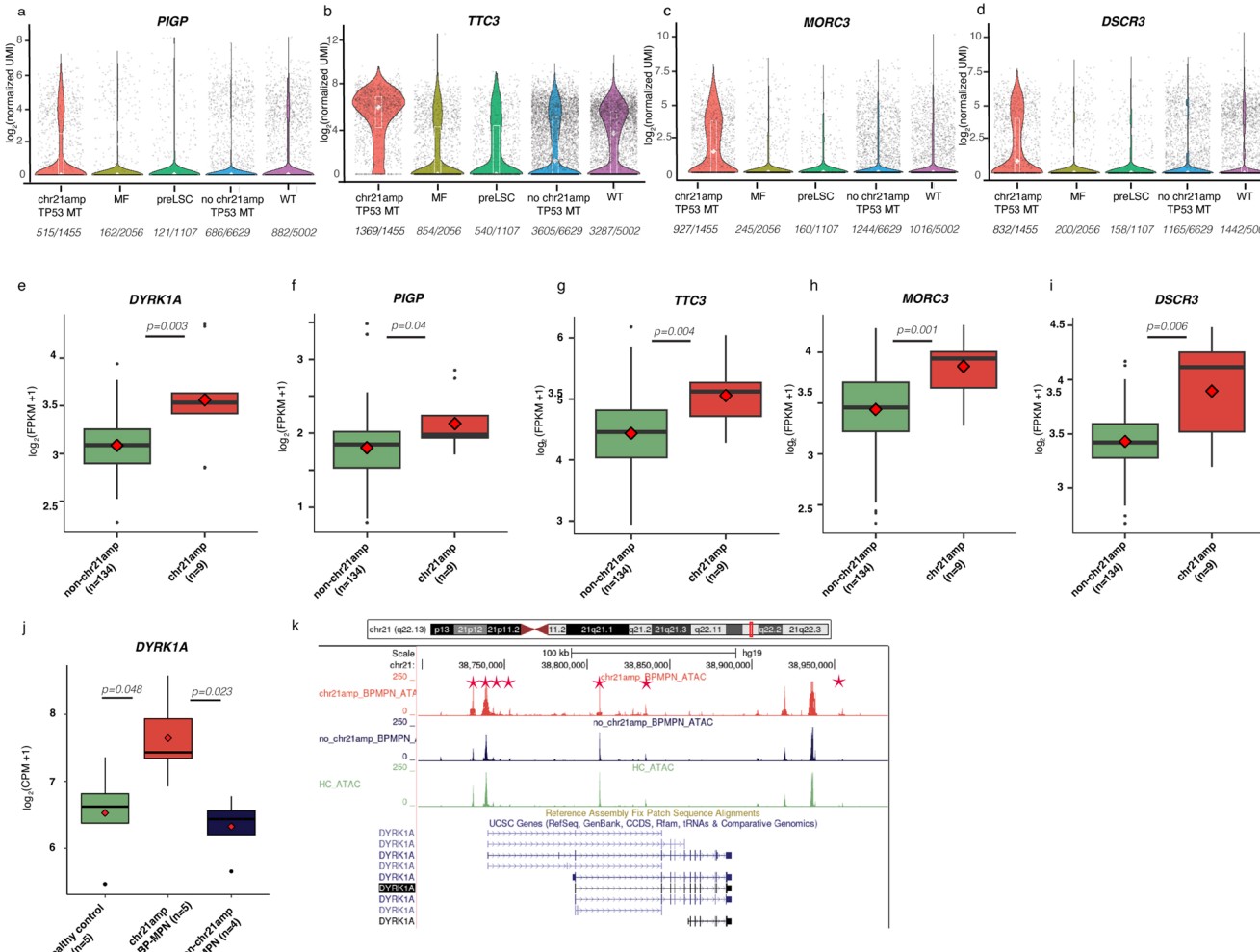

**Extended Data Fig. 3 | Supplemental information on prioritising genes in the minimally amplified region using TARGET-seq. a.-d.** Violin plots of the significantly DE genes in the MAR (log2FC>1, padj<0.05) in chr21amp HSPCs compared to non-chr21amp control cells including myelofibrosis (MF, n=2056 cells from 8 MF donors), pre-leukemic stem cells (preLSC, n=1107 non-mutant phenotypic HSC, identified in 12 BP-MPN donors), *TP53*-mutant-non-chr21amp BP-MPN (no chr21amp *mTP53*, n=6629 cells from 14 BP-MPN donors) and wild type cells (WT, n=5002 from 9 healthy donors). Each dot represents the expression value (log2-normalized UMI count) for each single cell, with median and quartiles shown in white. Expressing cell frequencies are shown on the bottom of each violin plot for each group **a**. *PIGP* **b**. *TTC3* **c**. *MORC3* **d**. *DSCR3*. For each gene, *padj<0.001* when comparing chr21amp_TP53_MT vs all other categories by Wilcoxon rank sum test, adjusted for multiple comparisons by Bonferroni correction. **e-i.** Box plots showing differential expression of

prioritised genes (**e**. *DYRK1A* **f**. *PIGP* **g**. *TTC3* **h**. *MORC3* **i**. *DSCR3*) in the chr21 MAR for AML patients with copy number data available in The Cancer Genome Atlas (n=134 non-chr21amp, n=9 chr21amp) by Wilcoxon rank sum test, adjusted for multiple comparisons by Bonferroni correction. The box-and-whiskers plots show the median and the interquartile range (IQR), with the whiskers extending +/-1.5*IQR. The mean is shown as a diamond. **j**. Box-and-whiskers plot of normalised counts (log2CPM+1) of *DYRK1A* expression on RNA_seq analysis, showing mean (diamond), median and IQR, with the whiskers extending +/-1.5*IQR. *DYRK1A* is overexpressed in chr21amp cases compared to controls (Wilcoxon rank-sum test, *p<0.05* for all comparisons). **k**. ATAC-seq tracks shown in UCSC genome browser window over the *DYRK1A* genomic location by condition (chr21amp in red, healthy control in green, non-chr21amp BP-MPN in blue) with differentially accessible (log2FC >1, *p-adj* < 0.05, DESeq2 analysis) peaks marked with a red star.

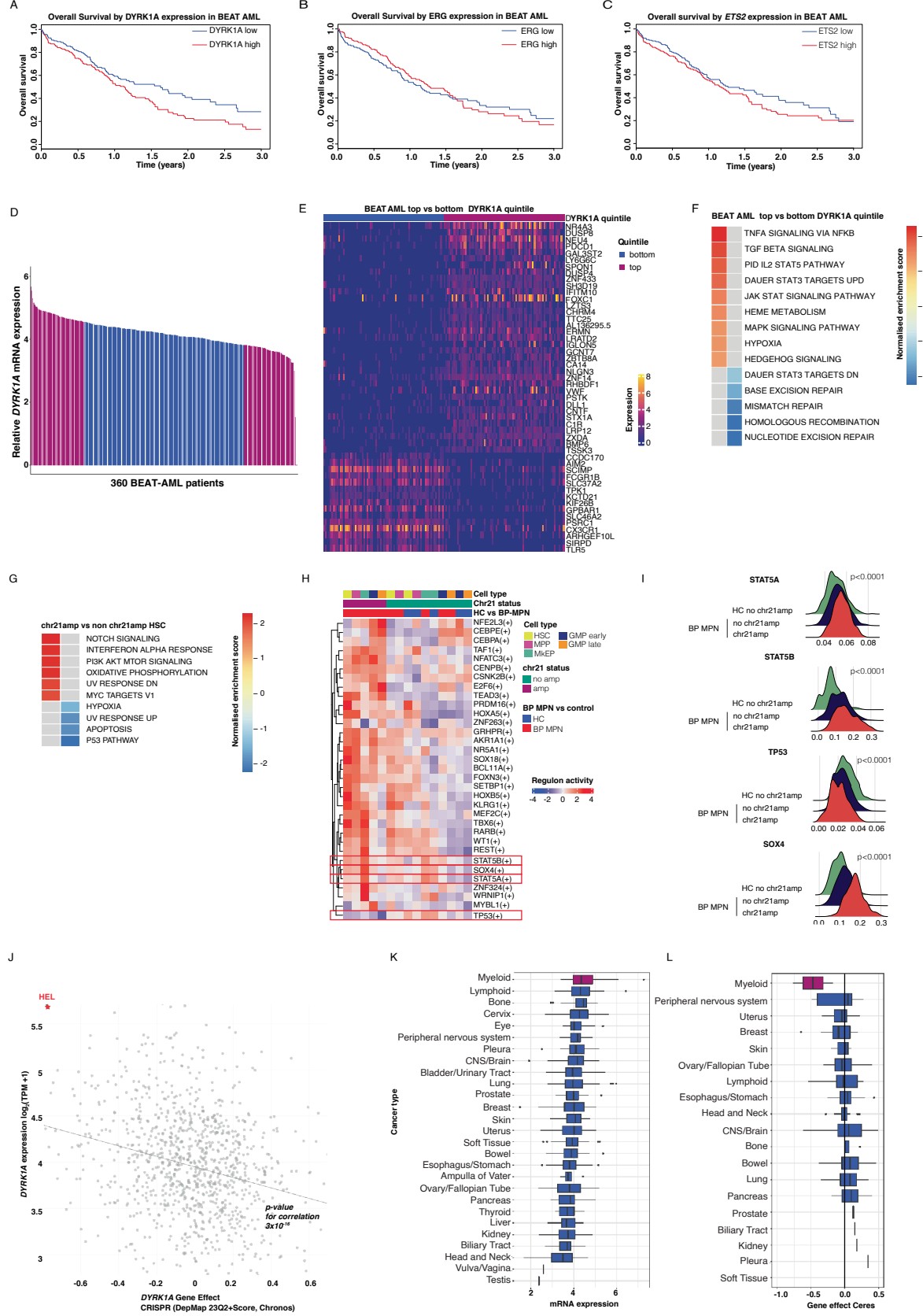

**Extended Data Fig. 4 | See next page for caption.**

**Extended Data Fig. 4 | *DYRK1A* expression in external AML datasets and *DYRK1A*-amplification associated gene regulatory programmes. a**. In the 360-patient BEAT AML cohort, overexpression of *DYRK1A* is associated with adverse outcome (HR 1.44 (95% CI 1.07-1.93, *p-value 0.03), Cox regression analysis*, 3year (y) OS 13.1% (95%CI 7.1-24.4%) vs 28.4%(95%CI 18.8-43.1%, *p=0.02*, Mantel–Cox log-rank test), while *ERG* and *ETS2*, two other genes in the chr21amp minimally amplified region, are not (**b.**: *ERG* 3y OS *ERG* high 16.7% (95%CI 9.3-30.1%) vs 22.0% low (95%CI 13.7-35.4%), *p=0.8* Mantel–Cox log-rank test) **c.**, *ETS2* 3y OS *ETS2* high 19.1%(95%CI 13.2-31.2%) vs 20.3% *ETS2* low (95%CI 10.5-34.8%, *p=0.2* Mantel–Cox log-rank test)) **d**. BEATAML cohort stratified by top (n=72) vs bottom (n=72) quintile of *DYRK1A* expression **e**. Heatmap of top differentially expressed genes of top vs bottom quintiles of the BEAT AML cohort stratified by *DYRK1A* expression. **f**. Hallmark and KEGG pathway GSEA of top altered pathways (NES, Normalized enrichment score >/<1) comparing patients in (**d.**) **g**. GSEA for HALLMARK pathways with normalized enrichment score (NES)>1 in chr21amp vs non-chr21amp HSCs. The heatmap shows the normalized enrichment score and the title indicates the cohort that the geneset is enriched/depleted for. Raw data can be found in Supplementary Table 10. **h**. Heatmap of SCENIC regulon analysis showing that chr21amp and *DYRK1A* overexpression leads to activation of divergent transcriptional programs. Regulons cluster by chr21amp status over cell type, when comparing chr21amp BP-MPN and non-chr21amp BP-MPN with healthy controls. STAT5A and STAT5B regulons are upregulated in a cell-type agnostic manner by chr21amp, while TP53 is downregulated. **i**. SCENIC regulons scored by the AUCell algorithm upregulated in chr21amp include *STAT5A, STAT5B* and *SOX4*, with *TP53* downregulated, corroborating the GSEA findings. Statistical significance testing performed by Kruskal Wallis test with Benjamini Hochberg adjustment for multiple testing. **j**. Plot of *DYRK1A* gene expression vs gene dependency from the Broad DepMap database, showing the chr21amp MPNAML cell line HEL as an outlier. The non-chr21amp K562 leukemia cell line is also labelled. Linear regression analysis shows correlation between expression and dependency (Pearson correlation coefficient -0.242, slope -5.67E-1, *p-value <0.001*) **k**. mRNA expression of *DYRK1A* by cell line lineage in the DepMap database. **l**. Ceres gene dependency scores for *DYRK1A* by cell line lineage in DepMap.

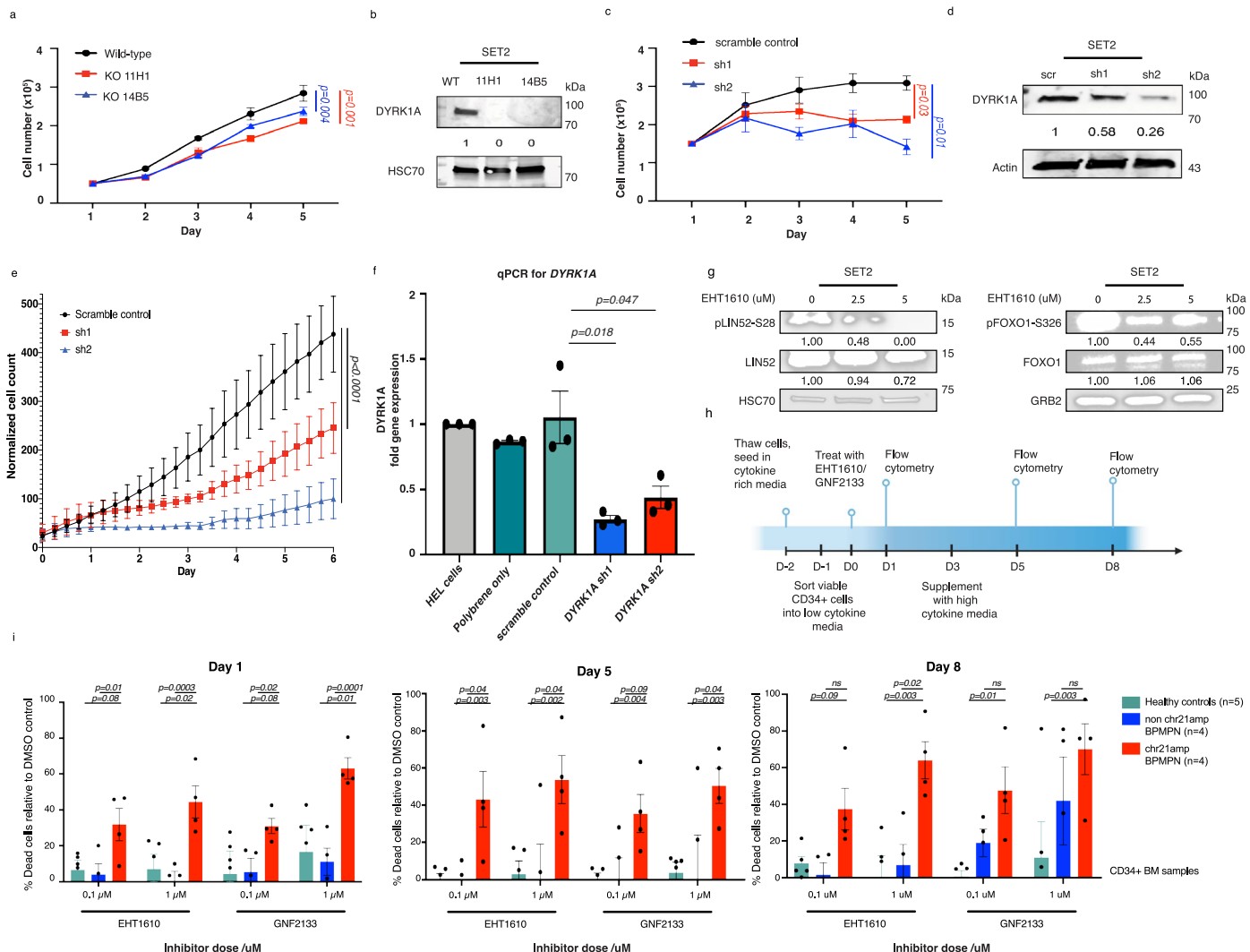

**Extended Data Fig. 5 | Supplemental experiments functionally validating *DYRK1A* as a target in BP-MPN. a**. Cell counts for two SET2 *DYRK1A* KO clones (14B5 & 11H1) generated by CRISPR/Cas9 and expanded in culture, confirming adverse effect on proliferation by *DYRK1A* KO over time (Shown are mean +/- SEM, n=3 biological replicates, *p<0.05* by day 5 by ANOVA). **b**. Western blot showing the knockout of *DYRK1A* in the 14B5 & 11H1 SET2 cell clones. Densitometric values normalized to HSC70 (representative of n=3 experiments). **c**. Cell proliferation assay for SET2 cells transduced with lentiviruses expressing *DYRK1A* specific shRNA or scramble control (Shown are mean +/- SEM, n=3 biological replicates, *p<0.05* by day 5, by ANOVA). **d**. Western blot showing the knockdown of *DYRK1A* expression in SET2 cells following transduction with lentiviruses expressing target specific shRNA or scramble control. Densitometric values were normalized to actin (representative of n=3 experiments). **e.** Independent replicate experiment of shRNA knockdown experiments. HEL cells were treated with lentiviruses expressing *DYRK1A* specific shRNA or scramble control, and

placed in the Incucyte for serial cell counts (summary of 3 biological replicates, mean +/- SEM, *p<0.001* by ANOVA). **f.** Knockdown of *DYRK1A* was validated by qRT-PCR (n=3 replicates, shown are mean +/- SEM, compared by t-test, p-values shown are two-sided). **g.** Western blots for phosphorylation of LIN52 at S28, FOXO1 at S329, total LIN52 and total FOXO1 protein levels after 4 hours of treatment with increasing doses of the DYRK1A inhibitor EHT1610 in SET2 cells are shown. Densitometric values were normalized to HSC70 and GRB2 protein levels, respectively (representative of n=2 experiments). **h.** Experimental layout for primary patient experiments. **i.** Timecourse of viability readouts for primary patient cells on days 1,5,8 post treatment with DYRK1A inhibitors EHT1610 and GNF2133 at 0.1 and 1μM doses. On the y axis is % cell death relative to DMSO control. Each sample (n=5 healthy controls, n=4 non-chr21amp and n=4 chr21amp patients) is shown by a dot, with mean and SD depicted in the boxplot. Significance testing by t-test with Bonferroni correction for multiple testing, shown are the adjusted q values *<0.1 **<0.05, ***<0.001.

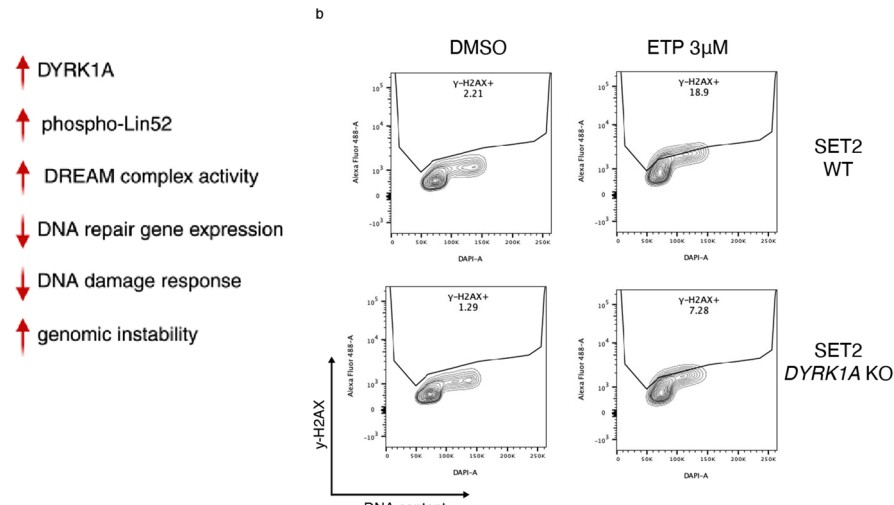

**Extended Data Fig. 6 | Supplemental data on *DYRK1A* and genomic instability. a**. Schematic diagram summarising the proposed model whereby *DYRK1A* overexpression impacts DNA damage response and genomic instability. **b**. Representative flow cytometry plots showing % of SET2 WT and SET2 *DYRK1A* KO cells staining positive for γ- H2AX at 8hrs post 3μM etoposide treatment.

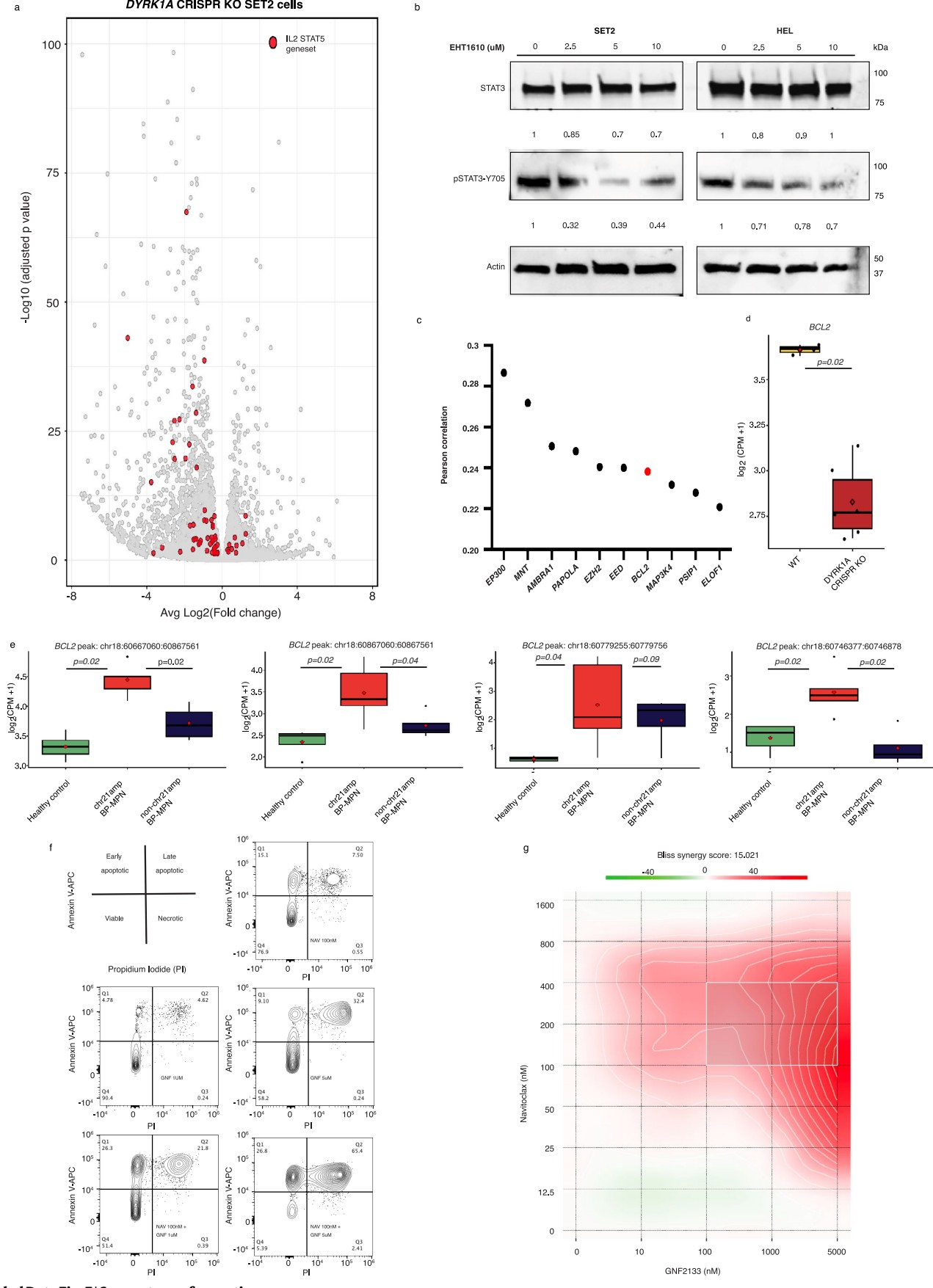

**Extended Data Fig. 7 | See next page for caption.**

**Extended Data Fig. 7 | Supplemental experiments evaluating the mechanism of *DYRK1A* and the concomitant upregulation of BCL2, supporting a rationale for therapeutic synergy with co-inhibition of DYRK1A and BCL2.**
**a**. Volcano plot of differentially expressed genes on RNA-seq of SET2 cell lines who underwent CRISPR KO of *DYRK1A* (n=5, 2 clones) vs WT (n=3) highlighting downregulation of HALLMARK STAT5 signalling pathway genes (GSEA NES -2.08, FWER p-value <0.01, DESeq2 analysis). **b**. Western blot for phosphorylation of STAT3 at Y705 and total STAT3 with actin normalization, at baseline and after 30 minutes of treatment with increasing doses of the DYRK1A inhibitor EHT1610 in HEL and SET2 cells. Blot representative of n=2 experiments. **c**. Co-dependencies for *DYRK1A* in DepMap, highlighting *BCL2* amongst the top 10 genes co-dependent on CRISPR screen (Pearson's correlation co-efficient 0.24). **d**. *DYRK1A* KO SET-2 cell line clones downregulate *BCL2* (n=5 CRISPR KO vs n=3 WT, *p-0.02 Wilcoxon rank-sum test*. The box-and-whiskers plots show the median and the interquartile range (IQR), with the whiskers extending +/-1.5*IQR. The mean is shown as a diamond.). **e**. ATAC-seq peaks in chr21amp vs non-chr21amp cells showing upregulation of peaks across the *BCL2* gene body in chr21amp patients (n=5 chr21amp vs n=4 non-chr21amp vs n=5 healthy controls, *two-sided paired Wilcoxon rank-sum test*. The box-and-whiskers plots show the median and the interquartile range (IQR), with the whiskers extending +/-1.5*IQR. The mean is shown as a diamond.). **f**. Representative flow cytometry plots for Annexin V PI apoptosis assay demonstrating synergy between the DYRK1A inhibitor GNF2133 ('GNF") and the BCL2/BCL-XL inhibitor navitoclax ("NAV"). The middle left panel shows that GNF 1 µM induces apoptosis in 9.4% of cells while NAV 100nM does so in 22.6% of cells (top right). In combination (bottom left), NAV 100+GNF induce apoptosis in 46.1%. **g**. Bliss synergy score and matrix contour plot highlighting areas of greatest synergy between DYRK1A and navitoclax inhibitor dosing.

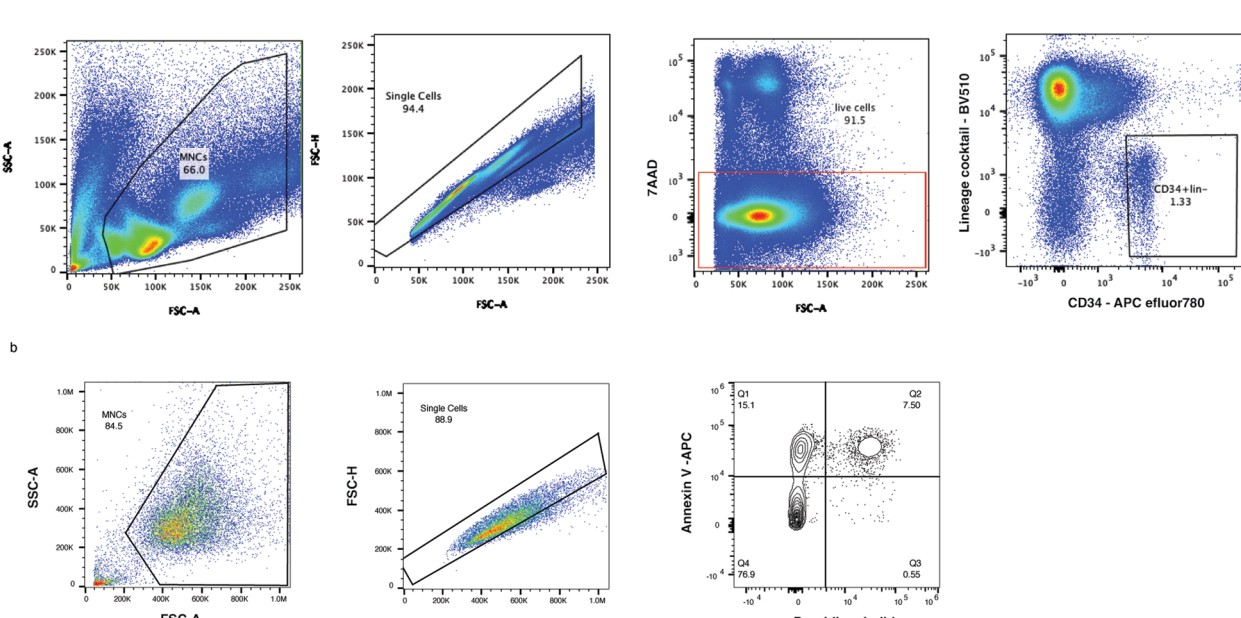

**Extended Data Fig. 8 | Details on FACS gating. a**. Sorting strategy for mini-bulk RNA, ATAC and 10X single cell RNA-seq experiments: Lineage-CD34+ cells were sorted for subsequent library preparation for data shown in Fig. 3, Fig. 4, Fig. 5g, Extended Data Figs. 3 and 7f. **b**. Gating strategy for Annexin V/PI apoptosis staining experiments shown in Fig. 7h and Extended Data Fig. 7g. MNCs: Mononuclear cells, FSC-A: Forward Scatter Area, SSC-A: Side Scatter Area, FSC-H: Forward-Scatter Height, 7-AAD: 7- aminoactinomycin D.

# Reporting Summary

## Statistics

For all statistical analyses, confirm that the following items are present in the figure legend, table legend, main text, or Methods section.

| n/a | Confirmed | |
|---|---|---|
| ☐ | ☒ | The exact sample size (*n*) for each experimental group/condition, given as a discrete number and unit of measurement |
| ☐ | ☒ | A statement on whether measurements were taken from distinct samples or whether the same sample was measured repeatedly |
| ☐ | ☒ | The statistical test(s) used AND whether they are one- or two-sided *Only common tests should be described solely by name; describe more complex techniques in the Methods section.* |
| ☐ | ☒ | A description of all covariates tested |
| ☐ | ☒ | A description of any assumptions or corrections, such as tests of normality and adjustment for multiple comparisons |
| ☐ | ☒ | A full description of the statistical parameters including central tendency (e.g. means) or other basic estimates (e.g. regression coefficient) AND variation (e.g. standard deviation) or associated estimates of uncertainty (e.g. confidence intervals) |
| ☐ | ☒ | For null hypothesis testing, the test statistic (e.g. *F*, *t*, *r*) with confidence intervals, effect sizes, degrees of freedom and *P* value noted *Give P values as exact values whenever suitable.* |
| ☒ | ☐ | For Bayesian analysis, information on the choice of priors and Markov chain Monte Carlo settings |
| ☒ | ☐ | For hierarchical and complex designs, identification of the appropriate level for tests and full reporting of outcomes |
| ☐ | ☒ | Estimates of effect sizes (e.g. Cohen's *d*, Pearson's *r*), indicating how they were calculated |

*Our web collection on statistics for biologists contains articles on many of the points above.*

## Software and code

Policy information about availability of computer code

| | |
|---|---|
| Data collection | - Targeted myeloid-panel sequencing libraries were generated from bulk genomic DNA using a TruSeq Custom Amplicon panel (Illumina) and were sequenced on a MiSeq (Illumina) instrument. |
| | - SNP-Array data was generated through hybridization of bulk genomic DNA to an Illumina Infinium OmniExpress v1.3 BeadChips Array and SNP-CGH CytoScan HD Array |
| | - Whole genome sequencing was performed on bulk genomic DNA isolated from CD3+ depleted cells. Samples underwent PCR-free library preparation prior to 80-100X whole genome sequencing on an Illumina NovaSeq S4 |
| | -- Mini-bulk RNA sequencing was performed on 200 CD34+ lineage negative cells isolated by FACS, with cDNA libraries generated using the Smart-Seq 2 kit prior to pooling followed by sequencing on the NextSeq 500 platform. |
| | -Mini-bulk ATAC-sequencing was performed on 1000 CD34+ lineage negative cells isolated by FACS followed by the Tn5 transposase reaction. cDNA libraries were generated, pooled and sequenced on the NextSeq 500 platform. |
| | -10X single-cell RNA-seq was performed on CD34+ lineage negative cells. Samples were processed according to the 10x protocol using the Chromium Single Cell 30 library and Gel Bead Kits v3.0. |
| | - Flow cytometry data was collected using BD FACS Diva Software (v8.0.2). |
| | - Data collection methods are fully described in the manuscript |
| Data analysis | - Targeted myeloid-panel sequencing data: SOPHiA DDM® (Sophia Genetics) and an in-house software GRIO-Dx®. |
| | - SNP Array data : Mocha WDL pipeline v2021-01-20 (https://software.broadinstitute.org/software/mocha/mocha.20210120.wdl), GISTIC2, |

Chromosome Analysis Suite software package (v4.1, Affymetrix), SHAPEIT v4.1.3.
-WGS: Isabl platform pipeline and interface (https://www.isabl.io/). BWA-mem (v0.7.17) as a part of the pcap-core v2.18.2 wrapper (https://github.com/cancerit/PCAP-core). Mosdepth 4 (https://github.com/brentp/mosdepth). cgpBattenberg (v1.4.0, https://github.com/cancerit/cgpBattenberg). Strelka2 (v2.9.1 with manta v1.3.1, https://github.com/Illumina/strelka), MuTect2 (gatk:v4.0.1.2, https://github.com/broadinstitute/gatk), CaVEMan (cgpCavemanWrapper v1.7.5, https://github.com/cancerit/cgpCaVEManWrapper), cgpCavemanPostprocessing (v1.5.2, https://github.com/cancerit/cgpCaVEManPostProcessing). Pindel (cgpPindel v1.5.4, https://github.com/cancerit/cgpPindel), SvABA (~v1.0.0 commit 47c7a88, https://github.com/walaj/svaba), GRIDSS (v2.2.2, https://github.com/PapenfussLab/gridss), BRASS (v4.0.5 with GRASS v1.1.6, https://github.com/cancerit/BRASS), ClusterSV (v1.0.0, https://github.com/cancerit/ClusterSV/).
- Amplification timing analysis: dpclust3p R package (v1.0.8, https://github.com/Wedge-lab/dpclust3p) and AmplificationTimeR R package (v1.1.1, https://github.com/Wedge-lab/AmplificationTimeR).
- Single cell TARGETSeq data: SingCellaR v1.2.0 was used for data analysis and plotting (https://supatt-lab.github.io/SingCellaR.Doc/index.html). CNA inference was performed using numbat (v1.4.0).
- RNA/ATAC-seq analysis: bcl2fastq v2.20.0.422, FastQC v0.11.5 (https://github.com/s-andrews/FastQC), TrimGalore v0.6.5 (https://github.com/FelixKrueger/TrimGalore), STAR v2.6.1d, Subread (v2.0.0), Bowtie2 v2.4.2, MarkDuplicates module from Picard v2.3.2, Samtools v1.9, ATACseqQC R package (v1.14.4) MACS2 v2.2.7.1, ChIPseeker R package (v1.34.1), TxDb.Hsapiens.UCSC.hg.knownGene (v3.2.2). GenomicRanges (v1.50.2) R package. DESeq2 R package (v1.28.1). Homer (v20201202). FactorMineR (v2.8), factoextra (v1.0.7), ComplexHeatmap R package (v 2.14.0), EnhancedVolcano R package (v1.16.0), pheatmap v1.0.12, GSEA software (Broad Institute; v4.3.2, RRID: SCR_003199).
- 10x Genomics single-cell RNA-seq analysis: bcl2fastq (2.20.0.422) and Cell Ranger software (version 7.0.0) from 10x Genomics. Cite-seq-count/1.4.4 Souporcell pipeline v2.0, troublet v2.4. Seurat v.4.0.1 in R 4.0.4, numbat v1.3.0, pyscenic (v0.10.0) implemented via singularity v3.2
- Synergy analysis: SynergyFinder web application v2 (https://github.com/IanevskiAleksandr/SynergyFinder).
-Genotyping of CRISPR clones: CRIS.py
-Western blot imaging: Odyssey CLx Imaging System (LI-COR).
- Flow cytometry data was analyzed using FlowJo (version 10.7.1.1, BD Biosciences) software.
-`Statistical analyses: R v4.0.4, Prism v7 or later, SPSS v29.0.0.
- Custom codes available at https://github.com/wimm-hscb-lab-published/ upon publication

For manuscripts utilizing custom algorithms or software that are central to the research but not yet described in published literature, software must be made available to editors and reviewers. We strongly encourage code deposition in a community repository (e.g. GitHub). See the Nature Portfolio guidelines for submitting code & software for further information.

## Data

Policy information about availability of data

All manuscripts must include a data availability statement. This statement should provide the following information, where applicable:
- Accession codes, unique identifiers, or web links for publicly available datasets
- A description of any restrictions on data availability
- For clinical datasets or third party data, please ensure that the statement adheres to our policy

All raw and processed sequencing data generated in this study will be made publically available at the NCBI Gene Expression Omnibus (GEO; https://www.ncbi.nlm.nih.gov/geo/) under accession number GSE228060 for CRISPR KO clones, GSE240407 for RNA/ATAC and GSE292030 for single cell primary patient data. The TARGET-seq single cell dataset is available in raw and processed format at GEO accession number GSE226340 and SRA accession number PRJNA930152. The raw and processed SNP array data, and single cell (10X) Seurat object generated in this manuscript is available at Zenodo at the following DOI: 10.5281/zenodo.14749740.

Whole genome sequencing data has been deposited at the European Genome-phenome Archive (EGA), which is hosted by the EBI and the CRG, under accession number EGAS00001007483.
Other publically available datasets accessed:

Tazi et al (Nature Communications, 2022)
BeatAML (Tyner et al, Nature, 2018)
The Cancer Genome Atlas (TCGA) (Ley et al, N Engl J Med, 2013)
The DepMap Cancer Dependency Map ( https://depmap.org/portal/)
The Cancer Cell Line Encyclopedia (https://sites.broadinstitute.org/ccle/)
Custom scripts will be available at https://github.com/wimm-hscb-lab-published/Brierley_NG_chr21amp

# Field-specific reporting

Please select the one below that is the best fit for your research. If you are not sure, read the appropriate sections before making your selection.

☒ Life sciences          ☐ Behavioural & social sciences          ☐ Ecological, evolutionary & environmental sciences

For a reference copy of the document with all sections, see nature.com/documents/nr-reporting-summary-flat.pdf

# Life sciences study design

All studies must disclose on these points even when the disclosure is negative.

Sample size | Sample size was determined based on similar studies in the field and availability of samples.

| Data exclusions | Data which didn't reach quality control parameters (as detailed in Methods section) were excluded from the analysis. |
| --- | --- |
| Replication | In vitro and in vivo experiments in the manuscript were repeated to reach 3 biological replicates in at least 2 independent experiments. Attempts at replication were successful. Details on numbers of replicates are provided in the relevant legend and/or methods section. |
| Randomization | Patients samples were separated according to their diagnosis and chr21amp status, randomization was not appropriate. Mice were allocated randomly to control or KO groups. |
| Blinding | Blinding was not relevant for single cell data, as the information on chr21amp status was required for analysis. For mouse experiments, blinding was performed for analysis of FACS data with an anonymized identification number for each mouse. |

# Reporting for specific materials, systems and methods

We require information from authors about some types of materials, experimental systems and methods used in many studies. Here, indicate whether each material, system or method listed is relevant to your study. If you are not sure if a list item applies to your research, read the appropriate section before selecting a response.

## Materials & experimental systems

| n/a | Involved in the study |
| --- | --- |
| ☐ | ☒ Antibodies |
| ☐ | ☒ Eukaryotic cell lines |
| ☒ | ☐ Palaeontology and archaeology |
| ☐ | ☒ Animals and other organisms |
| ☐ | ☒ Human research participants |
| ☒ | ☐ Clinical data |
| ☒ | ☐ Dual use research of concern |

## Methods

| n/a | Involved in the study |
| --- | --- |
| ☒ | ☐ ChIP-seq |
| ☐ | ☒ Flow cytometry |
| ☒ | ☐ MRI-based neuroimaging |

## Antibodies

| Antibodies used | All antibodies used for the study are detailed in Extended Data Table 12 and 16.<br><br>Antibodies used:<br>CD34-APC efluor780, eBiosciences (Thermo Fisher Scientific), Cat# 47-0349-42 RRID AB_2573956<br>Lineage antibody cocktail (CD3, CD14, CD16, CD19, CD20, CD56)-BV510, Biolegend, Cat# 328122 RRID AB_2561420<br>7AAD, BD Pharminogen, Cat# 51-68981E<br>γ-H2AX Alexa488, Abcam, Cat# ab195188<br>Propidium iodide, eBioscience, Cat# 88-8007-72<br>Annexin V, eBioscience, Cat# 88-8007-72<br>Total-Seq A Hashtag 1, Biolegend, Cat# 394601<br>Total-Seq A Hashtag 2, Biolegend, Cat# 394603<br>Total-Seq A Hashtag 4, Biolegend, Cat# 394607<br>Total-Seq A Hashtag3, Biolegend, Cat# 394605<br>DYRK1A, Abnova, Cat# H00001859-M01<br>HSC70 (HRP/AF680/AF790), Santa Cruz Biotechnology, Cat# sc-7298<br>IRDye® 680RD secondary, LI-COR Biosciences, Cat# 926-68073 & Cat# 926-68072<br>IRDye® 800CW secondary, LI-COR Biosciences, Cat# 926-32213 & Cat# 926-32212<br>beta-Actin, Santa Cruz Biotechnology, Cat# sc-47778 AF680<br>STAT3, Cell Signaling Technology, Cat# 9139S<br>pSTAT3-Y705, Cell Signaling Technology, Cat# 9145S<br>Alexa Fluor® 488 Anti-gamma H2A.X (phospho S139), Adcam, Cat# ab195188<br>LIN52, Invitrogen, Cat# PA5-64882<br>Phospho-S28-LIN52 , Gift from L. Litovchick (www.nature.com/articles/s41388-018-0490-y )<br>FOXO1, Cell Signaling technology , Cat# 2880S<br>GRB2, BD Biosciences, Cat# 610111 |
| --- | --- |
| Validation | Human and mouse antibodies were already validated, titrated and referenced in peer-reviewed publications, as described on the suppliers' websites (Biolegend, eBiosciences, BD horizon, BD Biosciences, BD Pharmingen, Beckman Coulter). Combination of antibodies for human hematopoietic stem cells and progenitors have been already tested in previous publications (Psaila et al, Mol cel 2020, Rodriguez-Meira et al, Nature Genetics 2023). |

## Eukaryotic cell lines

Policy information about cell lines

| Cell line source(s) | American Type Culture Collection (ATCC) for SET2, HEL and HEK293T cell lines |
| --- | --- |

| Authentication | STR testing was performed and confirmed expected identity. |
|---|---|
| Mycoplasma contamination | Cell lines underwent regular mycoplasma testing, which were negative. |
| Commonly misidentified lines (See ICLAC register) | None used. |

## Animals and other organisms

Policy information about studies involving animals; ARRIVE guidelines recommended for reporting animal research

| Laboratory animals | Mouse housing was carried out in individually ventilated cages (19-24°C, humidity 40-65%, 12/12 light dark cycle). Enrichment was done with nesting and bedding material. Mice were fed on standard croquettes, and supplemented with nutritionally complete gel diet after irradiation and in case of weight loss. Mice were maintained on a specific and opportunistic pathogen free health status. NOD SCID Gamma (NSG) mice obtained from The Jackson Laboratory and maintained in St Jude Animal Resource Center, St Jude Children's Hospital, Memphis, TN, USA. Female mice aged 5 weeks. |
|---|---|
| Wild animals | This study did not involve wild animals. |
| Field-collected samples | Study did not involve field-collected samples. |
| Ethics oversight | The animal study was approved by the St. Jude Children's Research Hospital Institutional Animal Care and Use Committee. All experiments were performed at St Jude under animal protocol number 657-100655. |

Note that full information on the approval of the study protocol must also be provided in the manuscript.

## Human research participants

Policy information about studies involving human research participants

| Population characteristics | Patients samples were selected based on their pathology (patients with blast phase myelopoliferative neoplasm). Healthy donor control samples were also used for the study. The gender was not take into account to select the population of interest. Data on individual samples is provide in Extended Data Table 1. |
|---|---|
| Recruitment | Samples were collected as part of patients' routine clinical care through previously established research study approvals as detailed below. |
| Ethics oversight | Peripheral blood and bone marrow samples were collected from BP-MPN patients and healthy donors from the PHAZAR study (Approval REC: 4/WM/1260; IRAS: 163072, 19 Jan 2015), the INForMeD Study (REC: 199833, 26 July 2016, University of Oxford), and the INSERM biobank (approved by the Inserm Institutional Review Board Ethical Committee, project C19-73, agreement 21-794, CODECOH n°DC-2020-4324). Patients and normal donors provided written informed consent in accordance with the Declaration of Helsinki for sample collection and use in research. The current study does not report outcomes of a clinical trial. |

Note that full information on the approval of the study protocol must also be provided in the manuscript.

## Flow Cytometry

### Plots

Confirm that:

☒ The axis labels state the marker and fluorochrome used (e.g. CD4-FITC).

☒ The axis scales are clearly visible. Include numbers along axes only for bottom left plot of group (a 'group' is an analysis of identical markers).

☒ All plots are contour plots with outliers or pseudocolor plots.

☒ A numerical value for number of cells or percentage (with statistics) is provided.

### Methodology

| Sample preparation | Human or cell line samples were stained in PBS + 5% FCS (respectively) with several antibodies, incubated during 20min at RT and washed before being analyzed.<br><br>All methods for sample preparation are fully described in the methods section of the manuscript. |
|---|---|
| Instrument | Cells were analyzed on a BD Fortessa X20 (BD Biosciences) or Attune NxT (Invitrogen, Model AFC2) instrument. Cells were sorted on a BD Fusion I or Fusion II instruments (Becton Dickinson). |
| Software | Analysis of the flow cytometry data was performed using FlowJo (version 10.7.1, BD Biosciences) softwares. |
| Cell population abundance | Human and mouse haematopoietic stem and progenitor (HSPC) populations represent minor cell types (in the majority of cases, less than 1-5% of the total sample), except when they display a competitive advantage in the context of leukemic |

transformation. Sorting was performed in purity mode for bulk experiments  Post-sort purify was checked by sorting 100 cells from selected HSPC fractions (e.g. Lin-CD34+ cells for human experiments) into an eppendorf tube containing 100 uL sorting buffer and analyzing the number of cells included within the same immunophenotype. Post-sort purity was consistently above 95%.

Gating strategy

For HSPC analyses, viable single cells were gated on expression of CD34+Lin- expression using a well-established lineage panel (Psaila et al, Mol Cell, 2020, Rodriguez-Meira et al, Mol Cell, 2019, Rodriguez-Meira et al, Nat Genetics, 2023) For viability analyses, single cells were gated in quadrants by Annexin-V/Propidium Iodide expression as per manufacturer's instruction (https://www.thermofisher.com/order/catalog/product/88-8007-74). Gating strategies are outlined in Extended Data Fig6B,8A &B

☒ Tick this box to confirm that a figure exemplifying the gating strategy is provided in the Supplementary Information.

