## [Peer Review File · Nature Genetics]

Chromothripsis-associated chromosome 21 amplification orchestrates transformation to blast phase MPN through targetable overexpression of DYRK1A

Corresponding Author: Professor Adam Mead

Version 0:

Decision Letter:

6th Nov 2023

Dear Professor Mead,

First, please allow me to apologise for the delay in returning this decision to you. Thank you for your patience.

Your Article, "Chromothripsis orchestrates leukemic transformation in blast phase MPN through targetable amplification of DYRK1A" has now been seen by 2 referees. Our third reviewer did not respond to any queries so we opted to move forward without their feedback. You will see from their comments below that while your two reviewers find your work of interest, some important points are raised. We are interested in the possibility of publishing your study in Nature Genetics, but would like to consider your response to these concerns in the form of a revised manuscript before we make a final decision on publication.

We therefore invite you to revise your manuscript taking into account all reviewer comments. Please highlight all changes in the manuscript text file. At this stage we will need you to upload a copy of the manuscript in MS Word .docx or similar editable format.

Your outstanding reviewer had expertise in DYRK1A biology and depending on your revisions, we will likely recruit someone to cover this aspect of the work in the next round.

*2) If you have not done so already please begin to revise your manuscript so that it conforms to our Article format instructions, available

[here](http://www.nature.com/ng/authors/article_types/index.html).

*3) Include a revised version of any required Reporting Summary: <https://www.nature.com/documents/nr-reporting-summary.pdf>

Link Redacted

We hope to receive your revised manuscript within four to eight weeks. If you cannot send it within this time, please let us know.

Sincerely,

Safia Danovi
Editor
Nature Genetics

Referee expertise:

Referee #1: genome instability

Referee #2: chromothripsis, single cell

Reviewers' Comments:

Reviewer #1:

Remarks to the Author:

Brierley et al report an interesting and important finding related to blast phase of a myeloproliferative neoplasm (BP-MPN). Their work identifies an amplification of a region found within chromosome 21 (chr21amp) in a subset of patients, correlating with poor prognosis. Brierley et al use computational, genomic, and functional assays providing evidence for a minimal amplification region shared among all cases, in which the gene DYRK1A, a serine threonine kinase and transcription factor is located. Using genomic analyses, the authors identify chromothripsis as one route by which chr21amp emerges, thereby providing a link between chromothripsis and a druggable target, DYRK1A. The work provides temporal contexts, showing the putative order of events in BP-MPN, and the occurrence of chr21amp immediately prior to the blast phase. They proceed to show a distinct transcriptional landscape shaped by chr21amp, and then, using functional assays, provide evidence for the role of DYRK1A as a driver of proliferation, and modulator of DNA damage repair. Finally, Brierley et al show an association between BCL-2 and DYRK1A, that can be clinically exploited to potentially benefit patients with chr21amp. Overall, the paper is coherent and interesting to read (this reviewer enjoyed reading the manuscript). It provides a beautiful example of top-down biology, starting from a chromosome-level perturbation and ending with a specific gene and associated cellular circuits.

Major comment:

The claim on chromothripsis as being a major driver of chr21amp is somewhat weak. The importance of chromothripsis to the generation of chr21amp is not much emphasized in this work. Actually the authors provide evidence for 5/64 cases in which chromothripsis might be a relevant driver. In additional 10 cases, chromothripsis does not appear to have a role in chr21amp. Therefore, the authors could consider changing the title, putting more emphasis on the findings related to chr21amp and DYRK1A, rather on chromothripsis itself. This does not, however, affect the view that the findings reported in this work are interesting and important.

Minor comments:

1. What coverage was used when calling for chromothripsis events?
2. Chr21amp appears to form through various mechanisms and not only chromothripsis. This should be emphasized and discussed.

3. Where is the chr21amp located? Intrachromosomal or extrachromosomal? Could the authors use Ampliconarchitect to determine that? Are there samples that could be examined using DNA-FISH?
4. Figure 2A – looks like chromothripsis – it would be helpful to mark the location of DYRK1A. Is it in ecDNA?
5. Figures 2C and 2E – it appears that inversions are dominating with CN changes – could this be BFB driven?
6. Timing the events is very interesting. Is there any data from the supporting AmplificationTimer analysis available?
7. Specificity of DYRK1A is shown nicely.
8. Ext data Fig3: panels D should be E and E should be F on the figure (the legend is fine). In panel F, could the authors mark the differentially accessible peaks?
9. Figure 4C: needs to be better explained. Provide a zoom on chr21. What are the numbers (1, 2, 3) and “n” values?
10. Figure 4 showing chr21amp stalls cells in a progenitor state is striking. Is that what the authors propose?
11. ED fig 5H: Y-axis: title is missing. Times should be indicated above plots.
12. Does fig5G = ED fig 5H? (time 5 days)? If so, this should be indicated. Also, please check the drug names are in the correct order on the x-axis: it appears opposite between the two figures.
13. DYRK1A KD/KO was only done on chr21amp cells in vitro. It could be relevant to examine in control cells as well (myeloid cancer cells without chr21amp for example) perhaps k562 cells could be relevant (supposing it is JAK2 WT and chr21amp negative).
14. What would be the effect of DYRK1A overexpression?
15. What is the impact of DYRK1A inhibitors on DYRK1A expression or function? Since it is a serine/threonine kinase, it would be important to show that its substrates are affected (reduced phosphorylation) in the cells used in the study.
16. Lines 586-590: less clear – please explain better.
17. Figure 6: the message is convoluted. Could the authors please clarify, perhaps using a diagram? Also, raw images of the H2AX staining would be helpful.
18. Lines 635-7: the authors claim that irradiation in KO vs WT cells generate fewer DSB in the KO. But, the amount of DSB formed should be similar, it is the level of repair that is different, right?
19. The cell lines used have a mutation in JAK2 (HEL, SET2). This could confound the interpretation of the involvement of DYRK1A in JAK/STAT signaling. This needs to be clarified.
20. ED Fig 6B: not very convincing. Dose dependent reduction seems more obvious in pSTAT3-Y705 than in the 727 site.
21. Fig 7D: need to better explain the design and system components. The data seems to support the role of DYRK1A in activation of STAT5B, but the involvement of JAK2 mutation is unclear. Also, is this the same mutation seen in HEL and SET2 cells?
22. Increased genomic instability in chr21amp DYRK1A-high cells should result in increased mutations or structural variations relative to BP-MPN cases without such background. Could the authors examine this?
23. DYRK1A cells should tolerate stress better? If so, why do KO cells have better survival under etoposide (Fig 6G) – shouldn't the increased expression of BCL-2 protect the DYRK1A cells?
24. What is the relative contribution of each component to BP-MPN (JAK/STAT, BCL2, DREAM)? Is one more dominant/important? (perhaps address in the discussion)
25. What is the incidence of BP-MPN in Down's syndrome?
26. In the discussion, although DYRK1A is convincingly an important reason why chr21amp is selected, perhaps the authors could discuss other potential genes from this region, that might be relevant.

Reviewer #2:

Remarks to the Author:

The authors present a compelling study highlighting DYRK1A overexpression as a potential driver of MPN progression to the blast phase, in association with recurrent chromosome 21 amplicon (chr21amp). This amplicon can arise from chromothripsis events as well as simpler duplications. Importantly, chr21amp BP-MPN exhibits an aggressive and treatment-resistant phenotype, with potential implications for AML as well. The adverse impact of chr21amp on overall survival remains significant after adjusting for relevant covariates, including age, sex, and high-risk molecular factors such as TP53 mutation status. The authors provide evidence for the druggability of chr21amp, particularly DYRK1A, and its vulnerability to DYRK1A inhibition, further potentiated by BCL2 co-inhibition, unveiling a promising therapeutic avenue for BP-AML. This is a strong and interesting manuscript, which convincingly progresses from integrative genomic analysis in leukemia genomes to comprehensive functional investigations of a target of chromothripsis.

However, several points require clarification and consideration:

1. The manuscript hints at DYRK1A's role in driving genomic instability, primarily through DREAM complex dysregulation. While suggestive, it is important to explicitly state the suggestive nature of this evidence.
2. The recurrent nature and adverse prognosis associated with chromothripsis-associated chr21amp are intriguing findings. However, given the use of SNP arrays as the main basis of their observation, it is imperative to emphasize the inferential aspect behind the chromothripsis calls initially made. SNP arrays generate predictions with respect to chromothripsis. Ultimately, only whole genome sequencing can convincingly verify the occurrence of this rearrangement process in cancer.
3. The manuscript mentions that chr21amp is the most common amplicon in the cohort. It should be clarified in the abstract that this pertains to amplicons, and not all cases harbor chromothripsis events.
4. For cases categorized as 'simple amplicons,' did any of them exhibit structural complexity when examined through WGS? Additional details on breakpoint features, such as microhomology or templated insertions, would be very valuable.

5. Is the measured gene expression within the amplicon allele-specific? This information would provide further insight into DYRK1A's role in chr21amp. Related to this, the manuscript reports that DYRK1A was the only upregulated gene within the minimally amplified region in chr21amp cells compared to non-chr21amp BP-MPN cells. However, extended transcriptomic analyses identified different candidate genes (DSCR3, MORC3, PIGP, TTC3). Could the authors clearly specify the significance thresholds used for these findings? Additionally, do any of these additional candidate genes exhibit evidence of allele-specific expression in the analyzed samples?

6. Among the candidate genes (DYRK1A, DSCR3, MORC3, PIGP, TTC3), which ones are overrepresented in samples from the AML cohort in association with chr21amp?

7. In the experiments presented in Fig 5A-D and Extended Data Fig 5A-F, where DYRK1A knockout and knockdown were assessed in HEL and SET2 BP-MPN cell lines, it would be advisable to include leukemic cell lines lacking chr21amp as controls. Similarly, for the experiments in Fig 5G and Extended Data Fig 5H, which claim selective therapeutic vulnerability in chr21amp patients, control experiments involving BM-MPN patients without the chr21amp event should be included.

8. The manuscript draws a parallel between chr21amp in BP-MPN and iAMP21 in B-ALL. Notably, it was previously documented that the consensus chromothripsis landscape in iAMP21 closely resembles the copy number profile of chromosome 21, as observed across thousands of cancer samples spanning various cancer types. This prior observation suggests a potential pivotal role of chromothripsis in fine-tuning the copy number landscape of chromosome 21. To bolster the manuscript's insights, it would be helpful if the authors could determine whether chr21amp events exhibit a similar alignment with this "consensus cancer copy number profile of chromosome 21"? This could be potentially assessed statistically, such as by permutation testing. Along similar lines, it would be very helpful if discussions centered around potential communalities and differences between chr21amp and iAMP21 could be extended in this manuscript.

Version 1:

Decision Letter:

Our ref: NG-A63264R

16th Oct 2024

Dear Dr Mead,

Thank you for submitting your revised manuscript "Chromosome 21 amplification orchestrates leukemic transformation in blast phase MPN through targetable overexpression of DYRK1A" (NG-A63264R). It has now been seen by the original referees and their comments are below. The reviewers find that the paper has improved in revision, and therefore we'll be happy in principle to publish it in Nature Genetics, pending minor revisions to satisfy the referees' final requests and to comply with our editorial and formatting guidelines.

Sincerely,

Safia Danovi, PhD
Senior Editor, Nature Genetics
ORCID: 0009-0007-7822-5479

Reviewer #1 (Remarks to the Author):

I would like to thank the authors for addressing my comments. I enjoyed reading the revised manuscript, and think this is a well-crafted, beautiful and important work.

Two very small requests - could the authors mark the location of DYRK1A on the plots shown in 2A/C/E? and could they please add a title to the y-axis in 5G?

Reviewer #2 (Remarks to the Author):

The authors have addressed my points very convincingly.

Reviewer #4 (Remarks to the Author):

In this paper, the authors report evidence for chr21 amp and/or chromothripsis, in patients with BP-MPN from a cohort of 64 patients. They uncover recurrent amplification of a region of chromosome 21q (chr21 amp) in 25% of cases, and provide evidence that chromothripsis led to the amplification in about 1/3 of cases in this subset. Although numbers are limited, the chr21 amp BP-MPN had a particularly aggressive and treatment-resistant phenotype. The chr21 amp event appeared to be clonal and present throughout the hematopoietic hierarchy. Using RNA/ATACseq, they found that DYRK1A, a gene encoding a serine threonine kinase and transcription factor, was the only gene in the 2.7Mb minimally amplified region which showed both increased expression and enhanced chromatin accessibility compared to non-chr21 amp BP-MPN controls. Rigorous multiomics studies reveal DYRK1A as a key regulator of multiple cellular functions critical for BP-MPN development, including DNA repair, STAT signaling, and BCL2 overexpression. DYRK1A was found to be essential for BP MPN cell proliferation in vitro and in vivo, and DYRK1A inhibition synergised with BCL2 targeting to induce BP-MPN cell apoptosis. Together, they identify chr21 amp event as a prognostic biomarker in BP-MPN and potentially druggable target.

The paper is a well-written, clear description of chromosomal aberrations that give rise to a new mechanism that drives MPN progression. The studies are well-controlled and rigorous and the conclusions are well-substantiated. Statistical analyses are highly rigorous.

The initial reviews were detailed and extensive, and the the critique was thoroughly addressed. In particular, the incorporation of findings from Down Syndrome AML were of interest.

Overall, this is an exciting and rigorous study.

Response to Reviews: Chromosome 21 amplification orchestrates leukemic transformation in blast phase MPN through targetable overexpression of *DYRK1A*

Reviewer #1:

Remarks to the Author:

Brierley et al report an interesting and important finding related to blast phase of a myeloproliferative neoplasm (BP-MPN). Their work identifies an amplification of a region found within chromosome 21 (chr21amp) in a subset of patients, correlating with poor prognosis. Brierley et al use computational, genomic, and functional assays providing evidence for a minimal amplification region shared among all cases, in which the gene *DYRK1A*, a serine threonine kinase and transcription factor is located. Using genomic analyses, the authors identify chromothripsis as one route by which chr21amp emerges, thereby providing a link between chromothripsis and a druggable target, *DYRK1A*. The work provides temporal contexts, showing the putative order of events in BP-MPN, and the occurrence of chr21amp immediately prior to the blast phase. They proceed to show a distinct transcriptional landscape shaped by chr21amp, and then, using functional assays, provide evidence for the role of *DYRK1A* as a driver of proliferation, and modulator of DNA damage repair. Finally, Brierley et al show an association between *BCL-2* and *DYRK1A*, that can be clinically exploited to potentially benefit patients with chr21amp. Overall, the paper is coherent and interesting to read (this reviewer enjoyed reading the manuscript). It provides a beautiful example of top-down biology, starting from a chromosome-level perturbation and ending with a specific gene and associated cellular circuits.

RESPONSE: We are very grateful to the reviewer for their positive comments.

Major comment:

The claim on chromothripsis as being a major driver of chr21amp is somewhat weak. The importance of chromothripsis to the generation of chr21amp is not much emphasized in this work. Actually the authors provide evidence for 5/64 cases in which chromothripsis might be a relevant driver. In additional 10 cases, chromothripsis does not appear to have a role in chr21amp. Therefore, the authors could consider changing the title, putting more emphasis on the findings related to chr21amp and *DYRK1A*, rather on chromothripsis itself. This does not, however, affect the view that the findings reported in this work are interesting and important.

RESPONSE: Many thanks for this fair comment. It is correct that chromothripsis only occurs in a subset of the amplified cases. We propose to amend the title as follows: "Chromosome 21 amplification orchestrates leukemic transformation in blast phase MPN through targetable overexpression of *DYRK1A*". We have retained a focus on chromothripsis in the abstract and text as we do think that this is an important conceptual aspect of our work. We make clear in the manuscript that a number of different structural variants of varying complexity, including chromothripsis, converge on *DYRK1A* amplification, an actionable molecular target. We would of course be happy to make further changes at the reviewer's and/or editor's request.

Minor comments:

1. What coverage was used when calling for chromothripsis events?

RESPONSE: Chromothripsis was called on whole genome sequencing with a median coverage of 81X [range 77-86]. We now highlight this in the manuscript (lines 221-223) "To determine the precise genetic architecture of the structural variant events that led to chr21 amplification, and confirm that

this is driven by *bona fide* chromothriptic events in some cases using current criteria, *we performed high-depth whole genome sequencing (WGS) in five chr21amp cases, to a median coverage of 81X [range 77-86X]...*”.

2. Chr21amp appears to form through various mechanisms and not only chromothripsis. This should be emphasized and discussed.

RESPONSE: Thank you, we have amended the discussion to reflect this and emphasize that chromothripsis is one of several mechanisms through which DYRK1A is amplified (lines 793-794) “Chr21amp occurs through several mechanisms, which include simple copy number gains, breakage-fusion-bridge cycles and chromothripsis.” In the abstract we have also now specified that chr21amp is “driven by chromothripsis in a third of these cases”.

3 & 4. Where is the chr21amp located? Intrachromosomal or extrachromosomal? Could the authors use Ampliconarchitect to determine that? Are there samples that could be examined using DNA-FISH? Figure 2A – looks like chromothripsis – it would be helpful to mark the location of DYRK1A. is it in ecDNA?

RESPONSE: We believe that the amplification event is intrachromosomal. To further explore this, we have now provided additional analyses for two cases showing chromothripsis of chromosome 21. Firstly, we applied *Decoil*, a computational method to detect and reconstruct complex ecDNA elements, to samples from the patient in **Figure 2A**.¹ This patient demonstrated copy number oscillations between one low (CN=2) and one very high (CN ≥ 10) event and met bioinformatic criteria for a chromothripsis associated double minute/extrachromosomal circular DNA, as defined in the pan-cancer analysis of whole genomes (PCAWG) cohort.² We performed high molecular weight DNA extraction and enrichment for circular DNA structures.³ We subsequently performed long-read nanopore sequencing and applied *Decoil*.^{1,3} On this occasion, this excluded the presence of any extrachromosomal circular DNA.

Additionally, we performed DNA FISH on samples from the patient in **Figure 2E**. This analysis clearly shows that the amplification event is intrachromosomal as shown in the new **Figure 2GH**.

We have added a comment (lines 251-260) that “One of the cases (Patient 1, **Fig 2A**) demonstrated copy number oscillations between one low (CN=2) and one very high (CN ≥ 10) event, possibly representing the presence of chromothripsis associated circular extrachromosomal DNA (ecDNA).² We further investigated for the presence of ecDNA by applying *Decoil*, an ecDNA detection algorithm¹, to long-read sequencing data obtained from Patient 1 after enriching for circular DNA structures (Methods).^{4,5} This definitely excluded the presence of ecDNA. Furthermore, DNA fluorescence in situ hybridisation analysis of a chr21amp sample from Patient 3 with a high-level copy number gain (Fig 2E) confirmed that the amplification event was intrachromosomal (Fig 2GH).²”

5. Figures 2C and 2E – it appears that inversions are dominating with CN changes – could this be BFB driven?

RESPONSE: Thank you for making this point. Yes, indeed, we are convinced that the initiating event underlying chr21amp is one or more breakage-fusion-bridge cycles. This is supported by the fact that the boundaries of the amplified regions are demarcated by fold-back inversion rearrangements, as shown in Figure 2C and 2F. We now make a specific comment relating to this in the result section (lines 250-251 “The presence of foldback loops in the cases profiled is consistent with breakage-fusion-bridge cycles as the initiating event.”

6. Timing the events is very interesting. Is there any data from the supporting AmplificationTimer analysis available?

RESPONSE: AmplificationTimeR is now published at <https://doi.org/10.1093/bioinformatics/btae281>.⁶ The algorithm uses copy number states in combination with information about the multiplicity of mutations within the copy number altered segment to work out the timing of individual gains in pseudotime. In the simplest scenario possible where only one chromosome copy is gained resulting in a copy number state of 2+1 (major allele + minor allele), mutations occurring on the major allele prior to the gain will be present with multiplicity 2 at the time of sampling. Mutations occurring after the gain on the major allele, or at any time during the history of the tumor on the minor allele, will be present with multiplicity 1. If one assumes that mutation rate is constant, one can tally the number of mutations at multiplicity 1 (n_1) and multiplicity 2 (n_2) and use these to calculate the time of the gain. This logic can be extended to higher copy number gains with higher multiplicity states, with the highest possible multiplicity state corresponding to the copy number of the major allele (n_{Maj}).

Patient samples 1-4 (as shown in Fig 2A-F and SFig2A&B) had sufficient mutations within the copy number segment to enable timing analysis. All mutations within the copy number segment that spans DYRK1A are present at the respective n_{Maj} multiplicity. This indicates that all mutations occurring within the copy number segment occurred before the first gain of the segment. The absence of any mutations at n_1 indicates that there was not enough time between the final gain of the segment and the time of sampling to allow for further mutation of the segment, suggesting that this event occurred late in the lifetime of each tumor. In addition, we observe that there were no mutations present at multiplicities between n_1 and n_{Maj} , suggesting that the gains occurred in rapid succession, with no extra mutations introduced in the segment between gains.

Figure 1 below depicts the possible and observed multiplicity states for the segment spanning DYRK1A for each sample. Figure 2 depicts the multiplicities observed in the segment spanning DYRK1A for each sample compared to the multiplicity states observed across the entire genome (including the segment spanning DYRK1A).

Figure 1. Multiplicity states across the gained copy number segment spanning DYRK1A. The x-axis indicates the potential multiplicity states that would be expected to be observed for the segment spanning DYRK1A in each sample given its copy number state. Shaded bars indicate the number of mutations observed in each multiplicity state.

Figure 2. Multiplicity states observed in segments spanning DYRK1A compared to the entire genome. Red shaded regions indicate the number of mutations in each multiplicity state identified in regions spanning DYRK1A for each sample. Grey shaded regions indicate the number of mutations at each multiplicity state across the entire genome, including the segment spanning DYRK1A. The red and grey bars overlap, rather than being additive.

We have added this description and explanation of the method in the methods section (lines 1360-1379) and also added the Figures as additional supplemental figure panels in extended data Fig 2 panels E and F.

7. Specificity of DYRK1A is shown nicely.

RESPONSE: Thank you.

8. Ext data Fig3: panels D should be E and E should be F on the figure (the legend is fine). In panel F, could the authors mark the differentially accessible peaks?

RESPONSE: Apologies for this oversight and thank you for identifying this mistake. The panels in this figure have shifted in response to reviewer comments. We have amended and updated the labelling, marked the differentially accessible peaks and corrected the legend accordingly.

9. Figure 4C: needs to be better explained. Provide a zoom on chr21. What are the numbers (1, 2, 3) and “n” values?

RESPONSE: We agree that this figure (the automated output from the numbat haplotype-aware copy number calling pipeline) is not entirely intuitive. Figure (4C) represents pseudobulking of clones from single cell RNA-seq from a single patient, where the clones are defined by copy number status. The n refers to the number of cells per clone, and for each clone there are two sections in the panel: The top panel shows the log₂-fold change of normalized copy number (ranging from -2 to +2, represented by the numbers on the y axis) and the bottom panel the parental haplotype frequency (ranging from 0-1 on the y axis) for each SNP across the chromosome. We have clarified the legend, and incorporated a “zoom” window on chr21 as requested. The revised figure and legend are shown below:

Revised figure legend:

C. Clone-specific pseudobulk profile for a representative patient showing detection of the chr21amp event in single cells by the copy-number calling software *numbat*. Each of the three plot subpanels defines a copy-number defined clone ascertained by *numbat*, with the chromosomal location denoted along the x axis. Each subpanel contains two sections; the top section shows the log₂-fold change (logFC) of normalized copy number (ranging from -2 to +2, represented by the numbers on the y axis) and the bottom panel the parental haplotype frequency (pHF) (ranging from 0-1 on the y axis), inferred from haplotype phasing of SNPs genotyped from single cell transcriptomes. CNV calls are colored by type of alteration (amplification in red, deletion in blue, copy-neutral loss of heterozygosity in green). The red zoom boxes highlight the chr21amp event.

10. Figure 4 showing chr21amp stalls cells in a progenitor state is striking. Is that what the authors propose?

RESPONSE: The reviewer is correct, this is exactly what we believe these data show i.e. that the bulk of the leukemic cells carry the chr21 amplification event and are arrested in an immature progenitor cell state with very little erythroid differentiation, consistent with a differentiation block (quantified in the barplots in Figures 4F & G). We have further clarified this in the text of the results section (lines 441-444) “Chr21amp cells were notably less frequent in late erythroid precursors, while non-chr21amp cells were exclusively detected in the late erythroid precursor cluster, implying presence of a differentiation block with leukemic cells carrying the chr21amp event frequently stalled in a progenitor state (Fig 4F & G).”

11. ED fig 5H: Y-axis: title is missing. Times should be indicated above plots.

RESPONSE: Apologies for these oversights and thank you for highlighting them; we have made the corrections accordingly. This figure has also been amended in response to Reviewer 2’s comments (point 7), with further non-chr21amp controls added to the experimental setup, crucially showing that the effect of DYRK1A inhibition is selective for leukemia cases with chr21amp.

12. Does fig5G = ED fig 5H? (time 5 days)? If so, this should be indicated. Also, please check the drug names are in the correct order on the x-axis: it appears opposite between the two figures.

RESPONSE: Yes, Fig 5G=ED Fig 5H – we chose to show these data again in the context of the timecourse in the supplementary data. We have now highlighted this intentional duplication in the legend and corrected the x-axis.

13. DYRK1A KD/KO was only done on chr21amp cells in vitro. It could be relevant to examine in control cells as well (myeloid cancer cells without chr21amp for example) perhaps k562 cells could be relevant (supposing it is JAK2 WT and chr21amp negative).

RESPONSE: Publicly available CRISPR screen data from the Broad DepMap database supports that overexpression of DYRK1A correlates with dependency as shown in the current Extended Data Figure 4G. DepMap data also shows a significant correlate between DYRK1A copy number and gene dependency (higher DYRK1A copy number correlates with increased DYRK1A dependency). We have specifically labelled K562 in Extended Data Figure 4G which shows low DYRK1A expression and no dependency on DYRK1A. A previous study reporting the results of a kinase domain focused CRISPR screen also supports selective dependency of HEL and SET2 cells on DYRK1A in comparison with non-DYRK1A amplified cell lines such as K562 and OCI AML3.⁷ We have added an additional comment on lines 528-530 of the results “However, myeloid leukemia cell lines with low DYRK1A expression did not show dependency on DYRK1A e.g. K562 (Extended Data Fig 4G), in line with a previous study.” We have also carried out an additional CRISPR knockout of DYRK1A using the same guides as used for SET2 and HEL cells for the non-Chromosome 21 amplified cell line OCI AML3. As shown below, the OCI AML3 cell line did not show any evidence of selection over time for edits causing a knockout of DYRK1A (also consistent with OCI AML3 data in DepMap). We have not included these data in the revised manuscript but would be happy to do so at the reviewer’s request.

We would also like to draw the reviewer's attention to new data with regards to primary BPMPN samples (in response to a suggestion from reviewer 2, point 7). We have now updated figure 5G to include additional cytotoxicity experiments with n=5 primary BPMPN non-chr21amp controls. These important new data show that the cytotoxic effect of DYRK1A inhibition is selective for leukemia cases with chr21amp. These data are shown in Fig 5G and Extended Data Fig 5I and are described in the results section on lines 567-569.

14. What would be the effect of DYRK1A overexpression?

RESPONSE: Thank you for this important question. To investigate the impact of DYRK1A overexpression in cell lines that are not *DYRK1A* dependent at baseline. We transiently overexpressed *DYRK1A* in Hek293T cell lines, and show that overexpression of *DYRK1A* directly upregulated STAT5B transcriptional activity in a reporter assay (Figure 7DE). We have updated the results section to emphasize this point (715-731) "These data support that an important effect of DYRK1A overexpression is to activate STAT5 activity, further amplifying activation of JAK/STAT signalling which is a cardinal feature of MPN in chronic phase."

15. What is the impact of DYRK1A inhibitors on DYRK1A expression or function? Since it is a serine/threonine kinase, it would be important to show that its substrates are affected (reduced phosphorylation) in the cells used in the study.

RESPONSE: There are several well-substantiated targets of DYRK1A serine threonine kinase activity in the literature, including phosphorylation of FOXO1 and LIN52.^{8,9} To confirm on-target efficacy of DYRK1A inhibitors in the BP-MPN cell line context, we treated SET2 cell line with the DYRK1A inhibitor EHT1610 and performed Western blots at 4 hours post treatment. These data confirm clear impact on these known DYRK1A substrates and these new data are now shown in Extended Data Figure 5G and are described on lines 552-556 of the results section "We next explored whether pharmacological inhibition of DYRK1A using the small molecule inhibitors GNF2133 and EHT1610 would have the same impact. We first confirmed that phosphorylation of the known DYRK1A substrates LIN52 and FOXO1 was reduced following EHT1610 treatment of SET2 cells (Extended Data Fig 5G" We have also added a description of the methods on lines 1699-1709).

16. Lines 586-590: less clear – please explain better.

RESPONSE: Apologies – we agree that this was a convoluted sentence and could have been phrased more clearly. We have reworded the entire paragraph for clarity. Please find the reworded statement below (lines 633-650):

“We hypothesized that DYRK1A overexpression in BP-MPN may activate the DREAM complex. Activation of DREAM leads to the transcriptional suppression of associated DNA repair pathways, consequently reducing DNA damage response thus increasing genomic instability (Extended Data Fig 6A for schema). ChIPseq experiments have identified a DREAM-controlled DNA repair geneset.¹⁰ We performed geneset enrichment analyses for the expression of this geneset across three datasets: chr21amp primary patient BP-MPN RNA-seq, DYRK1A CRISPR KO cell line RNA-seq, and the BEAT AML high DYRK1A expressing patient cohort. Our hypothesis was that when DYRK1A expression is high, this would stabilise DREAM and repress expression of the geneset (Extended Data Fig 6A for schema).

Indeed, in primary patient chr21amp BP-MPN cells vs controls, as well as in BEAT AML top DYRK1A expressors vs bottom, the DREAM DNA repair geneset was downregulated (**Fig 6A-D**, NES -1.74, FWER p-value 0.01 for chr21amp vs non, NES -2.13, FWER p-value <0.001 for BEAT AML top DYRK1A expressors vs bottom). Conversely, DYRK1A CRISPR KO SET2 cells showed significant upregulation of DREAM complex target genes (**Fig 6E and F**, NES 1.76, FWER p-value <0.001). These data support our hypothesis that DYRK1A overexpression leads to increased DREAM complex mediated transcriptional repression of DNA repair pathways in BP-MPN cells.”

17. Figure 6: the message is convoluted. Could the authors please clarify, perhaps using a diagram? Also, raw images of the H2AX staining would be helpful.

RESPONSE: We agree that this is somewhat convoluted and likely best demonstrated with a schematic diagram. We have inserted the following schematic into ED Fig 6A and incorporated into the text as below (also outlined in response 16 above and response 18 below) (lines 639-641). “Our hypothesis was that when DYRK1A expression is high, this would stabilise DREAM and repress expression of the geneset (Figure 6A for schema).”

The γ -H2AX assay was performed by an established flow cytometry assay.¹¹ We have now added representative γ -H2AX FACS plots in Extended Data Fig 6B.

18. Lines 635-7: the authors claim that irradiation in KO vs WT cells generate fewer DSB in the KO. But, the amount of DSB formed should be similar, it is the level of repair that is different, right?

RESPONSE: Thank you for highlighting this important point, you are correct that this was not optimally phrased. We agree that the number of DSBs formed should be similar as it is the kinetics of repair that differ. We have reworded the relevant section of the results accordingly (lines 656-663) “This finding was supported by DYRK1A KO leading to a reduction in detectable double-stranded DNA breaks as ascertained by γ -H2AX staining after 8 hour treatment with 3 mM etoposide in

DYRK1A CRISPR KO SET2 cells compared to wild type (**Fig 6H**). Consistent with this, induction of DNA damage by irradiation in DYRK1A KO vs WT SET2 cells led to fewer detectable double-stranded DNA breaks at 8 hours in the KO than wild type (**Fig 6I**), which we infer may be due to enhanced kinetics of repair.”

We would also like to highlight a recent paper (Leukemia volume 38, pages 521–529, 2024; DOI: 10.1038/s41375-024-02151-8)¹² which explores DYRK1A in Down Syndrome associated myeloid malignancies and concludes that “Increased dosage of the chromosome 21 (chr21) gene *DYRK1A* impairs homology-directed DNA repair as a mechanism of elevated mutagenesis.” They use a reporter system to test homology-directed repair (HDR) and conclude that there is deficiency in HDR in the trisomy 21 cells (Fig4D). Further, they show that overexpression of DYRK1A directly leads to defective HDR in a dose-specific manner (Fig4E-F). Conversely, KO of DYRK1A show a higher level of HDR (Suppl. Fig4K). Taken together these results strongly support our model and we now include a description of this recent paper in the discussion on lines 850-855 “A recent study in the context of Down Syndrome associated myeloid malignancies used a homology directed repair reporter system to demonstrate that increased expression of DYRK1A leads to impaired homology-directed DNA repair as a mechanism of elevated mutagenesis, providing additional support for our proposed model in BPMPN”.

19. The cell lines used have a mutation in JAK2 (HEL, SET2). This could confound the interpretation of the involvement of DYRK1A in JAK/STAT signalling. This needs to be clarified.

RESPONSE: Thank you for making this point. It is correct that both HEL and SET2 are *JAK2* mutant (we state this in the results section lines 531), as indeed are the majority (9/16, 56%) of chr21amp patients. We argue that *DYRK1A* amplification further activates JAK/STAT signalling in a feed-forward loop, and provide several lines of evidence for this. On geneset enrichment analysis (Figure 7A), we compare SET2 *DYRK1A* knock-out (*JAK2* mutant) with SET2 wild type (also *JAK2* mutant), and detect a significant downregulation of the HALLMARK STAT5 target geneset (NES -2.08 FWER p-value 0.001).

In a luciferase assay of STAT5B activity, we transfect Hek293T cells (non *JAK2* mutant/JAK/STAT signaling dependent) with either the mutant or wild type *JAK2* allele, and then overexpress *DYRK1A* (Figures 7D & E). Even in the absence of *JAK2* mutation (figure 7D), *DYRK1A* overexpression activates STAT5B transcription. In the presence of *JAK2V617F*, *DYRK1A* further potentiates STAT5B transcriptional activity (Figure 7E).

The observed synergy and potentiation between *DYRK1A* overexpression and basal JAK/STAT upregulation is supported by an observation in a previously published CRISPR screen of kinases in AML cell lines⁷ (Tarumoto et al, *Molecular Cell* 2018, PMID 29526696), which identified that those harboring *JAK2* mutations are uniquely sensitive to *DYRK1A* targeting. We highlight in the discussion (lines 883-886) that this further potentiation of an oncogenic pathway by CNAs already activated by mutation has also been demonstrated in other tumor settings – for example, in BRAF mutant solid cancers the acquisition of further copy number alterations can amplify and potentiate the same MAP kinase signaling pathway.

We have added a specific comment in the results section relating to the luciferase assay (728-731) “These data support that an important effect of *DYRK1A* overexpression is to activate STAT5 activity, further amplifying activation of JAK/STAT signalling which is a cardinal feature of MPN in chronic phase.”

20. ED Fig 6B: not very convincing. Dose dependent reduction seems more obvious in pSTAT3-Y705 than in the 727 site.

Thank you for your comment. We agree with the reviewer comment that STAT3-Tyr705 reduction is more convincing. This residue has been directly linked to JAK2 activation and several papers have shown the link between decrease STAT3-Tyr705 phosphorylation and JAK2 downregulation/inhibition.¹³⁻¹⁵ In our experiments we were able to show the link between DYRK1A inhibition and loss of STAT3-Tyr705 phosphorylation in SET2 and HEL cell lines (Extended data Fig 7B). We have revised the text accordingly (lines 708-714) stating “DYRK1A and JAK2 have both been shown to activate STAT3 at residue Tyr705.^{8,13-16} We investigated whether DYRK1A inhibition reduces STAT3 phosphorylation in BP-MPN cell lines, explaining the enrichment for the STAT3 gene set in chr21 amp BP-MPN. In line with prior observations, STAT3 Tyr705 phosphorylation occurred in both HEL and SET2 BP-MPN cell lines, and DYRK1A inhibition led to a dose-dependent reduction in STAT3-Tyr705 phosphorylation. (Extended Data Fig 7B)”

We have removed the image of the pSTAT3-727 as we agree that this is less evident in the Western blot and less substantiated in the current literature

21. Fig 7D: need to better explain the design and system components. The data seems to support the role of DYRK1A in activation of STAT5B, but the involvement of JAK2 mutation is unclear. Also, is this the same mutation seen in HEL and SET2 cells?

RESPONSE: We apologise for lack of clarity in the original manuscript, and agree that the complex experimental system requires a clearer explanation for readers.

The experiment is a dual luciferase transfection assay in HEK293T cells, which have the advantage of being readily transfected, diploid (with no endogenous upregulation of chromosome 21), *JAK2* wild type, and little endogenous JAKSTAT or TPO signaling. In these assays, STAT5B transcriptional activity is read out by the firefly luciferase reporter, while pRL-TK-driven renilla luciferase serves as an internal control. STAT5B luciferase activity is upregulated in this cell lines by either transfecting the *JAK2V617F* mutation, or by transfecting *STAT5B* in its constitutively active form. Additionally, the wild type TPOR is transfected in this assay as HEK cells express low levels of endogenous TPOR. TPO is then added to the system to stimulate JAKSTAT signaling through binding to the TPO-receptor. This is a widely used system for studying JAK/STAT signaling in the context of MPN.¹⁷⁻²⁰

The experimental aim was to demonstrate the impact of *DYRK1A* overexpression (versus empty vector control) on STAT5B transcriptional activity in the setting of either wild-type or mutant JAK2. To simplify the presentation of these data we have removed panel 7E so that we now just show 2 panels using this assay, 7D focused on wild-type JAK2 and 7E on mutant JAK2.

In panel 7D we show that in the presence of wild-type JAK2, overexpression of *DYRK1A* leads to upregulation of STAT5B transcription in a TPO-dependent manner, when coexpressed with constitutively active STAT5B. In panel 7E we show that in the presence of *JAK2V617F*, *DYRK1A* overexpression increases STAT5B transcriptional activity in a TPO-independent manner, an effect that is amplified by overexpression of wild-type STAT5B.

We have amended the results (lines 715-731) and the figure legend for Figure 7 for clarity.

22. Increased genomic instability in chr21amp *DYRK1A*-high cells should result in increased

mutations or structural variations relative to BP-MPN cases without such background. Could the authors examine this?

RESPONSE: Thank you for raising this relevant point. We were able to examine this within our cohort by comparing the number of copy number abnormalities in BP-MPN cases with chr21amp versus those without. We excluded any chromosome 21 associated copy number abnormalities from this comparison. Chr21amp BP-MPN cases have a higher number of non-chr21 copy number abnormalities than non-chr21amp cases (median= 6.5(IQR 4-10.3) vs median 1(IQR 1-5), $p<0.001$ by Wilcoxon rank-sum test). We highlight this in the manuscript and have moved the associated figure into the main figure (Figure 1F) described in the results section (lines 179-181).

23. DYRK1A cells should tolerate stress better? If so, why do KO cells have better survival under etoposide (Fig 6G) – shouldn't the increased expression of BCL-2 protect the DYRK1A cells?

Our hypothesis is that DYRK1A overexpression leads to increased DNA damage accumulation and genomic instability due to reduced DNA repair. Consequently, since DNA damage is more efficiently repaired in DYRK1A KO cells (Fig 6H and 6I), they show less sensitivity to the DNA-damage inducing agent etoposide than DYRK1A amplified counterparts (Fig 6G). We reasoned that treatments that are not dependent on inducing DNA damage should not show a similar effect. Consistent with this, as shown in Figure 7, DYRK1A inhibition shows synergy with BCL2 inhibition (a non DNA damage inducing agent). To test this further, we used shRNAs to genetically knock down DYRK1A expression in HEL cells in combination with either etoposide or two additional drugs that are not dependent on inducing DNA damage. We used ruxolitinib, which targets JAK2 activation and vincristine, which targets mitotic spindle formation, at two concentrations. We were able to reproduce the relative increase in cell survival when HEL cells are transfected with DYRK1A-targeting shRNAs in combination with a range of etoposide concentrations compared to control shRNA, corroborating our data in SET2 DYRK1A KO cells. Genetic knockdown of DYRK1A did not confer an advantage in cell survival when cells were treated with either ruxolitinib or vincristine, supporting a specific link between DYRK1A overexpression and aberrant DNA damage response. We have not included these additional data in the revised manuscript due to space constraints, but would be happy to do so at the reviewer's request.

REVIEWER FIGURE: HEL cells were transduced with lentiviruses targeting either scramble control or two distinct shRNAs targeting *DYRK1A* at 15MOI. Cells were washed and media changed at 24 hours. At 48 hours, cells were plated in a 96 well plate at 20,000 cells per well and treated with serial dilutions of either etoposide (ETO), vincristine (VINC) or ruxolitinib (RUX) in triplicate, with DMSO controls for each drug concentration. Cells were treated 4 hours after plating and then placed in the IncuCyte® Live Cell Imager system (Essen BioSciences, Inc., Ann Arbor, MI, USA). Images were captured in the IncuCyte ZOOM™ platform (Essen BioSciences, Inc.). Nine image sets were acquired from several points of the well, using a 10× objective lens. Viability was assessed at 3.5 days from plating and is shown relative to DMSO control for etoposide in (A), vincristine in (B) and ruxolitinib in (C).

24. What is the relative contribution of each component to BP-MPN (JAK/STAT, BCL2, DREAM)? Is one more dominant/important? (perhaps address in the discussion)

RESPONSE: This is an interesting point to speculate on, and we have expanded the discussion (lines 887-897) to elaborate on this: “As *DYRK1A* overexpression is orchestrating multiple cellular processes to promote disease progression in MPN, it is interesting to speculate which component (amplified JAK/STAT signaling versus increased genomic instability) is dominant. In our view, the

strong synergy between presence of p53 mutation and chr21amp, together with the striking increase in non-chr21amp CNAs in cases with *DYRK1A* amplification support that the impact on DNA repair is critical. The lack of durable responses to JAK2 inhibition in BPMPN also support that inhibition of amplified JAK/STAT signalling is insufficient alone to ameliorate the disease.²¹⁻²³ We speculate that JAK2 mutation provides ‘fertile ground’ for the acquisition of chr21amp, but once acquired the disease evolution is primarily driven by *DYRK1A* overexpression associated genomic instability.”

25. What is the incidence of BP-MPN in Down’s syndrome?

RESPONSE: Thank you for this comment. To our knowledge, there is no increased incidence of MPN in Down’s syndrome cases. However, there is a well-recognised phenomenon of transient abnormal myelopoiesis (TAM) in Down’s syndrome, of which a subset progress to myeloid leukaemia in Down Syndrome (ML-DS). The increased leukemic risk is estimated at 10-to-20-fold and is associated with a worse outcome than non-Down syndrome children with leukemia. While trisomy 21 is critical to this process, what the contribution of specific genes on chromosome 21 is to this phenomenon is not well-elucidated. However, *DYRK1A* lies in the centre of the Down syndrome critical region. The DS critical region was derived from decades of work investigating rare cases with partial trisomy 21, where genetic duplication of only a limited region of chr21 occurs but with full penetrance of the Down syndrome phenotype.²⁴ A role for *DYRK1A* in driving genomic instability in the DS context has been suggested, where HSPCs derived from human induced pluripotent stem cells with trisomy 21 demonstrated a greater accumulation of CNVs associated with upregulation of *DYRK1A* and a downregulation of DNA repair mechanisms.²⁵ It is also interesting to note that genetic analysis of ML-DS cases shows a very high prevalence (48%) of mutations leading to activation of JAK family kinases.²⁶ This further supports a direct synergy between amplified *DYRK1A* and other pathways leading to activation of JAK/STAT signalling.

We have amended the discussion to include some discussion relating to the role of *DYRK1A* in Down Syndrome on lines 850-855 and 868-871 to expand on this point this point “A recent study in the context of Down Syndrome associated myeloid malignancies used a homology directed repair reporter system to demonstrate that increased expression of *DYRK1A*, which lies in the centre if the Down Syndrome critical region, leads to impaired homology-directed DNA repair as a mechanism of elevated mutagenesis, providing additional support for our proposed model in BPMPN..... It is also interesting to note that genetic analysis of ML-DS cases shows a very high prevalence (48%) of mutations leading to activation of JAK family kinases as well as increased chromosomal CNAs, a finding recently linked to *DYRK1A* overexpression.²⁶”

26. In the discussion, although *DYRK1A* is convincingly an important reason why chr21amp is selected, perhaps the authors could discuss other potential genes from this region, that might be relevant.

RESPONSE: *DYRK1A* was the only gene both differentially expressed and differentially accessible on RNA-and ATAC-seq. However, on single cell analyses of a separate cohort of chr21amp cases, four other genes were also differentially expressed between chr21amp cells and controls (Supplementary Fig 4 A-D). These genes were *PIGP*, *TTC3*, *MORC3* and *DSCR3*. None of these genes show dependency in BPMPN cell lines and they have not previously been implicated in leukemogenesis. We have added some additional discussion on lines 905-912 to acknowledge that it remains possible that these or other additional genes in the minimally amplified region might contribute to the impact of chr21amp and we did not explore their functional role “Although *DYRK1A* was the only gene in the minimally amplified region that was both differentially expressed and differentially accessible in chr21amp BPMPN, it is also important to note that other genes in the minimally amplified region

were differentially expressed (PIGP, TTC3, MORC3 and DSCR3). None of these genes show dependency in BPMPN cell lines and they have not previously been implicated in leukemogenesis, we did not explore their functional role and it remains possible that they might act in concert with DYRK1A overexpression.”

Reviewer #2:

Remarks to the Author:

The authors present a compelling study highlighting DYRK1A overexpression as a potential driver of MPN progression to the blast phase, in association with recurrent chromosome 21 amplicon (chr21amp). This amplicon can arise from chromothripsis events as well as simpler duplications. Importantly, chr21amp BP-MPN exhibits an aggressive and treatment-resistant phenotype, with potential implications for AML as well. The adverse impact of chr21amp on overall survival remains significant after adjusting for relevant covariates, including age, sex, and high-risk molecular factors such as TP53 mutation status. The authors provide evidence for the druggability of chr21amp, particularly DYRK1A, and its vulnerability to DYRK1A inhibition, further potentiated by BCL2 co-inhibition, unveiling a promising therapeutic avenue for BP-AML. This is a strong and interesting manuscript, which convincingly progresses from integrative genomic analysis in leukemia genomes to comprehensive functional investigations of a target of chromothripsis.

RESPONSE: We are very grateful to the reviewer for their positive comments.

However, several points require clarification and consideration:

1. The manuscript hints at DYRK1A's role in driving genomic instability, primarily through DREAM complex dysregulation. While suggestive, it is important to explicitly state the suggestive nature of this evidence.

RESPONSE: We have reworded key paragraphs as summarised below to emphasize the suggestive nature of the mechanism described. We have also included a limitations paragraph which explicitly states the suggestive nature. Please find the reworded sections below:

Lines 664-667 “Taken together, these data support that chr21amp induced DYRK1A overexpression leads to suppression of DNA repair through aberrant DREAM complex activity. This might lead to increased genetic instability, in keeping with the increased number of CNAs we observed in chr21amp BP-MPN cases “

Lines 847-855 “Here, we propose that two biological pathways activated by DYRK1A are crucial in its oncogenic activity when upregulated in chr21amp BP-MPN. First, we suggest that the chr21amp event causes downregulation of DNA repair pathways to promote genomic instability. [...]“

Lines 903-905 “[...]while we were able to identify a gene signature suggesting a role for the DREAM complex in mediating genomic instability and confirm altered DNA repair in chr21amp patients, the link between the two is suggestive and further study is required to confirm this mechanistically.“

2. The recurrent nature and adverse prognosis associated with chromothripsis-associated chr21amp are intriguing findings. However, given the use of SNP arrays as the main basis of their observation, it is imperative to emphasize the inferential aspect behind the chromothripsis calls initially made. SNP arrays generate predictions with respect to chromothripsis. Ultimately, only whole genome sequencing can convincingly verify the occurrence of this rearrangement process in cancer.

RESPONSE: Thank you for highlighting this important point. We are hopeful that our amended limitations paragraph can fully address this valid point.

Lines 898-902 “Limitations of our study include that we used SNP arrays rather than WGS to call the initial incidence of chr21amp and chromothripsis, and performed WGS in a smaller selected cohort to validate and extend these findings. SNP arrays enabled us to screen for and infer chromothripsis, but only WGS has the required resolution to confirm its occurrence.”

3. The manuscript mentions that chr21amp is the most common amplicon in the cohort. It should be clarified in the abstract that this pertains to amplicons, and not all cases harbor chromothripsis events.

RESPONSE: Thank you for highlighting this point. This was also highlighted by Reviewer 1 (major comment and minor comments, point 2). It correct that chromothripsis only occurs in a subset of the amplified cases. We have amended the title to reflect “Chromosome 21 amplification orchestrates leukemic transformation in blast phase MPN through targetable overexpression of DYRK1A”, and we have reworded the abstract to emphasize that chr21amp is not caused by chromothripsis in all cases “In a cohort of 64 patients in blast phase of a myeloproliferative neoplasm (BP-MPN), we describe recurrent amplification of a region of chromosome 21q (‘chr21amp’) in 25%, driven by chromothripsis in a third of these cases.” We also amended the discussion to reflect this and emphasize that chromothripsis is one of several mechanisms through which DYRK1A is amplified (lines 798-799) “Chr21amp occurs through several mechanisms, which include simple copy number gains, breakage-fusion-bridge cycles and chromothripsis.” In the abstract we have also now specified that Chr21amp is “driven by chromothripsis in a third of these cases”. Nevertheless, we would like to retain a focus on chromothripsis in the abstract and text as we do think that this is an important conceptual aspect of our work. A number of different structural variants, including chromothripsis, converge on DYRK1A amplification, an actionable molecular target. We would of course be happy to make further changes at the reviewer’s and/or editor’s request.

4. For cases categorized as 'simple amplicons,' did any of them exhibit structural complexity when examined through WGS? Additional details on breakpoint features, such as microhomology or templated insertions, would be very valuable.

RESPONSE: To address the question, we deployed a structural variant (SV) clustering pipeline (ClusterSV; <https://github.com/cancerit/ClusterSV/>) to group rearrangement clusters and identify complex events. ClusterSV takes into consideration the total number and orientation of SVs in a sample, grouping rearrangements that occur in close chromosomal proximity and are unlikely to have occurred by chance. The genetic proximity and occurrence of specific SVs suggest that they arise from the same biological processes. SV rearrangements groups were then classified as simple or complex genomic events, as described by Li Y. et al.²⁷ In brief, some clusters contain single or <3 SV events, often of the same type, and are considered ‘simple’ SV clusters, while others contain >=3 interconnected SVs of varying types and are considered ‘complex’ events, such as chromothripsis, chromoplexy, templated insertions etc.

Of the cases classified as ‘simple amplicons’ on SNP karyotyping, two had undergone WGS. These were Patient 4 and 5, for whom the integrated SV and CN plots for chromosome 21 are shown in Supp Fig 2 A & C. For Pt 4 (Supp Fig 2A), there were no complex events identified using the clusterSV pipeline. However, as shown in the SV/CN plot in Supp Fig 2A, there was an amplification event demarcated by a fold-back inversion rearrangement, features in keeping with a breakage fusion bridge cycle. This was not detectable on SNP karyotyping and highlights the role for WGS in extending the SNP karyotyping findings.

For Pt 5 (Supp Fig 2C), clusterSV did indeed identify a complex event on chromosome 21, which involved deletions, duplications and inversions. Of note, this event was chromothripsis-like, meeting 3 of 4 chromothripsis criteria defined in Li *et al.*²⁷ These criteria are cluster size, fragment join, copy number oscillation and interleaved events, and this particular chr21 event met all criteria bar the number of interleaved events required.

To investigate breakpoint features, we deployed clusterSV to investigate for evidence of templated insertions at breakpoints, and none were identified in any of the cases. Additionally, we modified the approach detailed in Cortes-Ciriano *et al.*²⁸, (originally described in Yang *et al.*²⁹) to inspect breakpoint junctions. Each breakpoint was reviewed in IGV and discordant read pairs and soft-clipped reads overlapping breakpoint junctions were locally aligned to identify putative signatures of repair mechanisms. The classification criteria were as in Cortes-Ciriano *et al.* (originally adapted from Kidd *et al.*³⁰), whereby the sequence was interrogated for features of transposable element insertion (TEI), variable number of tandem repeats (VNTR), nonhomologous end joining (NHEJ), alternative end joining (alt-EJ), nonallelic homologous recombination (NAHR), and fork stalling and template switching/microhomology-mediated break induced repair (FoSTeS/MMBIR). A median of 5 (range 5-11) chromosome 21 breakpoints were examined per case. Breakpoints were frequently (25/31, 81%) characterised by none or small (0-6bp) insertions, consistent with NHEJ or alt-EJ. In 3 of the 31 breakpoints reviewed, there was a small stretch of microhomology, potentially in keeping with microhomology-mediated break-induced replication (MMBIR). A further 3 of 31 were in repetitive regions and not feasible to classify using our short read sequencing data. Of note, this approach is known to heavily rely on inference and here is further limited by the fact that we had tumor-only whole genome sequencing data to interrogate, with no matched normal sample. A recent publication (Hu *et al.*, Nature 2024) deployed an elegant CRISPR/Cas9 strategy to delete double strand break repair pathway components, and highlighted that the NHEJ pathway plays a critical role in the formation of chromothriptic chromosomes.³¹ This adds weight to the reports of chromothripsis repair mechanisms based on breakpoint classification approaches as we have deployed here, and supports that the predominant mechanism of repair implicated in chromothripsis is NHEJ, with a small proportion of breakpoint features harboring features indicative of MMBIR or alt-EJ pathways.^{28,32-34}

We have now expanded the methods and results to include the above findings in the manuscript (lines 239-249), and include the findings from clusterSV in Extended Data Table 3:

“We deployed ClusterSV, a structural variant (SV) clustering and classification pipeline (Methods) to identify and classify SVs as simple or complex (≥ 3 interconnected SVs).¹ In 4 of 5 cases (Fig 2A,C,E and Extended Data Fig 2C) the chr21amp event was classed as complex (Ext Data Table 3). In the case classed as a simple amplification event (**Extended Data Fig 2A**), this was demarcated by a fold-back inversion rearrangement in keeping with a breakage fusion bridge cycle.

Review of breakpoint features highlighted that these were frequently characterised by small (0-6bp) insertions, most consistent with NHEJ as the predominant mechanism of repair, as previously reported.^{28,32-34} There was no evidence of templated insertions.”

5. Is the measured gene expression within the amplicon allele-specific? This information would provide further insight into DYRK1A's role in chr21amp. Related to this, the manuscript reports that DYRK1A was the only upregulated gene within the minimally amplified region in chr21amp cells compared to non-chr21amp BP-MPN cells. However, extended transcriptomic analyses identified different candidate genes (DSCR3, MORC3, PIGP, TTC3). Could the authors clearly specify the significance thresholds used for these findings? Additionally, do any of these additional candidate

genes exhibit evidence of allele-specific expression in the analyzed samples?

RESPONSE: We have now performed allele-specific RNA-seq analysis to address this question. We were able to identify informative heterozygous SNPs in the whole genome sequencing data intersected with the RNA-seq data for all genes bar *MORC3*, and in 4 of 5 cases. These are listed in Extended Data Table 4A. We assigned parental haplotypes to these SNPs based on VAF, performed alignment correction of the RNA-seq data and read-counting for each gene to assess for allelic skew. The total transcription at the chr21amp region is highly allele specific, and for every gene and each case examined, there was evidence of allele specific expression from the chromosome 21 amplified haplotype. We have now updated the methods, added the plot demonstrating gene-specific allele-specific expression to Figure 3D and include the results in the manuscript: (lines 347-351) “We [...] investigated the chr21amp RNA-seq dataset to assess for evidence of allele specific expression of the previously prioritised genes. All genes with informative heterozygous SNPs showed a clear read bias towards the amplified allele (Figure 3D, SNP information in Extended Data Table 4A).”

DYRK1A was the only gene both differentially expressed and differentially accessible on mini-bulk RNA-and ATAC-seq generated from sorted CD34+Lin- cells from chr21amp, non-chr21amp and healthy control cells, using a significance threshold of log2fold change of >1 and adjusted p value (adjusted for multiple testing using the Benjamini and Hochberg method) of < 0.05. In a separate cohort of patients who underwent TARGET-seq comparing the chr21amp *TP53* mutant single cells against all others, four additional genes were also differentially expressed between chr21amp cells and controls (*PIGP*, *TTC3*, *MORC3* and *DSCR3*, Supplementary Fig 4 A-D). In this single cell cohort, differentially expressed genes were identified using a combination of the non-parametric Wilcoxon test, to compare the expression values for each group, and Fisher’s exact test, to compare the frequency of expression for each group, as previously described.³⁵ p values were then combined using Fisher’s method, and adjusted p-values derived using the Benjamini & Hochberg procedure. Significant genes were again selected on the basis of a log2(fold change)>1 and adjusted p value < 0.05.

We have now further highlighted this in the methods (lines 1405-1411) and expanded the discussion around the four other possible candidate genes (see response to point 6. below)

6. Among the candidate genes (DYRK1A, DSCR3, MORC3, PIGP, TTC3), which ones are overrepresented in samples from the AML cohort in association with chr21amp?

RESPONSE: We returned to The Cancer Genome Atlas (TCGA) dataset to perform this analysis, and identified that in fact all five genes are upregulated in the 9 patients with a chr21amp amplification event. We believe this additional analysis of TCGA is interesting and now include this analysis in Supp Figure 3E-I and describe in the results on lines 334-337.

DYRK1A was the only gene both differentially expressed and differentially accessible on RNA- and ATAC-seq. However, on single cell analyses of a separate cohort of chr21amp cases, four other genes were also differentially expressed between chr21amp cells and controls (Supplementary Fig 4 A-D). These genes were *PIGP*, *TTC3*, *MORC3* and *DSCR3*. None of these genes show dependency in BPMPN cell lines and they have not previously been implicated in leukemogenesis. We have added some additional discussion on lines 905-912 to acknowledge that it remains possible that these or other additional genes in the minimally amplified region might contribute to the impact of chr21amp and we did not explore their functional role “Although *DYRK1A* was the only gene in the minimally amplified region that was both differentially expressed and differentially accessible in chr21amp BPMPN, it is also important to note that other genes in the minimally amplified region were differentially expressed (*PIGP*, *TTC3*, *MORC3* and *DSCR3*). None of these genes show dependency in BPMPN cell lines and they have not previously been implicated in leukemogenesis, we did not explore their functional role and it remains possible that they might act in concert with *DYRK1A* overexpression.”

7. In the experiments presented in Fig 5A-D and Extended Data Fig 5A-F, where *DYRK1A* knockout and knockdown were assessed in HEL and SET2 BP-MPN cell lines, it would be advisable to include leukemic cell lines lacking chr21amp as controls. Similarly, for the experiments in Fig 5G and Extended Data Fig 5H, which claim selective therapeutic vulnerability in chr21amp patients, control experiments involving BM-MPN patients without the chr21amp event should be included.

RESPONSE: We thank the reviewer for this excellent suggestion, also raised by reviewer 1 (point 13). Publicly available CRISPR screen data from the Broad DepMap database supports that overexpression of *DYRK1A* correlates with dependency as shown in the current Extended Data Figure 4G. DepMap data also shows a significant correlate between *DYRK1A* copy number and gene dependency (higher *DYRK1A* copy number correlates with increased *DYRK1A* dependency). We have specifically labelled K562 in Extended Figure 4G which shows low *DYRK1A* expression and no dependency on *DYRK1A*. A previous study reporting the results of a kinase domain focused CRISPR screen also supports selective dependency of HEL and SET2 cells on *DYRK1A* in comparison with non-*DYRK1A* amplified cell lines such as K562 and OCI AML3.⁷ We have added an additional comment on lines 528-530 of the results “However, myeloid leukemia cell lines with low *DYRK1A* expression did not show dependency on *DYRK1A* e.g. K562 (Extended Data Fig 4G), in line with a previous study.” We have also carried out an additional CRISPR knockout of *DYRK1A* using the same guides as used for SET2 and HEL cells for the non-chromosome 21 amplified cell line OCI AML3. As shown below, the OCI AML3 cell line did not show any evidence of selection over time for edits causing a knockout of *DYRK1A* (also consistent with OCI AML3 data in DepMap). We have not included these data in the revised manuscript as we believe the primary cell data described in the next paragraph are compelling, but would be happy to include additional cell line data at the reviewer’s request.

With regards to primary BPMPN samples, we have now updated figure 5G to include additional cytotoxicity experiments with n=4 primary BPMPN non-chr21amp controls. These important new data show that the cytotoxic effect of DYRK1A inhibition is selective for leukemia cases with chr21amp. These data are shown in Figures 5G and are described in the results section on lines 567-569.

8. The manuscript draws a parallel between chr21amp in BP-MPN and iAMP21 in B-ALL. Notably, it was previously documented that the consensus chromothripsis landscape in iAMP21 closely resembles the copy number profile of chromosome 21, as observed across thousands of cancer samples spanning various cancer types. This prior observation suggests a potential pivotal role of chromothripsis in fine-tuning the copy number landscape of chromosome 21. To bolster the manuscript's insights, it would be helpful if the authors could determine whether chr21amp events exhibit a similar alignment with this "consensus cancer copy number profile of chromosome 21"? This could be potentially assessed statistically, such as by permutation testing. Along similar lines, it would be very helpful if discussions centered around potential communalities and differences between chr21amp and iAMP21 could be extended in this manuscript.

RESPONSE: Thank you for this important point, and for highlighting the data in Figure 4 of Li *et al*, Nature 2014. These data demonstrate that the impact of the iAMP21 chromothripsis event seen in B-ALL on the copy number landscape is non-random – when compared with copy number data over chr21 in thousands of cancers the copy number profile was conserved, suggesting that chromothripsis optimises the copy number landscape for cancer evolution.^{36,37} The figure below highlights that the chr21amp minimally amplified region in BP-MPN (red block) clearly aligns with the regions of chr21 always spared from deletion in Li *et al*³⁸ (navy blocks), and overlies the minimally amplified region in iAMP21 ALL (rose block).³⁹ The mean copy number across all cases (middle panel) mirrors the previously reported consensus cancer copy number profile of chromosome 21³⁸ as shown in the lowest panel of the figure below.

We have now extended the discussion (lines 815-833) around the communalities and differences between iAMP21 and chr21amp, which now reads as follows:

“The chr21amp event we describe in BP-MPN shares some similarities with iAMP21 in B-ALL but also is associated with distinct features. A recent study of 124 iAMP21 pediatric ALL cases identified that iAMP21 is an early, clonal event, comprising a range of patterns of chr21 amplification.³⁹ The copy number profiles observed are similar across the two disease contexts. In iAMP21-ALL, breakage-fusion-bridge cycles are typically the initiating event, often followed by chromothripsis, and in our BP-MPN cohort the boundaries of the amplified regions are similarly frequently demarcated by fold-back inversion rearrangements indicative of breakage-fusion-bridge cycles.^{39,40} Interestingly, the minimally amplified region on chr21 identified in iAMP21-ALL aligns closely with the smaller amplified region seen in BP-MPN, with *DYRK1A* at its center, with transcriptional upregulation of *DYRK1A* also observed in iAMP21-ALL.³⁹ However, there are also important differences between iAMP21-ALL and chr21amp in BP-MPN. In the MPN context, chr21amp arises in HSC as opposed to B-cell progenitors and also occurs as a late event immediately prior to leukemic transformation in the context of an antecedent MPN clone.”

References

1. Giurgiu, M. *et al.* Reconstructing extrachromosomal DNA structural heterogeneity from long-read sequencing data using Decoil. *Genome Res* gr.279123.124 (2024) doi:10.1101/gr.279123.124.
2. Shoshani, O. *et al.* Chromothripsis drives the evolution of gene amplification in cancer. *Nature* **591**, 137–141 (2021).
3. Henssen, A., MacArthur, I., Koche, R., Dorado-García, H. & Henssen, A. Purification and Sequencing of Large Circular DNA from Human Cells. *Protoc Exch* (2019) doi:10.1038/protex.2019.006.
4. Koche, R. P. *et al.* Extrachromosomal circular DNA drives oncogenic genome remodeling in neuroblastoma. *Nat Genet* **52**, 29–34 (2020).
5. Møller, H. D., Parsons, L., Jørgensen, T. S., Botstein, D. & Regenbreg, B. Extrachromosomal circular DNA is common in yeast. *Proceedings of the National Academy of Sciences* **112**, (2015).
6. Jakobsdottir, G. M., Dentre, S. C., Bristow, R. G. & Wedge, D. C. AmplificationTimeR: An R Package for Timing Sequential Amplification Events. *Bioinformatics* (2024) doi:10.1093/bioinformatics/btae281.
7. Tarumoto, Y. *et al.* LKB1, Salt-Inducible Kinases, and MEF2C Are Linked Dependencies in Acute Myeloid Leukemia. *Mol Cell* **69**, 1017-1027.e6 (2018).
8. Bhansali, R. S. *et al.* DYRK1A regulates B cell acute lymphoblastic leukemia through phosphorylation of FOXO1 and STAT3. *Journal of Clinical Investigation* **131**, (2021).
9. Litovchick, L., Florens, L. A., Swanson, S. K., Washburn, M. P. & DeCaprio, J. A. DYRK1A protein kinase promotes quiescence and senescence through DREAM complex assembly. *Genes Dev* **25**, 801–813 (2011).
10. Bujarrabal-Dueso, A. *et al.* The DREAM complex functions as conserved master regulator of somatic DNA-repair capacities. *Nat Struct Mol Biol* (2023) doi:10.1038/s41594-023-00942-8.
11. Lee, Y., Wang, Q., Shuryak, I., Brenner, D. J. & Turner, H. C. Development of a high-throughput γ -H2AX assay based on imaging flow cytometry. *Radiation Oncology* **14**, 150 (2019).
12. Chen, C.-C. *et al.* Inherent genome instability underlies trisomy 21-associated myeloid malignancies. *Leukemia* **38**, 521–529 (2024).
13. KUNDU, J., CHOI, B. Y., JEONG, C.-H., KUNDU, J. K. & CHUN, K.-S. Thymoquinone induces apoptosis in human colon cancer HCT116 cells through inactivation of STAT3 by blocking JAK2- and Src-mediated phosphorylation of EGF receptor tyrosine kinase. *Oncol Rep* **32**, 821–828 (2014).
14. Chen, Y. *et al.* Bt354 as a new STAT3 signaling pathway inhibitor against triple negative breast cancer. *J Drug Target* **26**, 920–930 (2018).
15. Severin *et al.* In Chronic Lymphocytic Leukemia the JAK2/STAT3 Pathway Is Constitutively Activated and Its Inhibition Leads to CLL Cell Death Unaffected by the Protective Bone Marrow Microenvironment. *Cancers (Basel)* **11**, 1939 (2019).
16. Kurabayashi, N., Nguyen, M. D. & Sanada, K. DYRK 1A overexpression enhances STAT activity and astroglialogenesis in a Down syndrome mouse model. *EMBO Rep* **16**, 1548–1562 (2015).
17. Girardot, M. *et al.* Persistent STAT5 activation in myeloid neoplasms recruits p53 into gene regulation. *Oncogene* **34**, 1323–1332 (2015).

18. Wood, T. J. J. *et al.* Specificity of transcription enhancement via the STAT responsive element in the serine protease inhibitor 2.1 promoter. *Mol Cell Endocrinol* **130**, 69–81 (1997).
19. Moucadel, V. & Constantinescu, S. N. Differential STAT5 Signaling by Ligand-dependent and Constitutively Active Cytokine Receptors. *Journal of Biological Chemistry* **280**, 13364–13373 (2005).
20. Defour, J.-P. *et al.* Tryptophan at the transmembrane–cytosolic junction modulates thrombopoietin receptor dimerization and activation. *Proceedings of the National Academy of Sciences* **110**, 2540–2545 (2013).
21. Mascarenhas, J. O. *et al.* Phase 2 study of ruxolitinib and decitabine in patients with myeloproliferative neoplasm in accelerated and blast phase. *Blood Adv* **4**, 5246–5256 (2020).
22. Bose, P. *et al.* A phase 1/2 study of ruxolitinib and decitabine in patients with post-myeloproliferative neoplasm acute myeloid leukemia. *Leukemia* **34**, 2489–2492 (2020).
23. Decitabine With Ruxolitinib or Fedratinib for the Treatment of Accelerated/Blast Phase Myeloproliferative Neoplasms. *NCT04282187*
<https://clinicaltrials.gov/ct2/show/NCT04282187>.
24. Pelleri, M. C. *et al.* Systematic reanalysis of partial trisomy 21 cases with or without Down syndrome suggests a small region on 21q22.13 as critical to the phenotype. *Hum Mol Genet* **ddw116** (2016) doi:10.1093/hmg/ddw116.
25. Chen, C.-C., Amon, A., Hemann, M. & Rowe, R. G. Inherent Genome Instability Underlies Trisomy 21-Associated Myeloid Malignancies. *Blood* **142**, 1388–1388 (2023).
26. Labuhn, M. *et al.* Mechanisms of Progression of Myeloid Preleukemia to Transformed Myeloid Leukemia in Children with Down Syndrome. *Cancer Cell* **36**, 123–138.e10 (2019).
27. Li, Y. *et al.* Patterns of somatic structural variation in human cancer genomes. *Nature* **578**, 112–121 (2020).
28. Cortés-Ciriano, I. *et al.* Comprehensive analysis of chromothripsis in 2,658 human cancers using whole-genome sequencing. *Nat Genet* **52**, 331–341 (2020).
29. Yang, L. *et al.* Diverse Mechanisms of Somatic Structural Variations in Human Cancer Genomes. *Cell* **153**, 919–929 (2013).
30. Kidd, J. M. *et al.* A Human Genome Structural Variation Sequencing Resource Reveals Insights into Mutational Mechanisms. *Cell* **143**, 837–847 (2010).
31. Hu, Q. *et al.* Non-homologous end joining shapes the genomic rearrangement landscape of chromothripsis from mitotic errors. *Nat Commun* **15**, 5611 (2024).
32. Zhang, C.-Z. *et al.* Chromothripsis from DNA damage in micronuclei. *Nature* **522**, 179–184 (2015).
33. Ly, P. *et al.* Selective Y centromere inactivation triggers chromosome shattering in micronuclei and repair by non-homologous end joining. *Nat Cell Biol* **19**, 68–75 (2017).
34. Ly, P. *et al.* Chromosome segregation errors generate a diverse spectrum of simple and complex genomic rearrangements. *Nat Genet* **51**, 705–715 (2019).
35. Giustacchini, A. *et al.* Single-cell transcriptomics uncovers distinct molecular signatures of stem cells in chronic myeloid leukemia. *Nat Med* **23**, 692–702 (2017).

36. Beroukhim, R. *et al.* The landscape of somatic copy-number alteration across human cancers. *Nature* **463**, 899–905 (2010).
37. Kim, T.-M. *et al.* Functional genomic analysis of chromosomal aberrations in a compendium of 8000 cancer genomes. *Genome Res* **23**, 217–227 (2013).
38. Li, Y. *et al.* Constitutional and somatic rearrangement of chromosome 21 in acute lymphoblastic leukaemia. *Nature* **508**, 98–102 (2014).
39. Gao, Q. *et al.* The genomic landscape of acute lymphoblastic leukemia with intrachromosomal amplification of chromosome 21. *Blood Journal* (2023) doi:10.1182/blood.2022019094.
40. Xie, W. *et al.* iAMP21 in acute myeloid leukemia is associated with complex karyotype, TP53 mutation and dismal outcome. *Modern Pathology* **33**, 1389–1397 (2020).